# QDOT: An Efficient Quantile-weighted Distance Metric for Geometric Comparison via Optimal Transport

## Abstract

Measuring the discrepancy between data distributions in heterogeneous metric spaces is a fundamental challenge. Existing methods, typically based on geometric structures, address this by embedding distributions into a shared space. However, these approaches face fundamental limitations, including the loss of geometric information, computationally intractable representations, and inability to preserve essential structural features. In this work, we introduce the Quantile-weighted Distance Optimal Transport (QDOT), a novel and efficient metric for geometric comparison. QDOT constructs a family of isometry-invariant distance representations by leveraging distance quantiles as structural weights in Euclidean space, thereby preserving essential geometric characteristics and enabling optimal transport coupling within a common space. We prove that, under mild conditions, QDOT is a well-defined metric with a convergence rate no slower than the classical Wasserstein distance. Moreover, we present an integral version that computes the loss in complexity of $\mathcal{O}(n \log n)$. Extensive experiments demonstrate that our methods achieves strong performance across diverse applications, including cross-space comparison, transfer learning, and molecule generation, while also achieving state-of-the-art results on several key metrics.

## 1 Introduction

The Wasserstein distance is a powerful metric for comparing probability distributions defined on a same metric space. It has found widespread adoption in a diverse range of machine learning tasks, such as generative models(Arjovsky et al., 2017; De Bortoli et al., 2021; Tong et al., 2024), language models(Kusner et al., 2015; Melnyk et al., 2024), multimodal learning(Xu & Chen, 2023; Alatkar & Wang, 2023; Shi et al., 2024) and reinforcement learning(Klink et al., 2022; Asadulaev et al., 2024). However, a notable limitation of the Wasserstein distance is its sensitivity to the spatial separation of the distributions' supports, meaning that two distributions with similar shapes but far-apart supports can still have a large Wasserstein distance. Furthermore, the applicability of optimal transport is predicated on a pre-defined ground metric between the supports of the distributions. This requirement renders it intractable for comparing distributions in heterogeneous spaces where such a metric is not readily available.

To address these challenges, metrics based on shape features have emerged (Gromov, 1981; Sturm, 2006; Mémoli, 2011). These methods leverage the distribution of distances within a metric space to compare distributions. Among these, the Gromov-Wasserstein (GW) distance (Sturm, 2006; Mémoli, 2011) stands out as a canonical example. It resolves the issue of comparing distributions in disparate spaces by seeking an optimal coupling of points in a shared, albeit implicitly defined, metric space. However, this formulation is a non-convex optimization problem, the structure of the latent space is not explicitly constructed, and its computation is prohibitively expensive, hindering its use on practical datasets. A related approach, EMD under Transformation Sets (EMD$^{\mathcal{G}}$) (Cohen & Guibasm, 1999), extends the GW concept to Euclidean spaces of the same dimension by finding an optimal orthogonal transformation. However, it is not applicable for cross-space comparisons and is susceptible to converging to local optima. Faster approximations, such as the Invariant Sliced Gromov-Wasserstein (RISGW) (Titouan et al., 2019), achieve a favorable $\mathcal{O}(n \log n)$ complexity, but at the cost of sacrificing theoretical guarantees, such as key metric properties.

Another line of work focuses on extracting shape-invariant features (Belongie et al., 2002; Sun et al., 2019; Yang et al., 2016). These methods map distributions from different spaces into a common feature space for comparison. However, they often suffer from unavoidable information loss due to the unidirectional nature of the feature mapping. To capture richer information, subsequent approaches have employed deep neural networks (Chen et al., 2019; Kim et al., 2020) or transformer based representations (Fuchs et al., 2020; Yu et al., 2023, TBR) for comparison. A key limitation is that these models are typically trained for specific tasks and thus lack generalizability.

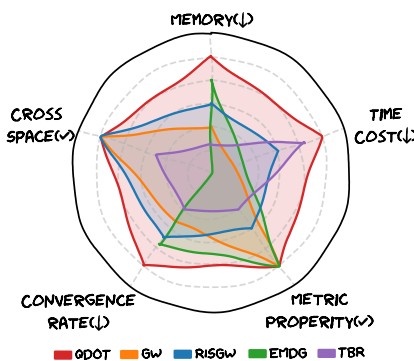

Figure 1: Comparison of various geometric metrics. Lower is better for memory, time cost, and convergence rate. We also evaluate metric properties and cross-space capabilities.

To the best of our knowledge, the existing geometric comparison methods can be unified under a common paradigm: they map distributions into a shared space for comparison. However, the mappings employed by these methods often suffer from critical issues: (1) they lack an explicit form; (2) they lead to significant information loss; or (3) they introduce non-geometric artifacts that can corrupt the comparison.

To overcome these challenges, we introduce QDOT, a novel framework that constructs an explicit, geometry-based representation derived from quantile-weighted distances. This principled construction guarantees the integrity of the intrinsic distance information. As illustrated in Figure 1, QDOT strikes a highly effective balance between theoretical properties, generalization capability, time complexity, and computational resource usage, addressing the key trade-offs that limit existing methods. The key contributions of QDOT can be summarized as follows:

**Theoretical Guarantees.** We prove that QDOT and its integral version constitute a metric on the space of isometry classes under certain conditions. Furthermore, we establish that their sample convergence rate is at least as fast as that of the Wasserstein distance.

**Computational Efficiency.** We propose a highly efficient algorithm. The computation of our representations has a complexity of only $\mathcal{O}(n \log n)$. The subsequent comparison step for the full QDOT requires an additional $\mathcal{O}(n^2 \log n)$ for a standard Wasserstein distance calculation. In contrast, its integral variant, IQDOT, leverages the closed-form solution of one-dimensional OT to achieve an overall complexity of just $\mathcal{O}(n \log n)$. Consequently, our family of methods offers a highly efficient, quasi-linear time solution.

**Versatility and Strong Performance.** Our method achieved strong performance across a diverse range of experiments. In point cloud comparison tasks, it accurately captured the geometric dissimilarities between distributions and produced high-quality alignments, showcasing its powerful cross-space capabilities. Furthermore, leveraging its computational efficiency, our method achieved excellent results in a large-scale transfer learning scenario. Finally, when integrated as a loss function for a molecular generation model, the QDOT loss significantly enhanced model generalization and achieved state-of-the-art performance on multiple key metrics.

## 2 PRELIMINARY

To establish a rigorous framework, we begin with the fundamental definitions. For any metric space $(X, d_X)$, we can define its Borel $\sigma$-algebra, denoted $\mathcal{B}(X)$. For any probability measure $\mu_X$ defined on the measurable space $(X, \mathcal{B}(X))$, its support, written as $\text{supp}(\mu_X)$, is the smallest closed set $C \subseteq X$ such that $\mu_X(C) = 1$. For a measurable map $f : X \to Y$, the *push-forward measure* $f_{\#}\mu_X$ on $(Y, \mathcal{B}(Y))$ is defined by $f_{\#}\mu_X(A) := \mu_X(f^{-1}(A))$ for any set $A \in \mathcal{B}(Y)$. For two measures $\mu_X$ and $\mu_Y$ in spaces $X$ and $Y$, a joint probability $\pi$ on the product space $X \times Y$ is a *coupling* if its marginals satisfy $(\text{proj}_X)_{\#}\pi = \mu_X$ and $(\text{proj}_Y)_{\#}\pi = \mu_Y$. When $\mu_X$ and $\mu_Y$ are defined on the same metric space $(\Omega, d)$, they can be compared using the Wasserstein distance (Villani et al., 2008), which is defined as:

$$\mathcal{W}_p(\mu_X, \mu_Y) := \left( \inf_{\pi \in \Pi(\mu_X, \mu_Y)} \int_{\Omega \times \Omega} d(x, y)^p \, d\pi(x, y) \right)^{\frac{1}{p}}. \tag{1}$$

The primary challenge arises when comparing distributions defined on different metric spaces. To handle this scenario, the measurable metric space(Mémoli, 2011, mm-space) is defined as the triplet $\mathcal{X} = (X, d_X, \mu_X)$. For two distinct mm-spaces, $\mathcal{X} = (X, d_X, \mu_X)$ and $\mathcal{Y} = (Y, d_Y, \mu_Y)$, Sturm (2006) proposed the Gromov-Wasserstein distance, extending the concept from equation 1:

$$\mathfrak{S}_p(\mathcal{X}, \mathcal{Y}) := \inf_{Z} \mathcal{W}_p(f(\mathcal{X}), g(\mathcal{Y})), \tag{2}$$

where the maps $f : X \rightarrow Z$ and $g : Y \rightarrow Z$ are isometric embeddings, satisfying $d_Z(f(x), f(x')) = d_X(x, x'), \forall x, x' \in \operatorname{supp}(\mu_X)$ and $d_Z(g(y), g(y')) = d_Y(y, y'), \forall y, y' \in \operatorname{supp}(\mu_Y)$. Intuitively, the definition in equation 2 can be understood as finding an optimal "lossless" projection of the two disparate spaces $X$ and $Y$ into a common latent space $Z$, within which their Wasserstein distance can be computed. A significant contribution of the Gromov-Wasserstein distance is that it provides a complete metric on the space of mm-spaces. This metric is formally defined as follows.

**Definition 1 (Metric on Isometry Classes of mm-spaces)** *For two mm-spaces $\mathcal{X} = (X, d_X, \mu_X)$ and $\mathcal{Y} = (Y, d_Y, \mu_Y)$, a map $f : X \rightarrow Y$ is called an isometry if it is a surjection satisfying $d_Y(f(x), f(x')) = d_X(x, x'), \forall x, x' \in \operatorname{supp}(\mu_X)$. Two mm-spaces are considered isometric if such an isometry exists between them. A function $\mathcal{L}$ that measures the dissimilarity between two mm-spaces is a metric on the isometry classes of mm-spaces if it satisfies: (1) **Identity of Indiscernibles:** $\mathcal{L}(\mathcal{X}, \mathcal{Y}) \geq 0$, and $\mathcal{L}(\mathcal{X}, \mathcal{Y}) = 0$ iff $\mathcal{X}$ and $\mathcal{Y}$ are isometric; (2) **Symmetry:** $\mathcal{L}(\mathcal{X}, \mathcal{Y}) = \mathcal{L}(\mathcal{Y}, \mathcal{X})$; (3) **Triangle Inequality:** $\mathcal{L}(\mathcal{X}, \mathcal{Y}) \leq \mathcal{L}(\mathcal{X}, \mathcal{Z}) + \mathcal{L}(\mathcal{Y}, \mathcal{Z})$.*

Inspired by the structure of the Gromov-Wasserstein distance in equation 2, we summarized those methods for comparing two mm-spaces can be expressed in a general form:

$$\mathcal{L}_p(\mathcal{X}, \mathcal{Y}) = \mathcal{W}_p(f(\mathcal{X}), g(\mathcal{Y})) \tag{3}$$

where $f : X \rightarrow Z$ and $g : Y \rightarrow Z$ are mapping functions into a common space $Z$. However, constructing such maps $f$ and $g$ that yield a valid metric satisfying Definition 1 imposes two key requirements. First, the maps must be **isometry-invariant**. That is, if mm-spaces $\mathcal{X}_1$ and $\mathcal{X}_2$ are isometric, their representations must be identically distributed, i.e., $f_{\#\mu_{X_1}} = f_{\#\mu_{X_2}}$. Second, to satisfy the identity of indiscernibles property, the representation must be **information-preserving**; it must uniquely encode the metric structure of the original space such that non-isometric spaces map to distinct distributions.

However, the requirement of isometry invariance often entails an unavoidable loss of structural information, and existing methods conforming to the structure of equation 3 fail to meet both requirements simultaneously, thereby falling short of constituting a well-defined metric.

Our work introduces a novel framework designed to explicitly resolve this conflict. Our work introduces a novel framework designed to explicitly resolve this conflict. To preserve information, we draw inspiration from the principle of trilateration. As illustrated in Figure 2, this principle dictates that the location of a target point can be uniquely determined from its distances to a sufficient number of known anchor points. This implies that the distance information can be fully characterized simply by the distances to specific anchor points. Another key problem is how to choose the anchors such that they are isometry-invariant. To address this issue, we introduce the Quantile-weighted Distance Mean (QDM), which utilizes a family of isometry-invariant weights to compute distinct means serving as the distribution's anchors, thereby ensuring isometry invariance between the anchors of different distributions. Consequently, the collection of distances from all points in the support to these QDMs

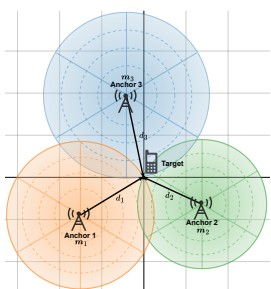

Figure 2: Illustration of trilateration

forms a new representation that effectively captures the intrinsic distance information of the entire distribution. In the following sections, we will formally introduce how to determine such isometry-invariant weights and the corresponding QDMs, and detail their theoretical properties.

## 3 PROPOSED METHOD

For a mm-space $\mathcal{X} = (X, d_X, \mu_X)$, its Barycenter $b_X$ is defined as $\operatorname{argmin}_{x \in X} \mathbb{E}_{\mu_X}\left(d_X^2(x, X)\right)$.

Note that, since it minimizes the aggregate squared distance to the distribution, the barycenter is isometry-invariant; consequently, the distribution of distances with respect to the Barycenter is also isometry-invariant. Specifically, if $\mathcal{X}$ and $\mathcal{Y}$ are isometric, then the corresponding distributions of distances to the Barycenter, denoted as $d_X(\cdot, b_X)_{\#\mu_X}$ and $d_Y(\cdot, b_Y)_{\#\mu_Y}$, must be identical. In Euclidean space, the Barycenter has an explicit solution corresponding to the mean of the distribution. Therefore, in the subsequent analysis, for two mm-spaces $\mathcal{X}$ and $\mathcal{Y}$ in Euclidean space, we assume without loss of generality that they are centered, and we utilize the norm as the distance to the Barycenter.

## 3.1 QUANTILE DISTANCE-WEIGHTED OPTIMAL TRANSPORT

Given a mm-space $\mathcal{X}$, since the distribution of its norm $\| \cdot \|_{2\#\mu_X}$ is isometry-invariant, to derive a family of distinct isometry-invariant Quantile Weights and QDMs, we employ Gaussian kernels centered at distinct quantiles of $\| \cdot \|_{2\#\mu_X}$ to perform weighting as follows:

**Definition 2 (Quantile-weighted Distance Mean (QDM))** *Given a centered mm-space $\mathcal{X} = (X, d_X, \mu_X)$, where $X \subseteq \mathbb{R}^p$ and the metric is the standard Euclidean distance, $d_X(\boldsymbol{x}_1, \boldsymbol{x}_2) = \|\boldsymbol{x}_1 - \boldsymbol{x}_2\|_2$, we can derive the distribution of its norms, denoted by $\mu_{\|X\|_2} = \| \cdot \|_{2\#\mu_X}$. Let $F_{\|X\|_2}$ denote the cumulative distribution function (CDF) of $\mu_{\|X\|_2}$, and let $F_{\|X\|_2}^{-1}$ be its associated quantile function. For any quantile level $q \in (0, 1)$, we define the Quantile Distance Weight function $w : X \times (0,1) \to \mathbb{R}$ as*

$$w^X(\boldsymbol{x}, q) := e^{-\sigma\left(\|\boldsymbol{x}\|_2 - F_{\|X\|_2}^{-1}(q)\right)^2}, \tag{4}$$

*where $\sigma$ is a bandwidth parameter. Based on these weights, the corresponding QDM $\boldsymbol{m}^X : (0,1) \to \mathbb{R}^p$ is defined as the weighted mean:*

$$\boldsymbol{m}^X(q) := \frac{\mathbb{E}_{\mu_X}[w^X(X, q)X]}{\mathbb{E}_{\mu_X}[w^X(X, q)]}. \tag{5}$$

Intuitively, for a given quantile level $q$, the weight function $w^X(\boldsymbol{x}, q)$ is concentrated on points whose norms are close to the $q$-th quantile of the norm distribution. Therefore, the QDM can be served as a canonical, isometry-invariant anchor point. We use it to define the Quantile Distance-weighted Mean Distance (QDMD), which is a function $\phi : X \times (0,1) \to \mathbb{R}$ that measures the distance from a point to the QDM:

$$\phi^X(\boldsymbol{x}, q) = d_X(\boldsymbol{x}, \boldsymbol{m}^X(q)). \tag{6}$$

Given a quantile level vector $\boldsymbol{q} = (q_1, \ldots, q_k) \in (0,1)^k$. The corresponding QDMD maps each point $\boldsymbol{x}$ to a feature vector in $\mathbb{R}^{k+1}$:

$$\boldsymbol{\phi}^X(x, \mathbf{q}) := [\phi_0^X(x), \phi^X(x, q_1), \phi^X(x, q_2), \ldots, \phi^X(x, q_k)],$$

where $\phi_0^X(x) = \|x\|_2$ is the original norm, included as a fundamental reference distance to the origin. This mapping transforms the measure $\mu_X$ into a push-forward measure on $\mathbb{R}^{k+1}$. Based on this transformation, we define the Quantile-weighted Distance Optimal Transport as follows:

**Definition 3 (Quantile-weighted Distance Optimal Transport (QDOT))** *Let $\mathcal{X} = (X, d_X, \mu_X)$ and $\mathcal{Y} = (Y, d_Y, \mu_Y)$ be two mm-spaces, where $X \subseteq \mathbb{R}^d$ and $Y \subseteq \mathbb{R}^s$, and their metrics $d_X$ and $d_Y$ are the standard Euclidean distances. For a given vector of quantile levels $\mathbf{q} \in (0,1)^k$, we compute their corresponding QDMD representations, $\boldsymbol{\phi}^X$ and $\boldsymbol{\phi}^Y$. The QDOT distance is then defined as the Wasserstein distance between the resulting push-forward measures:*

$$\mathcal{QD}_p(\mathcal{X}, \mathcal{Y}) = \mathcal{W}_p(\boldsymbol{\phi}_{\#\mu_X}^X, \boldsymbol{\phi}_{\#\mu_Y}^Y). \tag{7}$$

The overall procedure of the QDOT framework is illustrated in Figure 7.

## 3.2 THEORETICAL RESULTS

We will now establish the key theoretical properties of our proposed QDOT framework, demonstrating that it is a well-defined metric with favorable sample convergence guarantees.

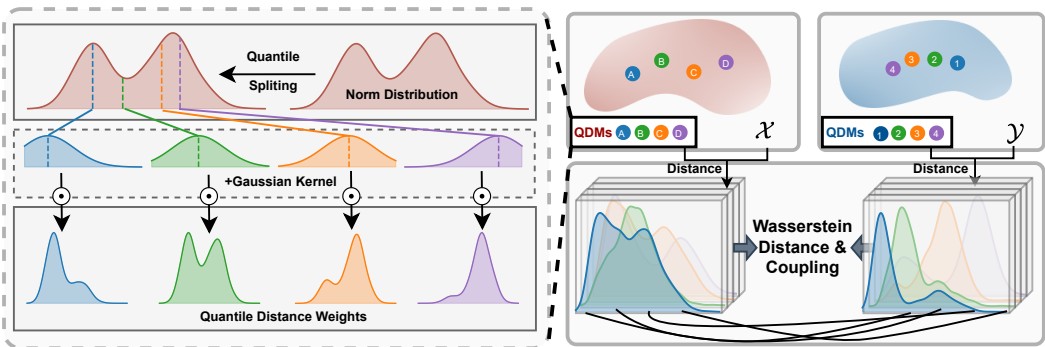

Figure 3: Illustration of the QDOT framework. The method first computes canonical QDM anchors for each distribution, then generates QDMD vector representations based on distances to these anchors. The final QDOT distance is the Wasserstein distance between these two resulting representation distributions.

**Theorem 1 (Metric Property on Isometry Classes)** *Let $\mathcal{X} = (X, d_X, \mu_X)$ and $\mathcal{Y} = (Y, d_Y, \mu_Y)$ be two mm-spaces embedded in Euclidean spaces. For a given quantile level vector $\mathbf{q} \in (0, 1)^k$, suppose their corresponding QDMs satisfy the dimensionality condition $\dim(\{\boldsymbol{m}^X(q_i)\}_{1 \le i \le k}) = \dim(X)$ or $\dim(\{\boldsymbol{m}^Y(q_i)\}_{1 \le i \le k}) = \dim(Y)$. Then, the $p$-QDOT distance, $\mathcal{QD}_p$, defines a metric on the isometry classes of these mm-spaces.*

The condition on dimensionality is inspired by the principle of trilateration (Thomas & Ros, 2005) and is generally satisfied in practical scenarios. The proof for Theorem 1 is provided in Appendix B.1. In addition to its metric properties, we also prove that QDOT has an empirical convergence rate.

**Theorem 2 (Empirical Convergence Rate)** *Let $\mathcal{X} = (X, d_X, \mu_X)$ be an mm-space where $X \subseteq \mathbb{R}^d$. Let $\{x_i\}_{i=1}^n$ be i.i.d. samples drawn from $\mu_X$, and let the empirical measure be defined as $\mu_n := \frac{1}{n} \sum_{i=1}^n \delta_{x_i}$. If we denote the corresponding empirical mm-space as $\mathcal{X}_n = (\{x_1, \ldots, x_n\}, d_X, \mu_n)$, then we have $\mathcal{QD}_p(\mathcal{X}, \mathcal{X}_n) \to 0$, as $n \to \infty$.*

*Moreover, under additional regularity conditions, namely (1) the distribution has a finite fourth moment and (2) its CDF is continuous at the chosen quantile levels and has a strictly positive density, the convergence rate of $\mathcal{QD}_p$ remains consistent with the standard rate for the Wasserstein distance (Fournier & Guillin, 2015). Specifically, for the common case where $d \ge 2$, the expected error is bounded by:*

$$\mathbb{E}[\mathcal{QD}_p(\mathcal{X}, \mathcal{X}_n)] = O(n^{-1/d}).$$

The finite moment requirement is a standard condition inherited from Fournier & Guillin (2015), while the continuity condition is a classical assumption that ensures the consistency of empirical quantiles. Both of these conditions are mild and hold for the vast majority of practical data distributions. For a detailed discussion and rates under other conditions, we refer the reader to Fournier & Guillin (2015) and our proof in Appendix B.2.

Based on the preceding theoretical results, we can summarize several key properties of the QDOT.

**Corollary 1** *Let $\mathcal{X} = (X, d_X, \mu_X)$ and $\mathcal{Y} = (Y, d_Y, \mu_Y)$ be two mm-spaces embedded in Euclidean spaces, where $X \subseteq \mathbb{R}^d$ and $Y \subseteq \mathbb{R}^s$. The QDOT distance exhibits the following properties:*

*(1) **Location Invariance.** For any translation vector $z \in \mathbb{R}^d$, let the translated mm-space be denoted by $\mathcal{X}_z = (X + z, d_X, \mu_{X+z})$, we have $\mathcal{QD}_p(\mathcal{X}_z, \mathcal{Y}) = \mathcal{QD}_p(\mathcal{X}, \mathcal{Y})$.*

*(2) **Rotation and Reflection Invariance.** For any orthogonal transformation $R : \mathbb{R}^d \to \mathbb{R}^d$, let $\mathcal{X}_R = (R(X), d_{R(X)}, R_{\#\mu_X})$, we have $\mathcal{QD}_p(\mathcal{X}_R, \mathcal{Y}) = \mathcal{QD}_p(\mathcal{X}, \mathcal{Y})$.*

*(3) **Numerical Convergence.** Let $\mathcal{X}_n$ and $\mathcal{Y}_n$ be the empirical mm-spaces constructed from $n$ i.i.d. samples drawn from $\mathcal{X}$ and $\mathcal{Y}$, respectively. The empirical QDOT distance converges*

*as $n \to \infty$: $\mathcal{QD}_p(\mathcal{X}_n, \mathcal{Y}_n) \to \mathcal{QD}_p(\mathcal{X}, \mathcal{Y})$. Furthermore, under the condition in 2, its expectation convergence rate is given by $\mathbb{E}|\mathcal{QD}_p(\mathcal{X}_n, \mathcal{Y}_n) - \mathcal{QD}_p(\mathcal{X}, \mathcal{Y})| = \mathcal{O}(n^{-\frac{1}{\max(d,s)}})$.*

## 3.3 NUMERICAL IMPLEMENTATION

In the numerical setting, we consider discrete probability measures. Let $\Delta^{n-1}$ denote the $(n-1)$-simplex. Given two sets of samples, $X_n = \{\boldsymbol{x}_i\}_{1 \leq i \leq n}$ in $\mathbb{R}^d$ and $Y_m = \{\boldsymbol{y}_j\}_{1 \leq j \leq m}$ in $\mathbb{R}^s$, we represent the sample sets as data matrices $\mathbf{X} = (\boldsymbol{x}_1, \boldsymbol{x}_2, \cdots, \boldsymbol{x}_n)^\top \in \mathbb{R}^{n \times d}$ and $\mathbf{Y} = (\boldsymbol{y}_1, \boldsymbol{y}_2, \cdots, \boldsymbol{y}_m)^\top \in \mathbb{R}^{m \times s}$. The corresponding probability vectors are denoted as $\boldsymbol{p}^{\mathbf{X}} \in \Delta^{n-1}$ and $\boldsymbol{p}^{\mathbf{Y}} \in \Delta^{m-1}$, and their associated discrete measures are $\mu_{\mathbf{X}} = \sum_{i=1}^n p_i^{\mathbf{X}} \delta_{\boldsymbol{x}_i}$ and $\mu_{\mathbf{Y}} = \sum_{j=1}^m p_j^Y \delta_{\boldsymbol{y}_j}$, respectively. For a given quantile level vector $\boldsymbol{q} \in (0,1)^k$, the procedure for computing the QDOT distance is detailed in Algorithm 1.

---

**Algorithm 1** QDOT

---

**Require:** $\mathbf{X}, \mathbf{Y}, \boldsymbol{p}^{\mathbf{X}}, \boldsymbol{p}^{\mathbf{Y}}, \boldsymbol{q}$
1: Initialize the data matrices $\mathbf{X}$ and $\mathbf{Y}$.
2: Compute the sample norms of $\mathbf{X}$ and $\mathbf{Y}$: $\boldsymbol{\phi}_0^{\mathbf{X}} \leftarrow (\|\boldsymbol{x}_i\|_2)_{1 \leq i \leq n}, \boldsymbol{\phi}_0^{\mathbf{Y}} \leftarrow (\|\boldsymbol{y}_j\|_2)_{1 \leq j \leq m}$.
3: Compute the $\boldsymbol{q}$-quantiles of the sample norms: $\quad\quad\quad\quad\quad\quad\quad \triangleright \mathcal{O}(n \log n + m \log m)$
$\quad\quad \boldsymbol{r}^{\mathbf{X}} \leftarrow (F_{\|\mathbf{X}\|_2}^{-1}(q_1), \ldots, F_{\|\mathbf{X}\|_2}^{-1}(q_k)), \quad \boldsymbol{r}^{\mathbf{Y}} \leftarrow (F_{\|\mathbf{Y}\|_2}^{-1}(q_1), \ldots, F_{\|\mathbf{Y}\|_2}^{-1}(q_k)).$
4: **for** $i \leftarrow 1$ to $k$ **do** $\quad\quad\quad\quad\quad\quad\quad\quad\quad\quad\quad\quad\quad\quad\quad\quad \triangleright \mathcal{O}(knd + kms)$
5: $\quad$ Compute the quantile weights:
$\quad\quad \boldsymbol{w}_{ij}^{\mathbf{X}} \leftarrow \frac{p_j^X \exp\{-\sigma(\boldsymbol{d}_j^{\mathbf{X}} - \boldsymbol{r}_i^{\mathbf{X}})^2\}}{\sum_{j'=1}^n p_{j'}^X \exp\{-\sigma(\boldsymbol{d}_{j'}^{\mathbf{X}} - \boldsymbol{r}_i^{\mathbf{X}})^2\}}, \quad \boldsymbol{w}_{ij}^{\mathbf{Y}} \leftarrow \frac{p_j^Y \exp\{-\sigma(\boldsymbol{d}_j^{\mathbf{Y}} - \boldsymbol{r}_i^{\mathbf{Y}})^2\}}{\sum_{j'=1}^m p_{j'}^Y \exp\{-\sigma(\boldsymbol{d}_{j'}^{\mathbf{Y}} - \boldsymbol{r}_i^{\mathbf{Y}})^2\}}$
6: $\quad$ Compute the $q_i$-quantile means: $\boldsymbol{m}_i^{\mathbf{X}} \leftarrow \mathbf{X}^\top \boldsymbol{w}_i^{\mathbf{X}}, \quad \boldsymbol{m}_i^{\mathbf{Y}} \leftarrow \mathbf{Y}^\top \boldsymbol{w}_i^{\mathbf{Y}}$
7: $\quad$ Compute the distances to quantile means:
$\quad\quad \boldsymbol{\phi}_i^{\mathbf{X}} \leftarrow (\|\boldsymbol{x}_j - \boldsymbol{m}_i^{\mathbf{X}}\|_2)_{1 \leq j \leq n}, \quad \boldsymbol{\phi}_i^{\mathbf{Y}} \leftarrow (\|\boldsymbol{y}_j - \boldsymbol{m}_i^{\mathbf{Y}}\|_2)_{1 \leq j \leq m}$
8: **end for**
9: Concatenate representations: $\boldsymbol{\Phi}^{\mathbf{X}} \leftarrow [\boldsymbol{\phi}_0^{\mathbf{X}}, \boldsymbol{\phi}_1^{\mathbf{X}}, \ldots, \boldsymbol{\phi}_k^{\mathbf{X}}], \quad \boldsymbol{\Phi}^{\mathbf{Y}} \leftarrow [\boldsymbol{\phi}_0^{\mathbf{Y}}, \boldsymbol{\phi}_1^{\mathbf{Y}}, \ldots, \boldsymbol{\phi}_k^{\mathbf{Y}}]$
10: $\mathcal{W}_p$ Computation: $\mathcal{QD}_p, \Pi_{\mathcal{QD}_p} \leftarrow \mathcal{W}_p\left((\boldsymbol{\Phi}^{\mathbf{X}})_{\#}\mu_X, (\boldsymbol{\Phi}^{\mathbf{Y}})_{\#}\mu_Y\right) \quad \triangleright \mathcal{O}(n^2 \log n + m^2 \log m)$
11: **return** $\mathcal{QD}_p, \Pi_{\mathcal{QD}_p}$

---

**Computational Cost.** For simplicity, we assume $m \leq n$ and $s \leq d$. The initial norm computation requires $\mathcal{O}(nd)$ operations. Computing the quantiles takes $\mathcal{O}(n \log n)$ time. The main loop for computing the QWs, QDMs, and QDMDs has a total cost of $\mathcal{O}(knd)$. It is noteworthy that the final representations are generated in nearly linear time with respect to $n$. The final step of calculating the Wasserstein distance has a complexity of $\mathcal{O}(n^2 \log n)$ by Sinkhorn Algorithm(Cuturi, 2013) or Earth Moving Distance(Rubner et al., 2000).

To mitigate the high complexity of the final step, we also propose an Integral-QDOT approach.

## 3.4 INTEGRAL-QDOT

To address the high computational cost of the standard Wasserstein distance, methods based on slicing(Bonneel et al., 2015; Deshpande et al., 2019) and the closed-form solution of one-dimensional Optimal Transport have become increasingly popular. We observe that the QDMD representation, $\phi^X(x, q)$, defined in the previous section, is a scalar value for any given quantile level $q$. This structure naturally inspires an alternative approach: instead of comparing the multi-dimensional representations in $\mathbb{R}^{k+1}$, we can compare the one-dimensional distributions of the QDMD scalars for each $q$ and then aggregate the results. This leads to the Integral-QDOT(IQDOT) approach.

**Definition 4 (Integral-QDOT)** *Given two mm-spaces $\mathcal{X} = (X, d_X, \mu_X)$ and $\mathcal{Y} = (Y, d_Y, \mu_Y)$, where $X \subseteq \mathbb{R}^d, Y \subseteq \mathbb{R}^s$, and $d_X, d_Y$ are the standard Euclidean distances. For any quantile level $q \in (0,1)$, we can obtain their corresponding scalar QDMD representations, $(\phi^X(\cdot, q))_{\#\mu_X}$ and $(\phi^Y(\cdot, q))_{\#\mu_Y}$. The Integral-QDOT distance is then defined as the $L_p$-norm of the 1-D Wasserstein*

*distances between these push-forward measures, integrated over all $q \in (0, 1)$:*

$$\mathcal{IQD}_p(\mathcal{X}, \mathcal{Y}) := \inf_{\pi \in \Pi(\mu_X, \mu_Y)} \left( \int_{(0,1)} \int_{X \times Y} |\phi^X(x, q) - \phi^Y(y, q)|^p \, d\pi(x, y) \, dq \right)^{1/p}. \quad (8)$$

We further establish that IQDOT also constitutes a well-defined metric.

**Theorem 3 (Metric Property of IQDOT)** *Let $\mathcal{X} = (X, d_X, \mu_X)$ or $\mathcal{Y} = (Y, d_Y, \mu_Y)$ be two mm-spaces embedded in Euclidean spaces. If the set of QDMs satisfies the dimensionality condition* $\dim(\{\boldsymbol{m}^X(q)\}_{q \in (0,1)}) = \dim(X)$ *and* $\dim(\{\boldsymbol{m}^Y(q)\}_{q \in (0,1)}) = \dim(Y)$, *then* $\mathcal{IQD}_p$ *defines a metric on the isometry classes of these mm-spaces.*

As for the numerical implementation of IQDOT, we consider computing the mean of the one-dimensional Wasserstein distances for the QDMD representations corresponding to $k$ quantiles. Since the 1d Wasserstein distance has a closed-form solution, the computational cost of the final step in Algorithm 1 is reduced to $\mathcal{O}(kn \log n)$, thereby achieving highly efficient computation. The specific implementation of IQDOT is detailed in Algorithm 2 in Appendix C.2.

## 4 EXPERIMENTS

We next present a series of experiments designed to empirically validate several key properties of our proposed framework. The evaluation aims to demonstrate QDOT's: (1) effectiveness in cross-space tasks; (2) fast computational efficiency; (3) transferability and versatility in comparing diverse distributions; and (4) strong performance in complex models. All CPU-based experiments were conducted on an Intel(R) Xeon(R) Platinum 8280 CPU @ 2.70GHz with 256GB RAM. All GPU-based experiments were performed on a single NVIDIA RTX 4090 GPU with 24GB VRAM.

### 4.1 CROSS SPACE TASKS

To evaluate the cross-space alignment capability and the metric accuracy of our proposed method, we conduct experiments on the camel-gallop data(Sumner & Popović, 2004). This data consists of a reference 3D point cloud model and a corresponding 48-frame sequence of a galloping camel. For our tests, we subsample the point clouds to 10,000 points. The experimental task is to match each frame of the 3D sequence against projections of the static reference model onto three distinct 2D subspaces. We assess alignment quality using the Transformed Mean Squared Error (TMSE) and the Inlier Ratio (IR), formally defined in Appendix D.2. TMSE measures the alignment cost in the original 3D space, while IR quantifies the percentage of correctly matched points within a given tolerance. Our comparisons are structured as follows: First, we evaluate our QDOT by Sinkhorn algorithm, against Entropic Gromov-Wasserstein (EGW) (Peyré et al., 2016), testing both methods with regularization parameters $\lambda = 0.1$ and $\lambda = 0.01$. Second, to assess performance with sparse couplings, we compare QDOT by EMD against the classical GW method.

Table 1: Cross Space Results on the camel-gallop dataset

| Methods | *Transformed MSE* ↓ | | | | | *Inlier Ratio (%)* ↑ | | | | | $Time_{(\times 10^2 s)}$ |
|---|---|---|---|---|---|---|---|---|---|---|---|
| | 3D | 2D$_{1st}$ | 2D$_{2nd}$ | 2D$_{3rd}$ | Avg. | 3D | 2D$_{1st}$ | 2D$_{2nd}$ | 2D$_{3rd}$ | Avg. | |
| EGW$_{(\lambda=0.1)}$ | 0.37 | 0.37 | 0.39 | 0.40 | 0.38 | 41.71 | 41.90 | 39.66 | 38.69 | 40.49 | 3.03 |
| **QDOT$_{(Sink-0.1)}$** | **0.32** | **0.32** | **0.32** | **0.33** | **0.32** | **48.75** | **48.25** | **48.84** | **47.03** | **48.22** | **0.13** |
| EGW$_{(\lambda=0.01)}$ | **0.21** | **0.23** | **0.22** | 0.33 | 0.25 | **72.12** | **64.92** | **69.51** | 50.89 | 64.36 | 21.51 |
| **QDOT$_{(Sink-0.01)}$** | 0.22 | 0.24 | 0.23 | **0.25** | **0.24** | 71.15 | 63.75 | 68.48 | **63.20** | **66.65** | **0.74** |
| GW | 0.26 | **0.25** | 0.30 | 0.35 | 0.29 | 61.29 | **63.04** | 56.78 | 49.16 | 58.06 | 14.63 |
| **QDOT$_{(EMD)}$** | **0.25** | 0.27 | **0.25** | **0.27** | **0.26** | **63.35** | 58.60 | **63.87** | **60.02** | **61.46** | **0.38** |

As shown in Table 1, the couplings produced by QDOT achieve comparable or superior results to those from GW and EGW. Notably, this performance is attained with a computational cost that is up to 30 times lower, underscoring the effectiveness and efficiency of our algorithm for cross-space alignment. Furthermore, to evaluate the ability of cross-space metrics, we plotted the dissimilarity trends over the point-cloud sequence, and included IQDOT, SGW, and RISGW for comparison.

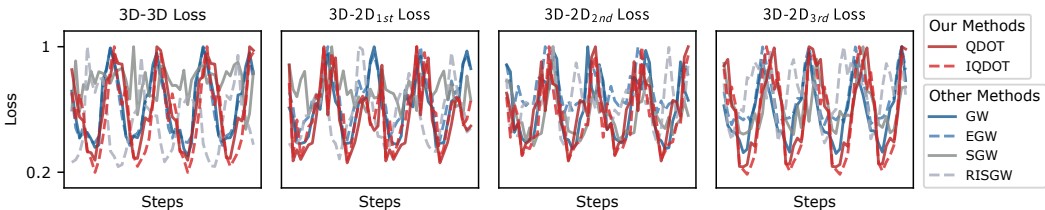

Figure 4: Loss trends between the camel-gallop sequence and the four reference in different spaces. It is evident that QDOT, IQDOT, GW, and EGW successfully capture the four periodic cycles of the camel's gallop, exhibiting consistent fluctuation amplitudes and trends across all reference spaces. In contrast, SGW and RISGW fail to reveal a clear, meaningful pattern.

The results, visualized in Figure 4, demonstrate that methods possessing well-defined metric properties consistently produce effective and meaningful dissimilarity curves. In stark contrast, SGW and RISGW, which lack these theoretical guarantees, fail to reveal this underlying geometric pattern, underscoring the importance of sound metric properties.

## 4.2 TIME COST

To benchmark the practical runtime performance of our methods, we conduct an experiment using randomly generated 2D point clouds, with the number of support points ranging from $10^2$ to $10^6$. We compare our methods against GW, EGW, SGW, and RISGW. For this test, all methods are configured with 50 projections (or quantiles); IQDOT is additionally implemented using 5 quantiles, and QDOT uses an EMD solver. The results are visualized in Figure 5. The plot clearly shows that for smaller sample sizes, the QDOT family of methods is faster than all baselines. In large-scale scenarios involving tens of thousands of points, IQDOT achieves a remarkable speedup, running up to thousands of times faster than the classic GW method. QDOT also maintains a significant advantage. The overall trends in the log-log plot confirm our complexity analysis: IQDOT exhibits a quasi-linear time complexity, whereas QDOT scales quadratically.

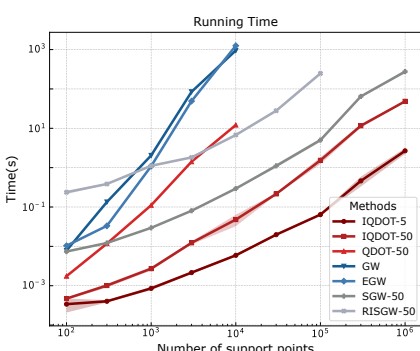

Figure 5: Runtime comparison of various methods. All experiments were conducted in a CPU environment.

## 4.3 TRANSFER LEARNING

Leveraging the robust cross-domain behavior of QDOT, we propose a parameter-free transfer learning approach for point cloud classification. We evaluate it on ModelNet40 (Wu et al., 2015) and ShapeNetPart (Fan et al., 2017), focusing on the seven shared classes (3,072 samples from ModelNet and 15,402 from ShapeNet). This setting entails tens of millions of pairwise comparisons, making computationally intensive approaches infeasible; we therefore restrict our baselines to linear-time methods. To test rotational robustness, random vertical-axis rotations are applied, and experiments are conducted with point clouds subsampled to $n = 1024$ and $n = 2048$. The task treats one dataset as the source and the other as the target: each target cloud is classified via 1-Nearest Neighbor using distances to all source clouds computed by our metric. Results are reported below.

As shown in Table 2, IQDOT demonstrates more consistent performance compared to SGW. In the ModelNet-to-ShapeNet transfer task, IQDOT achieves superior overall results. Furthermore, its performance in the ShapeNet-to-ModelNet direction is exceptionally strong, improving the average accuracy by 35% over the SGW method. Notably, IQDOT completed this large-scale task in just a few hours, highlighting its remarkable efficiency.

Table 2: Point Cloud Transfer Learning Classification Accuracy (%)

| Methods (Mo→Sh) | airplane | car | chair | guitar | lamp | laptop | table | Avg. | Time(h) |
|---|---|---|---|---|---|---|---|---|---|
| SGW-1024 | _96.69_ | **99.10** | 60.67 | 0.00 | _71.29_ | 98.66 | _94.25_ | 79.77 | 13.11 |
| SGW-2048 | **96.98** | **99.10** | 60.24 | 0.00 | **71.62** | 98.66 | **94.47** | 79.83 | 21.84 |
| **IQDOT-1024** | 94.86 | 84.18 | _83.82_ | **96.56** | 51.13 | 96.45 | 84.95 | _83.89_ | **1.59** |
| **IQDOT-2048** | 95.35 | 87.86 | **85.01** | _96.44_ | 52.55 | 98.00 | 87.15 | **85.42** | _5.98_ |
| **Methods (Sh→Mo)** | airplane | car | chair | guitar | lamp | laptop | table | Avg. | Time(h) |
| SGW-1024 | 89.66 | 37.03 | 43.68 | 0.00 | 80.55 | 20.11 | _99.59_ | 59.66 | 13.11 |
| SGW-2048 | 91.18 | 37.03 | 44.08 | 0.00 | 83.33 | 20.11 | **99.79** | 60.31 | 21.84 |
| **IQDOT-1024** | **98.89** | _87.20_ | _91.10_ | **98.43** | 81.94 | _97.04_ | 96.74 | _93.97_ | **1.59** |
| **IQDOT-2048** | **98.89** | **91.91** | **93.22** | _98.03_ | _77.77_ | **97.63** | 97.56 | **95.05** | _5.98_ |

## 4.4 MOLECULE GENERATION

Molecular generation is a central challenge in drug discovery and molecular science. Recent advances show that diffusion models with equivariant neural architectures have become the dominant paradigm (Hoogeboom et al., 2022; Xu et al., 2023; Song et al., 2024; Feng et al., 2025). Since molecular properties are invariant to absolute position and orientation, prior works attempted to enforce stability through explicit structural alignment during training (Song et al., 2023; Hassan et al., 2024), but such procedures incur high computational cost. We instead integrate the IQDOT distance directly into the diffusion loss, providing an alignment-free learning signal that is inherently invariant and emphasizes geometric fidelity. This encourages physically stable and generalizable molecule generation without the overhead of explicit alignment.

We evaluate this approach on QM9 (Ramakrishnan et al., 2014) by adapting the classic model EDM (Hoogeboom et al., 2022) and a recently model UniGEM (Feng et al., 2025), with IQDOT-augmented loss functions. Both are trained for 3000 epochs, and the results are reported below.

Table 3: Molecule Generation Results on QM9 Dataset

| Methods | Atom sta(%) | Mol sta(%) | Valid(%) | V * U(%) |
|---|---|---|---|---|
| Data | 99.0 | 95.2 | 97.7 | 97.7 |
| EDM | $98.70_{\pm0.01}$ | $86.56_{\pm0.27}$ | $93.73_{\pm0.12}$ | $92.04_{\pm0.03}$ |
| →$\text{MSE}_{0.1}$ | $98.81_{\pm0.04}$ | $87.63_{\pm0.21}$ | $94.54_{\pm0.15}$ | $\mathbf{92.81}_{\pm\mathbf{0.20}}$ |
| →$\text{MSE}_{0.3}$ | $98.68_{\pm0.07}$ | $86.04_{\pm0.74}$ | $93.83_{\pm0.43}$ | $92.39_{\pm0.55}$ |
| →$\text{IQDOT}_{0.1}$ | $99.15_{\pm0.01}$ | $90.91_{\pm0.21}$ | $96.75_{\pm0.33}$ | $89.06_{\pm0.19}$ |
| →$\text{IQDOT}_{0.3}$ | $\mathbf{99.35}_{\pm\mathbf{0.06}}$ | $\mathbf{93.29}_{\pm\mathbf{0.81}}$ | $\mathbf{97.80}_{\pm\mathbf{0.22}}$ | $81.21_{\pm2.03}$ |
| UniGEM | $98.90_{\pm0.03}$ | $89.40_{\pm0.02}$ | $94.58_{\pm0.07}$ | $92.75_{\pm0.11}$ |
| →$\text{MSE}_{0.1}$ | $98.91_{\pm0.04}$ | $89.00_{\pm0.22}$ | $95.00_{\pm0.16}$ | $93.13_{\pm0.10}$ |
| →$\text{MSE}_{0.3}$ | $99.00_{\pm0.07}$ | $89.48_{\pm0.70}$ | $95.16_{\pm0.16}$ | $93.34_{\pm0.18}$ |
| →$\text{IQDOT}_{0.1}$ | $99.24_{\pm0.03}$ | $92.73_{\pm0.09}$ | $96.85_{\pm0.11}$ | $\mathbf{93.42}_{\pm\mathbf{0.23}}$ |
| →$\text{IQDOT}_{0.3}$ | $\mathbf{99.44}_{\pm\mathbf{0.01}}$ | $\mathbf{95.23}_{\pm\mathbf{0.18}}$ | $\mathbf{97.94}_{\pm\mathbf{0.06}}$ | $83.88_{\pm0.30}$ |

[1] The notation "$\text{IQDOT}_{0.1}$" indicates that the error function $\mathcal{L} \leftarrow 0.1\mathcal{L}_{\text{IQDOT}} + 0.9\mathcal{L}_{\text{MSE}}$; "$\text{MSE}_{0.1}$" indicates that $\mathcal{L} \leftarrow 0.9\mathcal{L}_{\text{MSE}}$.

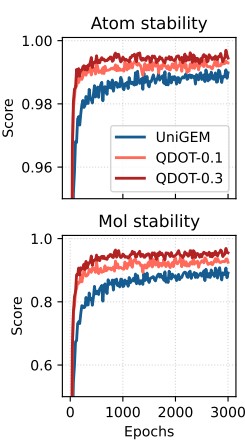

Figure 6: Stability trends across epochs.

As shown in Table 3, our IQDOT loss significantly enhances both EDM and UniGEM baselines. For UniGEM, a mere 10% QDOT weight ($\text{IQDOT}_{0.1}$) improves all four metrics, while a 30% weight ($\text{IQDOT}_{0.3}$) establishes a new state-of-the-art, boosting Atom and Molecule Stability to 99.44% and 95.23%, respectively. An ablation study confirms these gains stem from IQDOT's geometric guidance, not merely from re-weighting the original loss. Furthermore, as illustrated in Figure 6, IQDOT accelerates training, halving the convergence time for UniGEM to just 1000 epochs and thereby improving training efficiency and stability.

To assess the fine-tuning capability of the IQDOT loss, we conducted an additional experiment on GEOM-Drug (Axelrod & Gomez-Bombarelli, 2022). Starting from a UniGEM model pre-trained for 13 epochs (Feng et al., 2025), we introduced IQDOT into the loss and fine-tuned for 3 more epochs with a learning rate of $10^{-4}$. The results in Table 4 show that while continued training with the original MSE loss yields no further gains, incorporating IQDOT leads to substantial improvements in Atom Stability and Molecule Stability, with only a negligible drop in validity.

Table 4: Fine-tuning Results on GEOM-Drugs

| Methods | Atom sta(%) | Mol sta(%) | Valid(%) |
|---|---|---|---|
| UniGEM | 84.84 | 1.20 | 98.29 |
| →MSE | 84.39 | 1.13 | **99.01** |
| →$IQDOT_{0.1}$ | 87.94 | 6.04 | 98.46 |
| →$IQDOT_{0.2}$ | **91.87** | 17.14 | 97.86 |
| →$IQDOT_{0.3}$ | 91.73 | **22.05** | 94.62 |

## 5 CONCLUSION

This work introduces QDOT, a novel geometric metric. By constructing isometry-invariant anchors from distance quantiles and deriving a lossless distance representation through trilateration theory, QDOT establishes a rigorous metric on isometry classes of mm-spaces. Experimental results demonstrate its effectiveness across cross-space alignment and comparison, computational efficiency, transfer learning, and molecular generation tasks. Future directions include: (1) extending QDOT to hyperbolic and spherical spaces; (2) developing more general approaches for constructing isometry-invariant anchors and representations; and (3) applying QDOT-based alignment techniques and QDOT as a loss function in complex models.

## ETHICS STATEMENT

All datasets utilized in this study are publicly available, and their use raises no ethical concerns.

## REPRODUCIBILITY STATEMENT

We are committed to ensuring the reproducibility of our research. The source code for all experiments is provided in the supplementary materials, which can be used to reproduce the results presented in this paper. For the molecule generation experiment, we have included the evaluate checkpoint of QDOT-0.3. This checkpoint can also be reproduced using the provided training code. Furthermore, we have explicitly stated all key assumptions for the theorems presented. The sources and preprocessing scripts for all datasets used in this work are also provided.

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

## A  ADDITIONAL BACKGROUND

Here we introduce additional background concepts that are related to our work.

**One-Dimensional Wasserstein Distance.**  A key special case of the Wasserstein distance defined in equation 1 occurs in one dimension, where it admits a convenient closed-form solution. For two one-dimensional distributions $\mu_X$ and $\mu_Y$, let $F_X$ and $F_Y$ be their respective cumulative distribution functions , and let $F_X^{-1}$ and $F_Y^{-1}$ be the corresponding quantile functions. The $p$-Wasserstein distance is then given by:

$$\mathcal{W}_p(\mu_X, \mu_Y) = \left( \int_0^1 |F_X^{-1}(u) - F_Y^{-1}(u)|^p \, du \right)^{\frac{1}{p}}. \tag{9}$$

For discrete distributions, the empirical quantile functions are readily obtained by sorting, making the computation of the 1D Wasserstein distance explicit and efficient.

**Gromov-Wasserstein Distance(Mémoli, 2011).**  While the original formulation of the Gromov-Wasserstein distance in equation 2 is theoretically elegant, its computation is NP-hard. Consequently, an alternative and more commonly used formulation was introduced. Given two mm-spaces $\mathcal{X} = (X, d_X, \mu_X)$ and $\mathcal{Y} = (Y, d_Y, \mu_Y)$, for $p \geq 1$, the $p$-Gromov-Wasserstein distance is defined as:

$$\mathcal{GW}_p(\mathcal{X}, \mathcal{Y}) := \left( \inf_{\pi \in \Pi(\mu_X, \mu_Y)} \int_{X \times Y} \int_{X \times Y} |d_X(x, x') - d_Y(y, y')|^p \, d\pi(x, y) \, d\pi(x', y') \right)^{\frac{1}{p}}. \tag{10}$$

Mémoli (2011) proved that the formulation in equation 10 is bi-Hölder equivalent to the one in equation 2. Moreover, equation 2 constitutes a metric in the sense of Definition 1.

**Computational Costs.**  We now briefly review the computational complexity of these OT-based methods for discrete distributions supported on $n$ points. Computing the standard Wasserstein distance in equation 1 using a classic EMD solver has a worst-case complexity of $\mathcal{O}(n^3)$ and an average-case complexity of $\mathcal{O}(n^2)$ (Bonneel et al., 2011). Alternatively, solving the entropy-regularized OT problem with the Sinkhorn algorithm (Cuturi, 2013) has a complexity of up to $\mathcal{O}(n^2 \log n)$. While some targeted algorithms may achieve faster runtimes (Altschuler et al., 2019; Li et al., 2023a), this often comes at the cost of sacrificing metric properties. Computing the Gromov-Wasserstein distance in equation 10 involves solving a non-convex quadratic program, which has a complexity of at least $\mathcal{O}(n^3 \log n)$ (Xu et al., 2019). Faster algorithms, such as those employing entropic regularization (Peyré et al., 2016; Scetbon et al., 2022; Li et al., 2023b), typically represent a trade-off between speed and accuracy. As for the one-dimensional Wasserstein distance in equation 9 , the computation is significantly faster, requiring only $\mathcal{O}(n \log n)$ time via sorting. This efficiency has motivated many methods that project higher-dimensional distributions onto one-dimensional lines for comparison (Bonneel et al., 2015; Deshpande et al., 2019; Le et al., 2019; Li et al., 2024; 2025). However, for the Gromov-Wasserstein distance, the development of sliced variants is less mature, with Sliced Gromov-Wasserstein (SGW) (Titouan et al., 2019) being the most prominent example that has achieved empirical success.

## B  PROOFS

### B.1  PROOF OF THEOREM 1

The proof for the symmetry and triangle inequality properties is straightforward, as they are directly inherited from the Wasserstein distance $\mathcal{W}_p$. We focus on proving the **Identity of Indiscernibles**, which is the most involved part.

To do so, we first recall an equivalent characterization of isometry from (Sturm, 2023, Lemma 1.10):

   (i)  $\mathcal{X}, \mathcal{Y}$ are isometric.

(ii) There exists a coupling $\pi \in \Pi(\mu_X, \mu_Y)$ such that $d_X(\boldsymbol{x}_0, \boldsymbol{x}_1) = d_Y(\boldsymbol{y}_0, \boldsymbol{y}_1)$ holds for $\pi \otimes \pi$-almost every pair of points $((\boldsymbol{x}_0, \boldsymbol{y}_0), (\boldsymbol{x}_1, \boldsymbol{y}_1))$.

The "if" part is straightforward: if $\mathcal{X}$ and $\mathcal{Y}$ are isometric, their QDMD representations will be identically distributed by construction, making their QDOT distance zero. We now prove the "only if" part: if $\mathcal{QD}_p(\mathcal{X}, \mathcal{Y}) = 0$, then $\mathcal{X}$ and $\mathcal{Y}$ are isometric by satisfying condition (ii).

**Proof.** By definition, the condition $\mathcal{QD}_p(\mathcal{X}, \mathcal{Y}) = 0$ implies that there exists an optimal coupling $\pi$ between the pushforward measures on the representation space, such that $\phi^X(\boldsymbol{x}) = \phi^Y(\boldsymbol{y})$ holds for all $(\boldsymbol{x}, \boldsymbol{y})$ in the support of the corresponding coupling on $X \times Y$.

For a given quantile level vector $\boldsymbol{q} \in (0, 1)^k$, let us denote the corresponding QDMs as $\boldsymbol{m}_i^X = \boldsymbol{m}^X(q_i)$ and $\boldsymbol{m}_i^Y = \boldsymbol{m}^Y(q_i)$ for $i = 1, 2, \ldots, k$. The equality of the representations implies:

$$\begin{cases} \|\boldsymbol{x} - \boldsymbol{m}_i^X\|_2 = \|\boldsymbol{y} - \boldsymbol{m}_i^Y\|_2, \\ \|\boldsymbol{x}\|_2 = \|\boldsymbol{y}\|_2, \end{cases} \quad \forall(\boldsymbol{x}, \boldsymbol{y}) \in \mathrm{supp}(\pi).$$

From $\|\boldsymbol{x}\|_2 = \|\boldsymbol{y}\|_2$ and the fact that isometric spaces have identical norm distributions (and thus identical quantile functions), it directly follows that the weights are equal for coupled points:

$$w^X(\boldsymbol{x}, q_i) = w^Y(\boldsymbol{y}, q_i), \quad \forall(\boldsymbol{x}, \boldsymbol{y}) \in \mathrm{supp}(\pi), i = 1, 2, \ldots k.$$

For any $(\boldsymbol{x}, \boldsymbol{y}) \in \mathrm{supp}(\pi)$, expanding the squared distance equality yields:

$$\|\boldsymbol{x} - \boldsymbol{m}_i^X\|_2^2 = \|\boldsymbol{y} - \boldsymbol{m}_i^Y\|_2^2$$
$$\Rightarrow \boldsymbol{x}^\top \boldsymbol{x} + \boldsymbol{m}_i^{X\top} \boldsymbol{m}_i^X - 2\boldsymbol{x}^\top \boldsymbol{m}_i^X = \boldsymbol{y}^\top \boldsymbol{y} + \boldsymbol{m}_i^{Y\top} \boldsymbol{m}_i^Y - 2\boldsymbol{y}^\top \boldsymbol{m}_i^Y$$
$$\Rightarrow \boldsymbol{m}_i^{X\top} \boldsymbol{m}_i^X - 2\boldsymbol{x}^\top \boldsymbol{m}_i^X = \boldsymbol{m}_i^{Y\top} \boldsymbol{m}_i^Y - 2\boldsymbol{y}^\top \boldsymbol{m}_i^Y,$$

where the last step uses $\|\boldsymbol{x}\|_2^2 = \|\boldsymbol{y}\|_2^2$. Integrating this equality against the weight function $w(\cdot, q_j)$ over the respective spaces gives:

$$\int_X w^X(q_j, \boldsymbol{x})(\boldsymbol{m}_i^{X\top} \boldsymbol{m}_i^X - 2\boldsymbol{x}^\top \boldsymbol{m}_i^X) d\mu_X(\boldsymbol{x})$$
$$= \int_Y w^Y(q_j, \boldsymbol{y})(\boldsymbol{m}_i^{Y\top} \boldsymbol{m}_i^Y - 2\boldsymbol{y}^\top \boldsymbol{m}_i^Y) d\mu_Y(\boldsymbol{y})$$
$$= \mathbb{E}_X(w^X(q_j, X))\boldsymbol{m}_i^{X\top} \boldsymbol{m}_i^X - 2\mathbb{E}_X(w^X(q_j, X)X)$$
$$= \mathbb{E}_Y(w^X(q_j, Y))\boldsymbol{m}_i^{Y\top} \boldsymbol{m}_i^Y - 2\mathbb{E}_Y(w^Y(q_j, Y)Y).$$

By the definition of QDM in Eq. equation 5, this simplifies to:

$$\boldsymbol{m}_i^{X\top} \boldsymbol{m}_i^X - 2\boldsymbol{m}_i^{X\top} \boldsymbol{m}_j^X = \boldsymbol{m}_i^{Y\top} \boldsymbol{m}_i^Y - 2\boldsymbol{m}_i^{Y\top} \boldsymbol{m}_j^Y.$$

By setting $i = j$, we find $\|\boldsymbol{m}_i^X\|_2^2 = \|\boldsymbol{m}_i^Y\|_2^2$, which implies $\|\boldsymbol{m}_i^X\|_2 = \|\boldsymbol{m}_i^Y\|_2$. Substituting this back, we obtain the key result that the Gram matrices of the QDM sets are identical: $\boldsymbol{m}_i^{X\top} \boldsymbol{m}_j^X = \boldsymbol{m}_i^{Y\top} \boldsymbol{m}_j^Y$.

Now, for any $(\boldsymbol{x}, \boldsymbol{y}) \in \mathrm{supp}(\pi)$, the relation $\|\boldsymbol{x} - \boldsymbol{m}_i^X\|_2^2 = \|\boldsymbol{y} - \boldsymbol{m}_i^Y\|_2^2$ simplifies further:

$$\boldsymbol{x}^\top \boldsymbol{x} + \boldsymbol{m}_i^{X\top} \boldsymbol{m}_i^X - 2\boldsymbol{x}^\top \boldsymbol{m}_i^X = \boldsymbol{y}^\top \boldsymbol{y} + \boldsymbol{m}_i^{Y\top} \boldsymbol{m}_i^Y - 2\boldsymbol{y}^\top \boldsymbol{m}_i^Y$$
$$\Rightarrow \boldsymbol{x}^\top \boldsymbol{m}_i^X = \boldsymbol{y}^\top \boldsymbol{m}_i^Y.$$

Due to the dimensionality condition $\dim(\{\boldsymbol{m}^X(q_i)\}_{1 \le i \le k}) = \dim(X)$, any point $\boldsymbol{x} \in \mathrm{supp}(\mu_X)$ can be written as a linear combination $\boldsymbol{x} = \sum_{i=1}^k \alpha_i \boldsymbol{m}_i^X$ for some coefficients $\alpha_1, \ldots, \alpha_k$. We can show that its coupled counterpart $\boldsymbol{y}$ must be the same linear combination of the corresponding

QDMs in $Y$:

$$\|\boldsymbol{y} - \sum_{i=1}^{k} \alpha_i \boldsymbol{m}_i^Y\|_2^2$$

$$=\|\boldsymbol{y}\|_2^2 + \|\sum_{i=1}^{k} \alpha_i \boldsymbol{m}_i^Y\|_2^2 - 2\sum_{i=1}^{k} \alpha_i \boldsymbol{y}^\top \boldsymbol{m}_i^Y$$

$$=\|\boldsymbol{y}\|_2^2 + \sum_{i=1}^{k}\sum_{j=1}^{k} \alpha_i\alpha_j \boldsymbol{m}_i^{Y\top} \boldsymbol{m}_j^Y - 2\sum_{i=1}^{k} \alpha_i \boldsymbol{x}^\top \boldsymbol{m}_i^X$$

$$=\|\boldsymbol{y}\|_2^2 + \sum_{i=1}^{k}\sum_{j=1}^{k} \alpha_i\alpha_j \boldsymbol{m}_i^{X\top} \boldsymbol{m}_j^X - 2\boldsymbol{x}^\top \sum_{i=1}^{k} \alpha_i \boldsymbol{m}_i^X$$

$$=\|\boldsymbol{y}\|_2^2 + \boldsymbol{x}^\top \boldsymbol{x} - 2\boldsymbol{x}^\top \boldsymbol{x} = 0,$$

where the last equality uses $\|\boldsymbol{x}\|_2^2 = \|\boldsymbol{y}\|_2^2$. This implies $\boldsymbol{y} = \sum_{i=1}^{k} \alpha_i \boldsymbol{m}_i^Y$.

Finally, for any two pairs $(\boldsymbol{x}_1, \boldsymbol{y}_1), (\boldsymbol{x}_2, \boldsymbol{y}_2) \in \text{supp}(\pi)$, let $\boldsymbol{x}_1 = \sum_{i=1}^{k} \alpha_i \boldsymbol{m}_i^X$ and $\boldsymbol{x}_2 = \sum_{j=1}^{k} \beta_j \boldsymbol{m}_j^X$. It follows that their distances are preserved:

$$\|\boldsymbol{x}_1 - \boldsymbol{x}_2\|_2^2 - \|\boldsymbol{y}_1 - \boldsymbol{y}_2\|_2^2$$

$$=(\|\boldsymbol{x}_1\|_2^2 - \|\boldsymbol{y}_1\|_2^2) + (\|\boldsymbol{x}_2\|_2^2 - \|\boldsymbol{y}_2\|_2^2) - (2\boldsymbol{x}_1^\top \boldsymbol{x}_2 - 2\boldsymbol{y}_1^\top \boldsymbol{y}_2)$$

$$=-2\sum_{i=1}^{k}\sum_{j=1}^{k} \alpha_i\beta_j (\boldsymbol{m}_i^{X\top} \boldsymbol{m}_j^X - \boldsymbol{m}_i^{Y\top} \boldsymbol{m}_j^Y) = 0.$$

Thus, $d_X(\boldsymbol{x}_1, \boldsymbol{x}_2) = d_Y(\boldsymbol{y}_1, \boldsymbol{y}_2)$ holds for $\pi \otimes \pi$-a.e. pairs, fulfilling condition (ii) and completing the proof. $\qquad\square$

### B.2 PROOF OF THEOREM 2

This section provides the proof for Theorem 2. We begin by establishing the necessary notation and preliminary lemmas. Let $r_q = F_{\|X\|_2}^{-1}(q)$ be the true $q$-th quantile of the norm distribution, and let $\hat{r}_q = \hat{F}_{\|\mathbb{X}_n\|_2}^{-1}(q)$ be its empirical counterpart estimated from the sample set $\mathbb{X}_n = \{x_1, \ldots, x_n\}$. For brevity, we denote the weight functions as $w_i^X(x) = w^X(x, q_i)$ and its empirical version as $\hat{w}_i^{\mathbb{X}_n}(x) = e^{-\sigma(\|\boldsymbol{x}\|_2 - \hat{r}_q)^2}$.

Our proof relies on the well-established convergence rates for the Wasserstein distance, summarized in the following lemma.

**Lemma 1 (Convergence Rate of Wasserstein Distance (Fournier & Guillin, 2015))** *For a distribution $\mu$ defined on a vector measure space $(X, \mathcal{B}(X))$ and its empirical version $\mu_n$, if the $q$-th moment $\mathcal{M}_q(\mu)$ is finite, i.e., $\mathcal{M}_q(\mu) = \int_X \|\boldsymbol{x}\|^q d\mu(x) < \infty$, then the following holds:*

$$\mathbb{E}\left(\mathcal{W}_p\left(\mu_n, \mu\right)\right) \leq C M_q^{p/q}(\mu)$$

$$\times \begin{cases} n^{-1/2} + n^{-(q-p)/q} & \text{if } p > d/2 \text{ and } q \neq 2p \\ n^{-1/2}\log(1+n) + n^{-(q-p)/q} & \text{if } p = d/2 \text{ and } q \neq 2p \\ n^{-p/d} + n^{-(q-p)/q} & \text{if } p \in (0, d/2) \text{ and } q \neq d/(d-p) \end{cases}$$

As discussed in (Fournier & Guillin, 2015), for a sufficiently large moment order $q$, the term $n^{-(q-p)/q}$ becomes negligible compared to the leading term. For the common case of $p = 2$, this holds for $q \geq 4$. We will assume this condition holds in our subsequent analysis.

Next, we state the standard convergence rate for empirical quantiles.

**Lemma 2 (Convergence Rate of Quantiles)** *Let $F_X$ be the CDF of a distribution $\mu_X$, and assume it is continuous on $r_q$ and has a strictly positive density. Then, the mean squared error of the empirical quantile converges as follows:*

$$\mathbb{E}|r_q - \hat{r}_q|^2 = O(n^{-1}).$$

This is a classical result in asymptotic theory, which can be found in (Serfling, 1980, Chap 2.3).

Furthermore, our proof requires the following bounds. The first lemma ensures that the denominator in the QDM definition is well-behaved.

**Lemma 3** *If $\mathbb{E}\|X\|_2 < \infty$, then for any given quantile level $q_i$, the expected weight is bounded away from zero, i.e., $\frac{1}{\mathbb{E}_{\mu_X}(w_i^X(X))} < \infty$.*

**Proof.** Since $\mathbb{E}_{\mu_X}(\|X\|_2) < \infty$, for any given $p_0 < 1$, there exists an $M_0$ such that $\Pr(\|X\|_2 \leq M_0) \geq p_0$. Therefore, we have:

$$\mathbb{E}_{\mu_X}(w_i^X(X)) \geq \int_{\|\boldsymbol{x}\|_2 < M_0} e^{-\sigma(x-r_{q_i})^2} d\mu_X(x) \geq \int_{\|\boldsymbol{x}\|_2 < M_0} e^{-\sigma \max\{x, r_{q_i}\}^2} d\mu_X(x)$$

$$\geq \int_{\|\boldsymbol{x}\|_2 < M_0} e^{-\sigma \max\{M_0, r_{q_i}\}^2} d\mu_X(x) \geq p_0 e^{-\sigma \max\{M_0, r_{q_i}\}^2}.$$

This implies that the expectation is strictly positive, and thus $\frac{1}{\mathbb{E}_{\mu_X}(w_i^X(X))} < \infty$. $\qquad\square$

**Lemma 4** *For any quantile level $q_i$, let the normalized empirical weights be $\hat{p}_j = \frac{\hat{w}_i^{\mathbb{X}_n}(\boldsymbol{x}_j)}{\sum_{j'=1}^n \hat{w}_i^{\mathbb{X}_n}(\boldsymbol{x}_j')}$. Then for any power $1 \leq \alpha < \infty$, we have the following bound:*

$$\sum_{j=1}^n \mathbb{E}(\hat{p}_j^\alpha)^{1/\alpha} < \infty.$$

**Proof.** Let $M_\alpha(\hat{w}_i) = \mathbb{E}(\hat{w}_i^{\mathbb{X}_n}(\boldsymbol{x}_j)^\alpha)$. Since $\hat{w}_i^{\mathbb{X}_n}(\boldsymbol{x}_j) < 1$, it is clear that $M_\alpha(\hat{w}_i) < 1$. Define the sample mean of the weights as $\bar{w}(r) = \frac{1}{n} \sum_{j=1}^n e^{-\sigma(\|\boldsymbol{x}_j\|_2 - r)^2}$. We then have $\mathbb{E}(\bar{w}(\hat{r}_{q_i})) = M_1(\hat{w}_i)$. Now, consider the following probability:

$$\mathbb{P}\big(\bar{w}_i(\hat{r}_{q_i}) < \tfrac{1}{2} M_1(\hat{w}_i)\big) \leq \underbrace{\mathbb{P}\big(\bar{w}_i(r_{q_i}) < \tfrac{3}{4} M_1(\hat{w}_i)\big)}_{\mathbb{P}_1} + \underbrace{\mathbb{P}\big(|\bar{w}_i(r_{q_i}) - \bar{w}_i(r_{q_i})| > \tfrac{1}{4} M_1(\hat{w}_i)\big)}_{\mathbb{P}_2}.$$

For the first term, $\mathbb{P}_1$, we note that $\mathbb{E}(\bar{w}_i(r_{q_i})) = M_1(\hat{w}_i)$ and the terms in the sum are i.i.d. Applying Hoeffding's inequality yields:

$$\mathbb{P}_1 = \mathbb{P}\big(\mathbb{E}(\bar{w}_i(r_{q_i})) - \bar{w}_i(r_{q_i}) > \tfrac{n}{4} M_1(\hat{w}_i)\big)$$

$$= \mathbb{P}\Big(\sum_{j=1}^n \mathbb{E}(w_i^X(\boldsymbol{x}_j)) - \sum_{j=1}^n w_i^X(\boldsymbol{x}_j) > \tfrac{n}{4} M_1(\hat{w}_i)\Big)$$

$$\leq \exp(-\tfrac{n}{8} M_1^2(\hat{w}_i)).$$

For the second term, $\mathbb{P}_2$, we first bound the difference by applying the Mean Value Theorem:

$$\big|\bar{w}_i(r_{q_i}) - \bar{w}_i(r_{q_i})\big| \leq \tfrac{1}{n} \sum_{j=1}^n \big|e^{-\sigma(\|\boldsymbol{x}_j\|_2 - r_{q_i})} - e^{-\sigma(\|\boldsymbol{x}_j\|_2 - \hat{r}_{q_i})}\big|$$

$$\leq \tfrac{1}{n} \sum_{j=1}^n \big|2\sigma(\|\boldsymbol{x}_j\|_2 - r_{j0})e^{-\sigma(\|\boldsymbol{x}_j\|_2 - -r_{j0}0)^2}\big| \big|r_{q_i} - \hat{r}_{q_i}\big| \qquad (11)$$

$$\leq \sqrt{2\sigma e}^{-1/2} \big|r_{q_i} - \hat{r}_{q_i}\big| \quad (\text{since } \sup_{t \geq 0} 2\sigma t e^{-\sigma t^2} \leq \sqrt{2\sigma} e^{-1/2}).$$

Consequently, we have:

$$\mathbb{P}_2 \leq \mathbb{P}\Big(|r_{q_i} - \hat{r}_{q_i}| > \sqrt{\tfrac{e}{32\sigma}} M_1(\hat{w}_i)\Big) \leq 2 \exp(-2n\delta),$$

where $\delta = \min\{F_X(r_{p_i} + \sqrt{\tfrac{e}{32\sigma}} M_1(\hat{w}_i)) - p_i, p_i - F_X(r_{p_i} - \sqrt{\tfrac{e}{32\sigma}} M_1(\hat{w}_i))\} > 0$. The final inequality follows from a standard result on the concentration of empirical quantiles, see, e.g.,

(Serfling, 1980, Theorem 2.3.2). By the symmetry of the i.i.d. samples, $\mathbb{E}(\hat{p}_j^\alpha) = \mathbb{E}(\hat{p}_l^\alpha)$ for all $j, l \in \{1, \ldots, n\}$. Thus we can write:

$$
\begin{aligned}
\sum_{j=1}^n \mathbb{E}(\hat{p}_j^\alpha)^{1/\alpha} =& n\mathbb{E}(\hat{p}_1^\alpha)^{1/\alpha} \\
=& \mathbb{E}\big(\frac{\hat{w}_i^{\mathbb{X}_n}(\boldsymbol{x}_1)^\alpha}{\bar{w}_i(\hat{r}_{q_i})^\alpha}\big)^{1/\alpha} \\
\leq& \mathbb{E}_{\{\bar{w}_i(\hat{r}_{q_i}) \geq \frac{1}{2} M_1(\hat{w}_i)\}}\big(\frac{\hat{w}_i^{\mathbb{X}_n}(\boldsymbol{x}_1)^\alpha}{\bar{w}_i(\hat{r}_{q_i})^\alpha}\big)^{1/\alpha} + n\mathbb{P}(\bar{w}_i(\hat{r}_{q_i}) < \tfrac{1}{2} M_1(\hat{w}_i)) \\
\leq& \frac{2\mathbb{E}(\hat{w}_i^{\mathbb{X}_n}(\boldsymbol{x}_1)^\alpha)^{1/\alpha}}{M_1(\hat{w}_i))} + n\exp(-\tfrac{n}{8} M_1^2(\hat{w}_i)) + 2n\exp(-2n\delta) \\
\leq& \frac{2M_\alpha(\hat{w}_i)^{1/\alpha}}{M_1(\hat{w}_i))} + \frac{8e^{-1}}{M_1^2(\hat{w}_i)} + \frac{e^{-1}}{\delta} < \infty.
\end{aligned}
$$

The final line is finite, leveraging the result from Lemma 3 which ensures $M_1(\cdot)$ is bounded away from zero. This completes the proof. □

**Lemma 5** *If the fourth moment of the norm is finite, i.e., $\mathbb{E}\|X\|_2^4 < \infty$, then the expected squared norm of the empirical QDM is also finite: $\mathbb{E}\|\hat{m}_i^{\mathbb{X}_n}\|_2^2 < \infty$.*

**Proof.** The proof follows from applying the Cauchy-Schwarz or Hölder's inequality and leveraging the result from Lemma 4.

$$
\begin{aligned}
\mathbb{E}\|\hat{m}_i^{\mathbb{X}_n}\|_2^2 =& \sum_{j=1}^n \sum_{l=1}^n \mathbb{E}(\hat{p}_j \hat{p}_l \boldsymbol{x}_j^\top \boldsymbol{x}_l) \\
\leq& \sum_{j=1}^n \sum_{l=1}^n \mathbb{E}(\hat{p}_j^4)^{1/4} \mathbb{E}(\hat{p}_l^4)^{1/4} \mathbb{E}((\boldsymbol{x}_j^\top \boldsymbol{x}_l)^2)^{1/2} \\
\leq& \sum_{j=1}^n \mathbb{E}(\hat{p}_j^4)^{1/2} \mathbb{E}(\|\boldsymbol{x}_j\|_2^4)^{1/2} + \sum_{j \neq l} \mathbb{E}(\hat{p}_j^4)^{1/4} \mathbb{E}(\hat{p}_l^4)^{1/4} \mathbb{E}(\|\boldsymbol{x}_j\|_2^2 \|\boldsymbol{x}_l\|_2^2)^{1/2} \\
=& \mathbb{E}(\|X\|_2^4)^{1/2} \sum_{j=1}^n \mathbb{E}(\hat{p}_j^4)^{1/2} + \mathbb{E}(\|X\|_2^2)\Big(\sum_{j=1}^n \mathbb{E}(\hat{p}_j^4)^{1/2}\Big)^2 < \infty.
\end{aligned}
$$

□

**Lemma 6** *If $\mathbb{E}\|X\|_2^4 < \infty$, then the expectation of the product of the empirical QDM norm, a sample norm, and the quantile error converges at the rate of $O(n^{-1/2})$: $\mathbb{E}(\|\hat{m}_i^{\mathbb{X}_n}\|_2 \|\boldsymbol{x}_1\| |r_{q_i} - \hat{r}_{q_i}|) = O(n^{-1/2})$.*

**Proof.** This proof also relies on the Cauchy-Schwarz inequality and the results from Lemma 2 and Lemma 4.

$$
\begin{aligned}
\mathbb{E}(\|\hat{m}_i^{\mathbb{X}_n}\|_2 \|\boldsymbol{x}_1\|_2 |r_{q_i} - \hat{r}_{q_i}|) \leq& \mathbb{E}(\hat{q}_1 \|\boldsymbol{x}_1\|_2^2 |r_{q_i} - \hat{r}_{q_i}|) + \sum_{j=2}^n \mathbb{E}(\hat{q}_j \|\boldsymbol{x}_1\|_2 \|\boldsymbol{x}_j\|_2 |r_{q_i} - \hat{r}_{q_i}|) \\
\leq& \mathbb{E}(\|X\|_2^4)^{1/2} \mathbb{E}(|r_{q_i} - \hat{r}_{q_i}|^2)^{1/2} \\
&+ \sum_{j=2}^n \mathbb{E}(\hat{q}_j^4)^{1/4} \mathbb{E}(\|\boldsymbol{x}_1\|_2^4 \|\boldsymbol{x}_j\|_2^4)^{1/4} \mathbb{E}(|r_{q_i} - \hat{r}_{q_i}|^2)^{1/2} \\
=& \mathbb{E}(|r_{q_i} - \hat{r}_{q_i}|^2)^{1/2} \mathbb{E}(\|X\|_2^4)^{1/2}\Big(1 + \sum_{j=2}^n \mathbb{E}(\hat{q}_j^4)^{1/4}\Big).
\end{aligned}
$$

From Lemma 2, we know that $\mathbb{E}(|r_{q_i} - \hat{r}_{q_i}|^2)^{1/2} = O(n^{-1/2})$. Since $\mathbb{E}(\|X\|_2^4) < \infty$ is assumed, and Lemma 4 ensures that $\sum_{j=2}^n \mathbb{E}(\hat{q}_j^4)^{1/4} < \infty$, the result follows. □

**Proof of Theorem 2.** First, we use a property derived from the triangle inequality:

$$|d(\boldsymbol{x}, \boldsymbol{x}') - d(\boldsymbol{y}, \boldsymbol{y}')| \leq |d(\boldsymbol{x}, \boldsymbol{x}') - d(\boldsymbol{x}, \boldsymbol{y}')| + |d(\boldsymbol{x}, \boldsymbol{y}') - d(\boldsymbol{y}, \boldsymbol{y}')|$$

$$\leq d(\boldsymbol{x}, \boldsymbol{y}) + d(\boldsymbol{x}', \boldsymbol{y}').$$

This allows us to bound the QDOT distance.

$$
\begin{aligned}
\mathcal{QD}_p(\mathcal{X}, \mathcal{Y})^p &= \inf_{\pi \in \Pi(\mu_X, \mu_Y)} \int \left\| \boldsymbol{\phi}^X(\boldsymbol{x}, \boldsymbol{q}) - \boldsymbol{\phi}^Y(\boldsymbol{y}, \boldsymbol{q}) \right\|_p^p \, d\pi(x, y) \\
&\leq \inf_{\pi \in \Pi(\mu_X, \mu_Y)} \int \sum_{i=0}^k \left| \phi_i^X - \phi_i^Y \right|^p \, d\pi(x, y) \\
&\leq \inf_{\pi \in \Pi(\mu_X, \mu_Y)} \int \sum_{i=0}^k \left| d(\boldsymbol{x}, \boldsymbol{y}) + d(\boldsymbol{m}_i^X, \boldsymbol{m}_i^Y) \right|^p \, d\pi(x, y) \\
&\leq \inf_{\pi \in \Pi(\mu_X, \mu_Y)} \int (k+1) 2^{p-1} d(\boldsymbol{x}, \boldsymbol{y})^p + 2^{p-1} \sum_{i=0}^k d(\boldsymbol{m}_i^X, \boldsymbol{m}_i^Y)^p \, d\pi(x, y) \quad (12) \\
&= (k+1) 2^{p-1} \inf_{\pi \in \Pi(\mu_X, \mu_Y)} \int d(\boldsymbol{x}, \boldsymbol{y})^p \, d\pi(x, y) + 2^{p-1} \sum_{i=0}^k d(\boldsymbol{m}_i^X, \boldsymbol{m}_i^Y)^p \\
&= C_0 \underbrace{\mathcal{W}_p(\mu_X, \mu_Y)^p}_{\text{Wasserstein Discrepancy}} + C_1 \underbrace{\sum_{i=1}^k d(\boldsymbol{m}_i^X, \boldsymbol{m}_i^Y)^p}_{\text{QDM Discrepancy}}
\end{aligned}
$$

where $\boldsymbol{m}_0^X = \boldsymbol{0}, \boldsymbol{m}_0^Y = \boldsymbol{0}$, and the constants $C_0 = (k+1) 2^{p-1}$ and $C_1 = 2^{p-1}$ are finite. From (Fournier & Guillin, 2015, Theorem 1), we have the bound for the Wasserstein Discrepancy between the true measure and its empirical version:

$$\mathbb{E}(\mathcal{W}_p(\mu_X, \mu_n)^p) \leq C n^{-p/d}.$$

Next, we analyze the QDM Discrepancy term, $d(\boldsymbol{m}_i^X, \hat{\boldsymbol{m}}_i^{\mathbb{X}_n})$.

$$
\begin{aligned}
d(\boldsymbol{m}_i^X, \hat{\boldsymbol{m}}_i^{\mathbb{X}_n}) &= \left\| \frac{\frac{1}{n} \sum_{j=1}^n \hat{w}_i^{\mathbb{X}_n}(\boldsymbol{x}_j) \boldsymbol{x}_j}{\frac{1}{n} \sum_{j=1}^n \hat{w}_i^{\mathbb{X}_n}(\boldsymbol{x}_j)} - \frac{\mathbb{E}_{\mu_X}(w_i^X(X) X)}{\mathbb{E}_{\mu_X}(w_i^X(X))} \right\|_2 \\
&= \left\| \frac{\frac{1}{n} \sum_{j=1}^n \hat{w}_i^{\mathbb{X}_n}(\boldsymbol{x}_j) \boldsymbol{x}_j \mathbb{E}_{\mu_X}(w_i^X(X)) - \mathbb{E}_{\mu_X}(w_i^X(X) X) \frac{1}{n} \sum_{j=1}^n \hat{w}_i^{\mathbb{X}_n}(\boldsymbol{x}_j)}{\frac{1}{n} \sum_{j=1}^n \hat{w}_i^{\mathbb{X}_n}(\boldsymbol{x}_j) \mathbb{E}_{\mu_X}(w_i^X(X))} \right\|_2 \\
&\leq C_2 \underbrace{\left| \mathbb{E}_{\mu_X}(w_i^X(X)) - \frac{1}{n} \sum_{j=1}^n w_i^X(x_j) \right|}_{\text{Term (I)}} \\
&\quad + C_2 \underbrace{\left\| \hat{m}_i^{\mathbb{X}_n} \right\|_2 \left\| \frac{1}{n} \sum_{j=1}^n \hat{w}_i^{\mathbb{X}_n}(x_j) x_j - \mathbb{E}_{\mu_X}(w_i^X(X) X) \right\|_2}_{\text{Term (II)}}.
\end{aligned}
$$

$$(13)$$

Here, $C_2 = \frac{1}{\mathbb{E}_{\mu_X}(w_i^X(X))}$. By Lemma 3, under the condition that $q \geq 4$, $C_2$ is a finite constant. We now focus on the two main terms. Term (I) can be decomposed as:

$$
\text{Term (I)} \leq \underbrace{\left| \mathbb{E}_{\mu_X}(w_i^X(X)) - \frac{1}{n} \sum_{j=1}^n w_i^X(\boldsymbol{x}_j) \right|}_{\text{Term (I.1)}} + \underbrace{\left| \frac{1}{n} \sum_{j=1}^n w_i^X(\boldsymbol{x}_j) - \frac{1}{n} \sum_{j=1}^n \hat{w}_i^{\mathbb{X}_n}(\boldsymbol{x}_j) \right|}_{\text{Term (I.2)}}. \quad (14)
$$

For Term (I.1), standard results for the mean of i.i.d. variables give:

$$
\mathbb{E}(\text{Term (I.1)}) \leq \sqrt{\mathbb{E}\left( \mathbb{E}_{\mu_X}(w_i^X(X)) - \frac{1}{n} \sum_{j=1}^n w_i^X(\boldsymbol{x}_j) \right)^2} = \sqrt{\frac{1}{n} \text{Var}(w_i^X(\boldsymbol{x}_1))} = O(n^{-1/2}).
$$

Using the bound from equation 11, we have:

$$\text{Term (I.2)} \leq \sqrt{2\sigma}e^{-1/2}\left|r_{q_i} - \hat{r}_{q_i}\right|.$$

Since $\mathbb{E}\left|r_{q_i} - \hat{r}_{q_i}\right| = O(n^{-1/2})$, it follows that $\mathbb{E}(\text{Term (I)}) = O(n^{-1/2})$. For Term (II), we use a similar decomposition:

$$\text{Term (II)} \leq \underbrace{\left\|\hat{m}_i^{\mathbb{X}_n}\right\|_2 \left\|\frac{1}{n}\sum_{j=1}^{n}\hat{w}_i^{\mathbb{X}_n}(x_j)\,x_j - \frac{1}{n}\sum_{j=1}^{n}w_i^X(x_j)\,x_j\right\|_2}_{\text{Term (II.1)}}$$

$$+ \underbrace{\left\|\hat{m}_i^{\mathbb{X}_n}\right\|_2\left\|\sum_{j=1}^{n}w_i^X(x_j)\,x_j - \mathbb{E}_{\mu_X}(w_i^X(X)X)\right\|_2}_{\text{Term (II.2)}}.$$

For Term (II.1), we have:

$$\text{Term (II.1)} = \left\|\hat{m}_i^{\mathbb{X}_n}\right\|_2 \cdot \left\|\frac{1}{n}\sum_{j=1}^{n}\boldsymbol{x}_j\left(e^{-\sigma(\|\boldsymbol{x}_j\|_2 - r_{q_i})} - e^{-\sigma(\|\boldsymbol{x}_j\|_2 - \hat{r}_{q_i})}\right)\right\|_2$$

$$\leq \frac{1}{n}\left\|\hat{m}_i^{\mathbb{X}_n}\right\|_2\sum_{j=1}^{n}\left(\|\boldsymbol{x}_j\|_2\left|e^{-\sigma(\|\boldsymbol{x}_j\|_2 - r_{q_i})} - e^{-\sigma(\|\boldsymbol{x}_j\|_2 - \hat{r}_{q_i})}\right|\right)$$

$$\leq \frac{\sqrt{2\sigma}e^{-1/2}}{n}\sum_{j=1}^{n}\left\|\hat{m}_i^{\mathbb{X}_n}\right\|_2\|\boldsymbol{x}_j\|_2|r_{q_i} - \hat{r}_{q_i}|.$$

From Lemma 6, we know $\mathbb{E}(\|\hat{m}_i^{\mathbb{X}_n}\|_2\|\boldsymbol{x}_j\|_2|r_{q_i} - \hat{r}_{q_i}|) = O(n^{-1/2})$, which implies $\mathbb{E}(\text{Term (II.1)}) = O(n^{-1/2})$. For Term (II.2), since $w_i^X(x) \leq 1$, we have $\mathbb{E}\|w(X)X\|_2^2 < \infty$. Using the Cauchy-Schwarz inequality, we get:

$$\mathbb{E}(\text{Term (II.2)}) \leq \sqrt{\mathbb{E}\left\|\hat{m}_i^{\mathbb{X}_n}\right\|_2^2 \cdot \mathbb{E}\left(\mathbb{E}_{\mu_X}\left\|w_i^X(X)X\right\| - \frac{1}{n}\sum_{j=1}^{n}w_i^X(\boldsymbol{x}_j)\boldsymbol{x}_j\right\|_2^2\right)}$$

$$= \sqrt{\mathbb{E}\left\|\hat{m}_i^{\mathbb{X}_n}\right\|_2^2}\sqrt{\frac{1}{n}\operatorname{tr}\operatorname{Cov}\left(w_i^X(\boldsymbol{x}_1)\boldsymbol{x}_1\right)} = O(n^{-1/2}).$$

Thus, we also have $\mathbb{E}(\text{Term (II)}) = O(n^{-1/2})$. In summary, we have shown that $\mathbb{E}(d(\boldsymbol{m}_i^X, \hat{\boldsymbol{m}}_i^{\mathbb{X}_n})) = O(n^{-1/2})$, which means the convergence rate for the QDM Discrepancy is $O(n^{-p/2})$. For typical cases where $d \geq 2$, this rate is faster than the Wasserstein rate, so the overall convergence is dominated by the Wasserstein Discrepancy. Therefore, we obtain the final rate:

$$\mathbb{E}(\mathcal{QD}_p(\mathcal{X}, \mathcal{X}_n)) = O(n^{-1/d}).$$

$\square$

### B.3 PROOF OF THEOREM 3

The proof for the Identity of Indiscernibles is analogous to that of Theorem 1, and symmetry holds trivially by definition. Therefore, we focus on proving the Triangle Inequality. We begin by introducing the following well-known Gluing Lemma.

**Lemma 7 (Gluing Lemma)** *For three mm-spaces $\mathcal{X}, \mathcal{Y}, \mathcal{Z}$ with corresponding measures $\mu_X, \mu_Y, \mu_Z$, and given couplings $\pi_{X,Z} \in \Pi(\mu_X, \mu_Z)$ and $\pi_{Y,Z} \in \Pi(\mu_Y, \mu_Z)$, there exists a joint coupling $\pi_{X,Y,Z} \in \Pi(\mu_X, \mu_Y, \mu_Z)$ such that its marginals satisfy*

$$(\text{proj}_{(X,Z)})_\#\pi_{X,Y,Z} = \pi_{X,Z}, \quad (\text{proj}_{(Y,Z)})_\#\pi_{X,Y,Z} = \pi_{Y,Z},$$

*and its $(X,Y)$-marginal, $(\text{proj}_{(X,Y)})_\#\pi_{X,Y,Z}$, is a coupling in $\Pi(\mu_X, \mu_Y)$.*

A proof of this lemma can be found in (Villani et al., 2008, Chap 1).

**Proof of Triangle Inequality.** Consider three mm-spaces $\mathcal{X} = (X, d_X, \mu_X)$, $\mathcal{Y} = (Y, d_Y, \mu_Y)$, and $\mathcal{Z} = (Z, d_Z, \mu_Z)$. Let $\pi^\star_{X,Y} \in \Pi(\mu_X, \mu_Y)$, $\pi^\star_{X,Z} \in \Pi(\mu_X, \mu_Z)$, and $\pi^\star_{Y,Z} \in \Pi(\mu_Y, \mu_Z)$ denote the respective pairwise optimal couplings for the IQDOT distance. According to Lemma 7, we can find a joint coupling $\pi_{X,Y,Z} \in \Pi(\mu_X, \mu_Y, \mu_Z)$ such that

$$(\text{proj}_{(X,Z)})_\# \pi_{X,Y,Z} = \pi^\star_{X,Z}, \quad (\text{proj}_{(Y,Z)})_\# \pi_{X,Y,Z} = \pi^\star_{Y,Z}.$$

Let $\pi_{X,Y} := (\text{proj}_{(X,Y)})_\# \pi_{X,Y,Z}$. By construction, $\pi_{X,Y} \in \Pi(\mu_X, \mu_Y)$. We can now bound the IQDOT distance as follows:

$$
\begin{aligned}
\mathcal{IQD}_p(\mathcal{X}, \mathcal{Y}) &= \mathbb{E}_{(0,1) \times \pi^\star_{X,Y}} \left( |\phi^X(x,q) - \phi^Y(y,q)|^p \right)^{1/p} \\
&\leq \mathbb{E}_{(0,1) \times \pi_{X,Y}} \left( |\phi^X(x,q) - \phi^Y(y,q)|^p \right)^{1/p} \\
&\leq \mathbb{E}_{(0,1) \times \pi_{X,Y,Z}} \left( (|\phi^X(x,q) - \phi^Z(z,q)| + |\phi^Z(z,q) - \phi^Y(y,q)|)^p \right)^{1/p} \\
&\leq \mathbb{E}_{(0,1) \times \pi^\star_{X,Z}} \left( |\phi^X(x,q) - \phi^Z(z,q)|^p \right)^{1/p} + \mathbb{E}_{(0,1) \times \pi^\star_{Y,Z}} \left( |\phi^Z(z,q) - \phi^Y(y,q)|^p \right)^{1/p} \\
&= \mathcal{IQD}_p(\mathcal{X}, \mathcal{Z}) + \mathcal{IQD}_p(\mathcal{Z}, \mathcal{Y}),
\end{aligned}
$$

where the last inequality follows from Minkowski's inequality. This completes the proof. $\square$

### B.4 RELATIONSHIP BETWEEN QDOT AND IQDOT

It is evident that both QDOT and IQDOT utilize the same QDMD representations in their numerical implementations. Since IQDOT is formulated as a pairwise 1-D Wasserstein distance, the inequality $\mathcal{IQD} \leq \mathcal{QD}$ consistently holds in practice. Consequently, the convergence $\mathcal{QD}(\mu_n, \mu) \to 0$ implies $\mathcal{IQD}(\mu_n, \mu) \to 0$. Furthermore, as both constitute distinct well-defined metrics, under the requisite conditions, they share the equivalence property regarding in distinguish ability: $\mathcal{QD}(\mu, \nu) = 0 \Leftrightarrow \mathcal{IQD}(\mu, \nu) = 0$.

### B.5 DISCUSSIONS ON THE SYMMETRIC SHAPES.

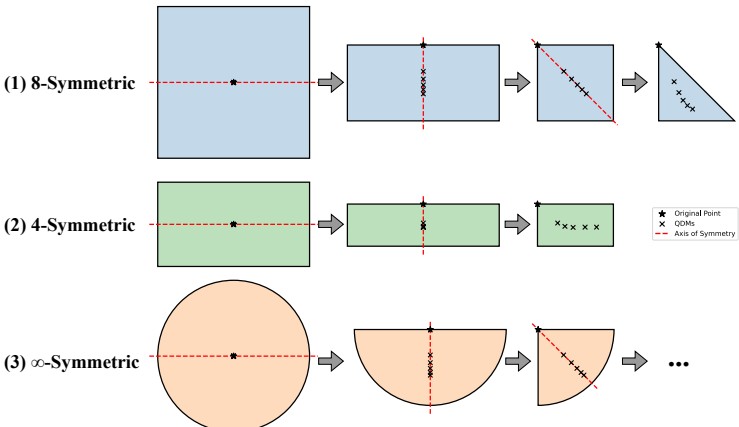

Figure 7: Illustrations of simple symmetric shapes. Note that for non-radially symmetric shapes, utilizing a finite number of QDMs suffices to satisfy the dimension conditions in Theorem 1 (i.e., ensuring that they are not collinear). For radially symmetric shapes, infinite subdivision implies a degeneration into a radial distribution.

## C IMPLEMENTATION DETAILS

### C.1 DETAILS OF ALGORITHM 1

The following provides further details on some of the operations and parameter choices in Algorithm 1.

**Data Initialization.** For two unknown distributions situated in Euclidean spaces, preprocessing the data is necessary. The objective is to obtain the distribution of distances relative to the barycenter, which is isometry-invariant. However, this initialization step is not necessary for intra-space comparisons or in cases where a correspondence between the origins is already known. The intuition of scalization aligns with the numerical implementation of Gromov-Wasserstein (Flamary et al., 2021), serving as a common technique for metric space alignment.

**Quantile Level Vector $q$.** The quantile levels in the vector $q$ are theoretically chosen from the open interval $(0, 1)$. In practice, we select these levels by taking equispaced points within a truncated interval $[\delta, 1 - \delta]$. The default value for $\delta$ is 0.1, which avoids numerical instability at the extreme tails of the distribution. For larger datasets where the empirical quantiles are more stable, a smaller $\delta$ may be used to satisfy the theoretical conditions more closely.

**Choice of the Number of Quantiles $k$.** A large number of quantiles, $k$, is often not required to achieve strong performance. For low-dimensional data (e.g., 3D point clouds), the trilateration condition is readily satisfied, and a small value such as $k = 5$ can be sufficient. For high-dimensional data, which often exhibits low-rank structure, a value of $k$ smaller than the ambient dimension can also be effective. While a larger $k$ can lead to improved numerical stability, we find that $k = 50$ provides a robust default choice across most applications.

**Bandwidth Parameter $\sigma$.** The choice of the bandwidth parameter $\sigma$ in the Gaussian kernel is related to the number of support points, $n$. A very small $\sigma$ will cause the weights to concentrate on a single point, while a very large $\sigma$ will lead to nearly uniform weights across all samples. Empirically, we recommend choosing $\sigma$ such that the maximum weight is on the order of the average weight, i.e., $\max_i\{w_i\} \sim n^{-1/2}$. In practice, a default value of $\sigma = 10$ provides robust and effective performance across most of our experiments.

We analyze the robustness of our method to these parameters through a toy example detailed in Appendix D.1.

## C.2 Algorithm of IQDOT

For numerical inputs, given by discrete sample matrices $\mathbf{X} \in \mathbb{R}^{n \times d}$ and $\mathbf{Y} \in \mathbb{R}^{m \times q}$ with corresponding probability vectors $p^X \in \Delta^{n-1}$ and $p^Y \in \Delta^{m-1}$, we now detail the numerical implementation of the IQDOT, as introduced in Definition 4. The procedure is summarized in Algorithm 2.

---

**Algorithm 2** Intergal-QDOT

---

**Require:** $\mathbf{X}, \mathbf{Y}, p^{\mathbf{X}}, p^{\mathbf{Y}}, q$

1: Initialize the data matrices $\mathbf{X}$ and $\mathbf{Y}$.
2: Compute the sample norms of $\mathbf{X}$ and $\mathbf{Y}$: $\phi_0^{\mathbf{X}} \leftarrow (\|x_i\|)_{1 \leq i \leq n}$, $\phi_0^{\mathbf{Y}} \leftarrow (\|y_j\|)_{1 \leq j \leq m}$.
3: Compute the $q$-quantiles of the sample norms:  $\triangleright \mathcal{O}(n \log n + m \log m)$

$$r^{\mathbf{X}} \leftarrow (F_{\|\mathbf{X}\|}^{-1}(q_1), \ldots, F_{\|\mathbf{X}\|}^{-1}(q_k)), \quad r^{\mathbf{Y}} \leftarrow (F_{\|\mathbf{Y}\|}^{-1}(q_1), \ldots, F_{\|\mathbf{Y}\|}^{-1}(q_k)).$$

4: **for** $i \leftarrow 1$ to $k$ **do**  $\triangleright \mathcal{O}(knd + kms)$
5:     Compute the quantile weights:

$$w_{ij}^{\mathbf{X}} \leftarrow \frac{p_j^X \exp\{-\sigma(d_j^{\mathbf{X}} - r_i^{\mathbf{X}})^2\}}{\sum_{j'=1}^n p_{j'}^X \exp\{-\sigma(d_{j'}^{\mathbf{X}} - r_i^{\mathbf{X}})^2\}}, \quad w_{ij}^{\mathbf{Y}} \leftarrow \frac{p_j^Y \exp\{-\sigma(d_j^{\mathbf{Y}} - r_i^{\mathbf{Y}})^2\}}{\sum_{j'=1}^m p_{j'}^Y \exp\{-\sigma(d_{j'}^{\mathbf{Y}} - r_i^{\mathbf{Y}})^2\}}$$

6:     Compute the $q_i$-quantile means: $m_i^X \leftarrow \mathbf{X}^\top w_i^{\mathbf{X}}, \quad m_i^Y \leftarrow \mathbf{Y}^\top w_i^{\mathbf{Y}}$
7:     Compute the distances to quantile means:

$$\phi_i^{\mathbf{X}} \leftarrow (\|x_j - m_i^X\|)_{1 \leq j \leq n}, \quad \phi_i^{\mathbf{Y}} \leftarrow (\|y_j - m_i^Y\|)_{1 \leq j \leq m}$$

8:     Calculate the 1D-$\mathcal{W}_p$ loss: $\mathcal{L}_i \leftarrow \mathcal{W}_p\left((\phi_i^{\mathbf{X}})_\# \mu_{\mathbf{X}}, (\phi_i^{\mathbf{Y}})_\# \mu_{\mathbf{Y}}\right)$  $\triangleright \mathcal{O}(n \log n + m \log m)$
9: **end for**
10: Set the final loss : $\mathcal{IQD}_p \leftarrow \frac{1}{k} \sum_{i=1}^k \mathcal{L}_i$
11: **return** $\mathcal{IQD}_p$

---

**Computational Cost.** Similar to the analysis for QDOT, the pre-computation of QDMs and QD-MDs is highly efficient. The final one-dimensional Wasserstein distances can also be computed in $\mathcal{O}(n \log n)$ time by North-West corner rule(Peyré et al., 2019). Consequently, the overall time complexity of the IQDOT algorithm is $\mathcal{O}(n \log n)$, assuming $k$ and the data dimensions are small constants.

# D ADDITIONAL EXPERIMENT RESULTS

## D.1 TOY EXAMPLE AND PARAMETER ANALYSIS

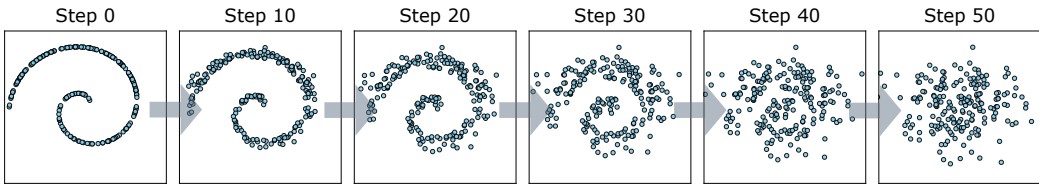

Figure 8: Illustration of the Wasserstein flow from a spiral point cloud to a random Gaussian noise distribution over 50 steps.

To provide an intuitive visualization of QDOT's behavior and to analyze its sensitivity to key parameters, we present a toy example. As illustrated in Figure 8, the experiment consists of a Wasserstein flow sequence that interpolates between a 2D spiral distribution and a random Gaussian noise distribution. Each point cloud in the 50-step sequence comprises $n = 200$ points. We conduct a series of tests to: (1) compare the dissimilarity trends produced by different geometric metrics as the spiral deforms into noise; (2) analyze the effect of varying the bandwidth parameter $\sigma$; (3) analyze the effect of varying the quantile interval parameter $\delta$; and (4) assess the impact of different initialization and scaling strategies.

The results, presented in Figure 9, reveal several key insights. First, when compared to baselines such as GW, EGW, SGW, and RISGW, both QDOT and IQDOT produce significantly smoother and more stable dissimilarity curves. Second, the analysis of the bandwidth parameter shows that for $\sigma = 1$ and $\sigma = 10$, the resulting curves are stable and smooth. A larger value of $\sigma = 100$ introduces some volatility, which can be attributed to the interaction between a high bandwidth and the relatively small sample size. Third, the dissimilarity curve is largely insensitive to the choice of the quantile interval parameter $\delta$ in this experiment, indicating good robustness. Finally, while different initialization and scaling strategies affect the absolute values of the dissimilarity, they all produce smooth, monotonically increasing curves that preserve the overall trend, demonstrating the robustness of the underlying geometric representation.

## D.2 CROSS SPACE TASKS

**Evaluation Metrics.** For the numerical experiments, the inputs are a 3D point cloud sequence, represented by sample matrices $\mathbf{X} \in \mathbb{R}^{n \times 3}$, and a reference shape. The reference is provided both in its original 3D form, $\mathbf{Y} \in \mathbb{R}^{m \times 3}$, and as a 2D projection, $\hat{\mathbf{Y}} \in \mathbb{R}^{m \times 2}$. Both distributions have corresponding probability vectors, $\boldsymbol{p}^{\mathbf{X}}$ and $\boldsymbol{p}^{\mathbf{Y}}$, we assume access to a ground-truth alignment between the sequence frame $\mathbf{X}$ and the reference shape $\mathbf{Y}$, denoted by a map $\mathcal{T}$, i.e., $\mathbf{X} = \mathcal{T}(\mathbf{Y})$. Algorithm 1 computes a coupling matrix $\Pi \in \mathbb{R}^{n \times m}$ between the input distributions, which satisfies the marginal constraints $\Pi \mathbf{1}_m = \boldsymbol{p}^{\mathbf{X}}$ and $\Pi^\top \mathbf{1}_n = \boldsymbol{p}^{\mathbf{Y}}$. To evaluate the quality of the resulting match, we define the following metrics.

The **Transformed Mean Squared Error (TMSE)** evaluates the alignment cost in the original 3D ambient space. Even when the coupling $\Pi$ is computed between a 3D shape $\mathbf{X}$ and a 2D projection $\hat{\mathbf{Y}}$, the error is measured by using the original 3D coordinates of the reference shape, $\mathbf{Y}$:

$$\text{TMSE} = \sum_{i=1}^{n} \sum_{j=1}^{m} \Pi_{ij} d_X(\boldsymbol{x}_i, \mathcal{T}(\boldsymbol{y}_j));$$

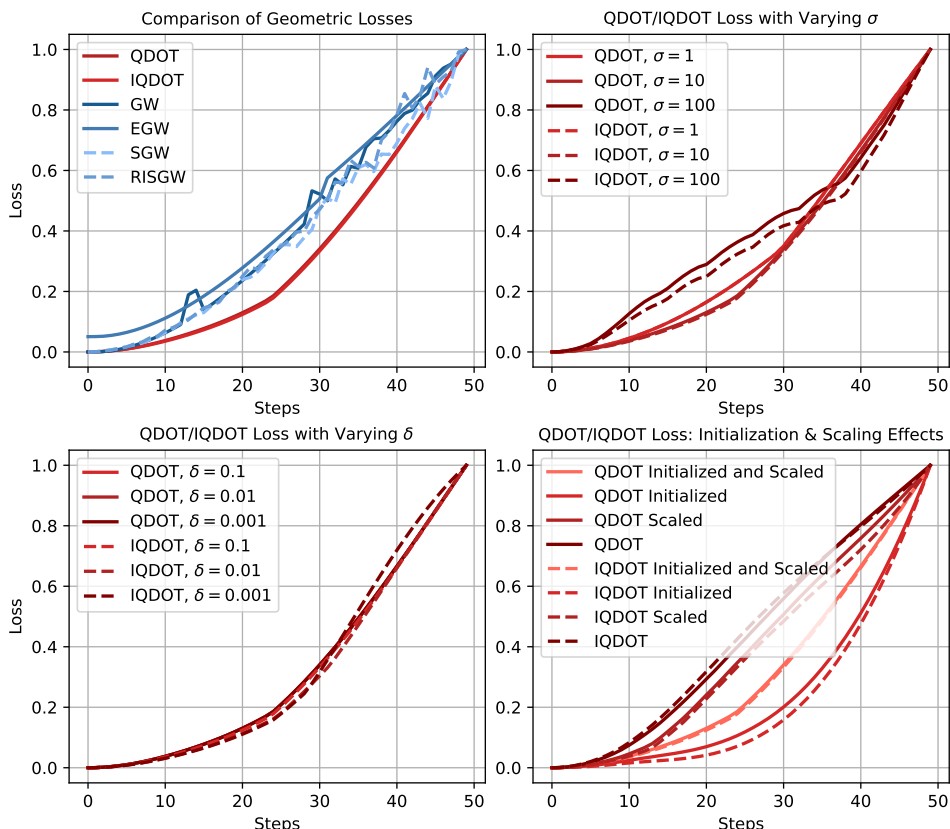

Figure 9: Results from the four experimental settings. The x-axis represents the step in the Wasserstein flow sequence, and the y-axis represents the computed dissimilarity (loss).

The **Inlier Ratio (IR)** then measures the total probability mass of the coupling $\Pi$ that is placed on geometrically correct matches. A match between $\boldsymbol{x}_i$ and $\boldsymbol{y}_j$ is considered correct if $\boldsymbol{x}_i$ is close to the ground-truth location corresponding to $\boldsymbol{y}_j$:

$$\text{IR} = \sum_{i=1}^{n} \sum_{j=1}^{m} \Pi_{ij} \mathbf{1}(d_X(\boldsymbol{x}_i, \mathcal{T}(\boldsymbol{y}_j)) < \tau) \times 100\%;$$

where the threshold $\tau$ is set to the 0.2-quantile of the intra-point distance distribution within $\mathbf{X}$, i.e., $\tau = F_{d_X}^{-1}(0.2)$.

**Additional comparisons.** To demonstrate the efficiency of QDOT, we compared it against several fast GW/EGW variants, such as PGW (Kerdoncuff et al., 2021), QGW(Chowdhury et al., 2021), and LRGW(Scetbon et al., 2022), as shown below. The penalty parameter for all methods was set to 0.01.

Table 5: Additional comparisons on the camel-gallop dataset

| Methods | Transformed MSE ↓ | | | | | Inlier Ratio (%) ↑ | | | | | $Time_{(\times 10^2 s)}$ |
|---|---|---|---|---|---|---|---|---|---|---|---|
| | 3D | 2D$_{1st}$ | 2D$_{2nd}$ | 2D$_{3rd}$ | Avg. | 3D | 2D$_{1st}$ | 2D$_{2nd}$ | 2D$_{3rd}$ | Avg. | |
| PGW | 0.35 | 0.34 | 0.32 | 0.33 | 0.33 | 53.85 | 55.38 | 55.71 | 51.35 | 54.07 | 0.42 |
| QGW | 0.56 | 0.62 | 0.53 | 0.64 | 0.59 | 23.42 | 18.30 | 27.98 | 17.74 | 21.85 | **0.19** |
| LRGW | 0.38 | 0.35 | 0.42 | 0.45 | 0.40 | 46.00 | 49.53 | 39.40 | 36.35 | 42.82 | 1.53 |
| **QDOT** | **0.22** | **0.24** | **0.23** | **0.25** | **0.24** | **71.15** | **63.75** | **68.48** | **63.20** | **66.65** | 0.74 |

As shown in Table 5, it is evident that the QDOT method achieves the best performance under the same order of computational cost, highlighting its efficiency.

**Distortion.** We substitute the coupling obtained via Algorithm 1 into the Gromov-Wasserstein loss function in 10. By plotting the loss curves over the optimization steps, we compare the behavior of the QDOT coupling with respect to both the QDOT loss and the GW loss. We employed equidistant sampling to select approximately $n \approx 1000$ points. The experimental results are presented below.

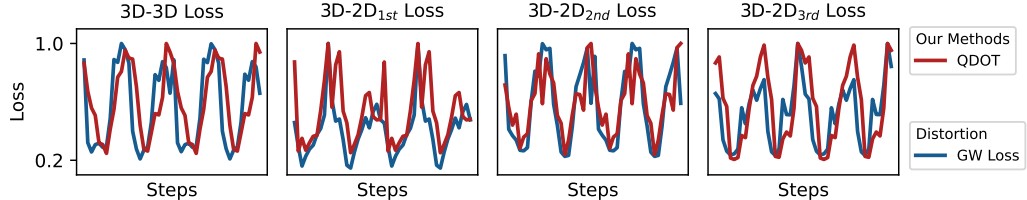

Figure 10: Distortion results on the camel-gallop dataset. The x-axis represents the time steps of the galloping camel sequence, and the y-axis indicates the loss corresponding to the coupling at each time step.

As shown in Figure 10, the QDOT loss and the GW loss maintain a high degree of consistency under the same coupling. This validates the effectiveness of QDOT as a metric.

**Ablation study.** To analyze the sensitivity of the results to the parameter settings, we consider $\sigma \in \{200, 1000, 5000\}$ and $k \in \{20, 100, 500\}$. The experimental results are presented below.

Table 6: QDOT results with different parameter selections on the camel-gallop dataset.

| Parameters | *Transformed MSE* ↓ | | | | | *Inlier Ratio (%)* ↑ | | | | |
|---|---|---|---|---|---|---|---|---|---|---|
| | 3D | 2D$_{1st}$ | 2D$_{2nd}$ | 2D$_{3rd}$ | Avg. | 3D | 2D$_{1st}$ | 2D$_{2nd}$ | 2D$_{3rd}$ | Avg. |
| $\sigma = 200, k = 20$ | 0.22 | 0.25 | 0.23 | 0.26 | 0.24 | 70.49 | 62.79 | 67.70 | 62.82 | 65.95 |
| $\sigma = 200, k = 100$ | 0.22 | 0.24 | 0.23 | 0.25 | 0.24 | 70.82 | 63.34 | 68.05 | 63.09 | 66.33 |
| $\sigma = 200, k = 500$ | 0.22 | 0.24 | 0.23 | 0.25 | 0.24 | 70.79 | 63.36 | 68.05 | 63.09 | 66.32 |
| $\sigma = 1000, k = 20$ | 0.22 | 0.24 | 0.23 | 0.25 | 0.24 | 70.68 | 63.68 | 68.11 | 62.82 | 66.32 |
| $\sigma = 1000, k = 100$ | 0.22 | 0.24 | 0.23 | 0.25 | 0.24 | 71.15 | 63.75 | 68.49 | 63.20 | 66.65 |
| $\sigma = 1000, k = 500$ | 0.22 | 0.24 | 0.23 | 0.25 | 0.24 | 71.12 | 63.73 | 68.46 | 63.18 | 66.62 |
| $\sigma = 5000, k = 20$ | 0.22 | 0.24 | 0.23 | 0.26 | 0.24 | 70.83 | 64.46 | 68.07 | 62.71 | 66.52 |
| $\sigma = 5000, k = 100$ | 0.22 | 0.24 | 0.23 | 0.25 | 0.23 | 71.35 | 64.04 | 68.52 | 63.22 | 66.78 |
| $\sigma = 5000, k = 500$ | 0.22 | 0.24 | 0.23 | 0.25 | 0.23 | 71.33 | 63.97 | 68.50 | 63.23 | 66.76 |
| **Summary**$_{\text{mean}\pm\text{std}}$ | **0.22** | **0.24** | **0.23** | **0.25** | **0.24** | **70.95** | **63.68** | **68.22** | **63.04** | **66.47** |
| | (±**0.00**) | (±**0.01**) | (±**0.00**) | (±**0.01**) | (±**0.01**) | (±**0.28**) | (±**0.45**) | (±**0.27**) | (±**0.19**) | (±**0.25**) |

As shown in Table 6, the QDOT alignment results are not particularly sensitive to the selection of $\sigma$ and $k$. Generally, $k = 100$ yields sufficiently good performance, with larger $k$ providing minimal additional gains.

### D.3 TRANSFER LEARNING

**Comparison with other sliced methods.** Setting $n = 2048$, we compared the results against recently efficient sliced methods, including Sliced Wasserstein(Bonneel et al., 2015), Max-Sliced Wasserstein(Deshpande et al., 2019), and Hilbert Curve Projection(Li et al., 2024), as shown below.

As shown in Table 7, it is evident that IQDOT exhibits superior comprehensive performance. Notably, in the 'guitar' category, QDOT achieves near-perfect accuracy, whereas other methods fail almost entirely. This is attributed to the differing orientations of the guitars in the two datasets. As illustrated in Figure 11, guitars in ModelNet are horizontally aligned, while those in ShapeNet are vertically aligned, leading to the failure of other sliced-based methods. This highlights the critical importance of isometry invariance in transfer learning.

**Ablation study.** On the ablation study, we consider $n = 1024$, $\sigma \in \{50, 200, 1000\}$, and $k \in \{10, 50, 200\}$. The results are shown below.

As indicated in Table 8, apart from some numerical instability when $k = 10$, the differences between results are not substantial. This demonstrates that our method is relatively insensitive to parameter variations.

Table 7: Additional Transfer Learning Classification Accuracy Results (%)

| Methods (Mo→Sh) | airplane | car | chair | guitar | lamp | laptop | table | Avg. | Time(h) |
|---|---|---|---|---|---|---|---|---|---|
| SW | 96.69 | 95.77 | 94.81 | 0.00 | 68.71 | 97.78 | 89.60 | 86.03 | 17.36 |
| MSW | 96.25 | 95.55 | 95.53 | 0.00 | 67.29 | 98.23 | 89.91 | 86.09 | 17.05 |
| HCP | 93.38 | 95.55 | 95.85 | 0.25 | 74.73 | 98.23 | 91.29 | 86.90 | 22.58 |
| **IQDOT** | 95.35 | 87.86 | 85.01 | 96.44 | 52.55 | 98.00 | 87.15 | 85.42 | 5.98 |
| Methods (Sh→Mo) | airplane | car | chair | guitar | lamp | laptop | table | Avg. | Time(h) |
| SW | 100.00 | 95.96 | 96.76 | 0.00 | 93.06 | 95.86 | 97.76 | 89.36 | 17.36 |
| MSW | 99.86 | 95.62 | 97.67 | 0.00 | 93.75 | 96.45 | 96.54 | 89.45 | 17.05 |
| HCP | 98.48 | 96.63 | 97.47 | 0.00 | 92.36 | 97.63 | 98.78 | 89.52 | 22.58 |
| **IQDOT** | 98.89 | 91.91 | 93.22 | 98.03 | 77.77 | 97.63 | 97.56 | 95.05 | 5.98 |

Table 8: Point Cloud Transfer Learning Results with different QDOT parameter selections (%)

| Params. (Mo→Sh) | airplane | car | chair | guitar | lamp | laptop | table | Avg. |
|---|---|---|---|---|---|---|---|---|
| $\sigma = 50, k = 10$ | 94.80 | 78.06 | 81.93 | 96.82 | 50.42 | 96.90 | 83.53 | 82.53 |
| $\sigma = 50, k = 50$ | 95.46 | 82.74 | 82.70 | 96.44 | 50.81 | 96.67 | 84.50 | 83.46 |
| $\sigma = 50, k = 200$ | 95.32 | 82.96 | 82.78 | 96.44 | 50.36 | 96.90 | 84.50 | 83.42 |
| $\sigma = 200, k = 10$ | 94.94 | 80.07 | 83.10 | 97.46 | 50.10 | 96.45 | 84.03 | 83.12 |
| $\sigma = 200, k = 50$ | 94.86 | 84.18 | 83.82 | 96.56 | 51.13 | 96.45 | 84.95 | 83.89 |
| $\sigma = 200, k = 200$ | 95.65 | 84.41 | 84.11 | 96.32 | 50.23 | 97.12 | 85.30 | 84.15 |
| $\sigma = 1000, k = 10$ | 94.98 | 80.73 | 82.01 | 97.71 | 48.48 | 95.57 | 82.68 | 82.26 |
| $\sigma = 1000, k = 50$ | 95.35 | 84.08 | 84.78 | 95.93 | 50.81 | 97.12 | 84.90 | 84.14 |
| $\sigma = 1000, k = 200$ | 95.72 | 85.63 | 84.73 | 96.19 | 50.55 | 96.90 | 85.18 | 84.37 |
| **Summary**$_{mean\pm std}$ | **95.23** | **82.54** | **83.33** | **96.65** | **50.32** | **96.68** | **84.40** | **83.48** |
| | (**±0.33**) | (**±2.30**) | (**±1.02**) | (**±0.55**) | (**±0.72**) | (**±0.46**) | (**±0.81**) | (**±0.70**) |
| Params. (Sh→Mo) | airplane | car | chair | guitar | lamp | laptop | table | Avg. |
| $\sigma = 50, k = 10$ | 98.62 | 83.84 | 89.79 | 97.65 | 79.17 | 95.27 | 94.92 | 92.58 |
| $\sigma = 50, k = 50$ | 98.90 | 86.53 | 90.19 | 98.43 | 78.47 | 95.27 | 95.33 | 93.13 |
| $\sigma = 50, k = 200$ | 98.76 | 88.89 | 90.09 | 98.82 | 79.86 | 95.86 | 95.73 | 93.49 |
| $\sigma = 200, k = 10$ | 98.62 | 83.16 | 90.60 | 97.25 | 79.86 | 95.27 | 96.14 | 92.97 |
| $\sigma = 200, k = 50$ | 98.89 | 87.20 | 91.10 | 98.43 | 81.94 | 97.04 | 96.74 | 93.97 |
| $\sigma = 200, k = 200$ | 98.76 | 86.87 | 92.11 | 98.82 | 79.86 | 97.04 | 96.54 | 94.14 |
| $\sigma = 1000, k = 10$ | 98.48 | 81.14 | 88.27 | 97.65 | 76.39 | 96.45 | 95.12 | 91.76 |
| $\sigma = 1000, k = 50$ | 98.21 | 83.84 | 89.69 | 99.22 | 81.25 | 97.63 | 96.95 | 93.13 |
| $\sigma = 1000, k = 200$ | 97.80 | 83.84 | 91.30 | 98.82 | 79.86 | 97.04 | 94.92 | 93.10 |
| **Summary**$_{mean\pm std}$ | **98.56** | **85.03** | **90.35** | **98.34** | **79.63** | **96.32** | **95.82** | **93.14** |
| | (**±0.34**) | (**±2.31**) | (**±1.04**) | (**±0.63**) | (**±1.50**) | (**±0.87**) | (**±0.75**) | (**±0.67**) |

**ModelNet40** ⟺ **ShapeNetPart**

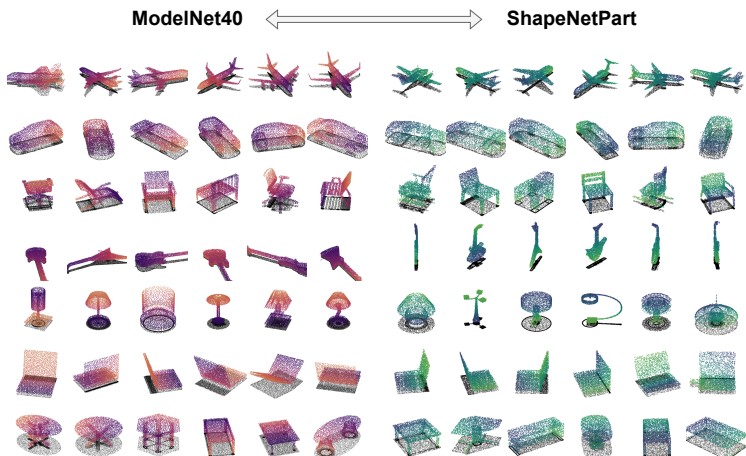

Figure 11: Visualization of the ModelNet40 and ShapeNetPart datasets.

## D.4 HIGH DIMENSIONAL CAPABILITIES.

Following the experimental setup in Li et al. (2024), we sample source and target data from multivariate normal distributions $\mu_X \sim N(\mathbf{0}_d, \mathbf{\Sigma}_X)$ and $\mu_Y \sim N(\mathbf{0}_d, \mathbf{\Sigma}_Y)$, respectively, where the covariance matrices are defined as $\mathbf{\Sigma}_X = \text{diag}(3\mathbf{I}_2, \mathbf{I}_{d-2})$ and $\mathbf{\Sigma}_Y = \text{diag}(3\mathbf{I}_2 + 3\theta\mathbf{B}_2, \mathbf{I}_{d-2})$ and $\mathbf{I}_2, \mathbf{B}_2$ are identity and backward identity matrices. We evaluate the distance between these distributions as a function of the varying parameter $\theta$. We fix the sample size at $n = 100$ and consider high-dimensional settings with $d \in \{50, 100, 200\}$. We compare our method against GW, EGW, SGW, and RISGW. The results are presented below:

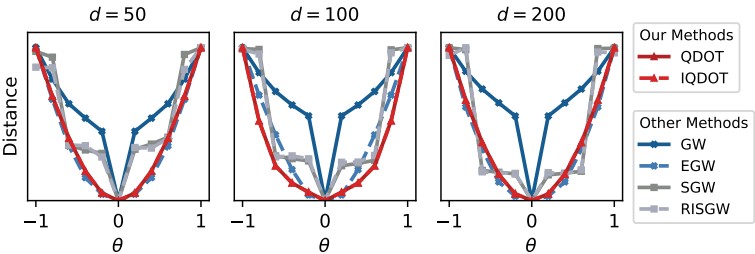

Figure 12: High dimension results.

As illustrated in Figure 12, QDOT/IQDOT and EGW exhibit remarkably stable and smooth trends across varying dimensions and $\theta$ values. In contrast, other methods display significant volatility. These results underscore the robustness of QDOT in high-dimensional settings.

## D.5 MOLECULE GENERATION

**Visualization results.** The visualization results of molecular generation are presented in Figure 13. It can be observed that the IQDOT-regularized model tends to preserve stronger structural information and exhibits greater stability.

For the molecular generation experiment, we present the complete results in Table 9.

As shown in Table 9, simply reducing the weight of the MSE loss does not yield significant improvements for the baseline models. We also observe that the benefits of the IQDOT loss do not necessarily increase monotonically with its proportion in the final objective. Overall, these results suggest that the standard MSE loss primarily captures information about absolute atomic positions, while the IQDOT loss focuses on the intrinsic molecular structure. The strongest performance is

Table 9: Molecule Generation Results on QM9 Dataset.

| Methods | Atom sta. (%) | Mol sta. (%) | Validity (%) | Uniqueness (%) | V * U (%) |
|---|---|---|---|---|---|
| Data | 99.0 | 95.2 | 97.7 | 100.0 | 97.7 |
| EDM | $98.70_{\pm 0.01}$ | $86.56_{\pm 0.27}$ | $93.73_{\pm 0.12}$ | $98.19_{\pm 0.14}$ | $92.04_{\pm 0.03}$ |
| $\rightarrow$MSE$_{0.1}$ | $98.81_{\pm 0.04}$ | $87.63_{\pm 0.21}$ | $94.54_{\pm 0.15}$ | $98.17_{\pm 0.08}$ | $92.81_{\pm 0.20}$ |
| $\rightarrow$MSE$_{0.2}$ | $98.78_{\pm 0.01}$ | $87.60_{\pm 0.30}$ | $94.18_{\pm 0.15}$ | $98.25_{\pm 0.10}$ | $92.54_{\pm 0.19}$ |
| $\rightarrow$MSE$_{0.3}$ | $98.68_{\pm 0.07}$ | $86.04_{\pm 0.74}$ | $93.83_{\pm 0.43}$ | $\mathbf{98.47_{\pm 0.22}}$ | $92.39_{\pm 0.55}$ |
| $\rightarrow$MSE$_{0.4}$ | $98.80_{\pm 0.02}$ | $87.44_{\pm 0.35}$ | $94.59_{\pm 0.25}$ | $98.38_{\pm 0.07}$ | $\mathbf{93.05_{\pm 0.31}}$ |
| $\rightarrow$IQDOT$_{0.1}$ | $99.15_{\pm 0.01}$ | $90.91_{\pm 0.21}$ | $96.75_{\pm 0.33}$ | $92.06_{\pm 0.15}$ | $89.06_{\pm 0.19}$ |
| $\rightarrow$IQDOT$_{0.2}$ | $99.30_{\pm 0.03}$ | $92.73_{\pm 0.21}$ | $97.63_{\pm 0.11}$ | $84.48_{\pm 0.59}$ | $82.48_{\pm 0.49}$ |
| $\rightarrow$IQDOT$_{0.3}$ | $99.35_{\pm 0.06}$ | $93.29_{\pm 0.81}$ | $97.80_{\pm 0.22}$ | $83.05_{\pm 2.27}$ | $81.21_{\pm 2.03}$ |
| $\rightarrow$IQDOT$_{0.4}$ | $\mathbf{99.48_{\pm 0.01}}$ | $\mathbf{94.37_{\pm 0.18}}$ | $\mathbf{98.16_{\pm 0.13}}$ | $79.42_{\pm 0.27}$ | $77.96_{\pm 0.17}$ |
| UniGEM | $98.90_{\pm 0.03}$ | $89.40_{\pm 0.02}$ | $94.58_{\pm 0.07}$ | $98.07_{\pm 0.05}$ | $92.75_{\pm 0.11}$ |
| $\rightarrow$MSE$_{0.1}$ | $98.91_{\pm 0.04}$ | $89.00_{\pm 0.22}$ | $95.00_{\pm 0.16}$ | $98.04_{\pm 0.07}$ | $93.13_{\pm 0.10}$ |
| $\rightarrow$MSE$_{0.2}$ | $98.65_{\pm 0.06}$ | $85.27_{\pm 0.35}$ | $93.93_{\pm 0.50}$ | $98.05_{\pm 0.13}$ | $92.15_{\pm 0.42}$ |
| $\rightarrow$MSE$_{0.3}$ | $99.00_{\pm 0.07}$ | $89.48_{\pm 0.70}$ | $95.16_{\pm 0.16}$ | $\mathbf{98.09_{\pm 0.23}}$ | $93.34_{\pm 0.18}$ |
| $\rightarrow$MSE$_{0.4}$ | $99.03_{\pm 0.05}$ | $90.09_{\pm 0.53}$ | $95.17_{\pm 0.30}$ | $98.06_{\pm 0.13}$ | $93.32_{\pm 0.18}$ |
| $\rightarrow$IQDOT$_{0.1}$ | $99.24_{\pm 0.03}$ | $92.73_{\pm 0.09}$ | $96.85_{\pm 0.11}$ | $96.45_{\pm 0.21}$ | $\mathbf{93.42_{\pm 0.23}}$ |
| $\rightarrow$IQDOT$_{0.2}$ | $99.35_{\pm 0.07}$ | $94.06_{\pm 0.43}$ | $97.42_{\pm 0.28}$ | $92.59_{\pm 0.13}$ | $90.20_{\pm 0.36}$ |
| $\rightarrow$IQDOT$_{0.3}$ | $\mathbf{99.44_{\pm 0.01}}$ | $\mathbf{95.23_{\pm 0.18}}$ | $\mathbf{97.94_{\pm 0.06}}$ | $85.64_{\pm 0.28}$ | $83.88_{\pm 0.30}$ |
| $\rightarrow$IQDOT$_{0.4}$ | $99.38_{\pm 0.02}$ | $94.81_{\pm 0.25}$ | $97.66_{\pm 0.24}$ | $78.67_{\pm 0.38}$ | $76.83_{\pm 0.39}$ |

[1] The notation $\rightarrow$IQDOT$_{\mathbf{0.2}}$ indicates that the objective function $\mathcal{L} = 0.2\mathcal{L}_{\text{IQDOT}} + 0.8\mathcal{L}_{\text{EDM}}$.

achieved when these two complementary objectives are fused, indicating that they provide a more comprehensive learning signal.

As illustrated in Figure 14, models trained with the IQDOT loss achieve stable convergence on both Atom and Molecule Stability metrics in just 1000 epochs, for both the EDM and UniGEM architectures. In contrast, the baseline models and those fine-tuned by simply adjusting the MSE loss weight require significantly more training: approximately 2500 epochs on EDM and over 2000 epochs on UniGEM to reach a similar level of stability. This ablation confirms that simply reducing the MSE loss weight does not accelerate convergence, highlighting the efficiency gains are a direct result of the structural guidance provided by the IQDOT loss.

## THE USE OF LARGE LANGUAGE MODELS

We acknowledge the use of Large Language Models in the preparation of this manuscript. Their role was strictly limited to functioning as a tool for language polishing and improving readability. We affirm that all definitions and theorems presented in this paper are the original intellectual contributions of the authors. The core code is intricately linked to the algorithm we propose, and the experiments were designed specifically to investigate the properties of our method. We take full responsibility for the integrity of this work, which is free from any form of academic misconduct.

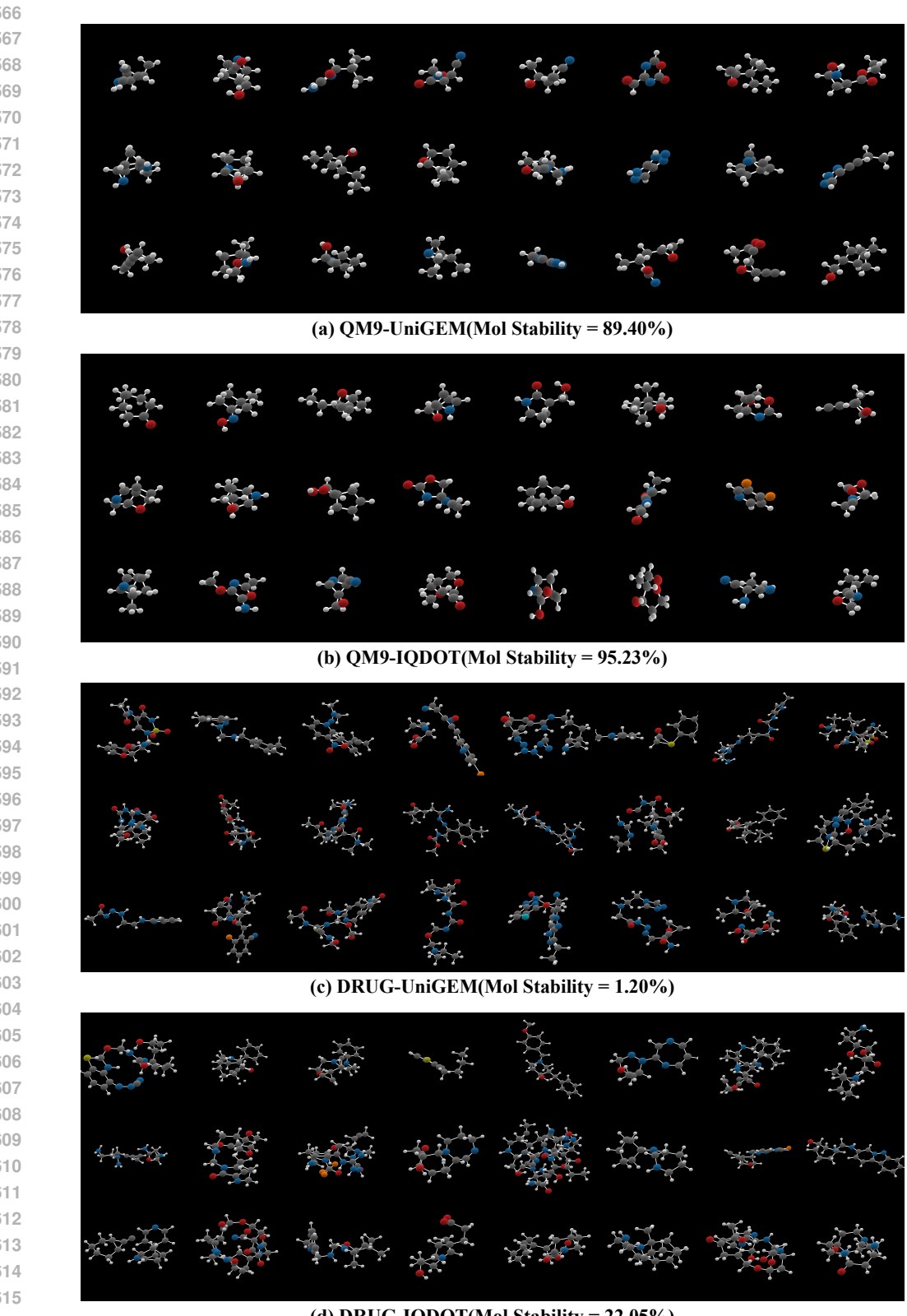

(a) QM9-UniGEM(Mol Stability = 89.40%)

(b) QM9-IQDOT(Mol Stability = 95.23%)

(c) DRUG-UniGEM(Mol Stability = 1.20%)

(d) DRUG-IQDOT(Mol Stability = 22.05%)

Figure 13: Visualization of molecular generation results comparing IQDOT and UniGEM on the QM9 and GEOM-Drugs datasets.

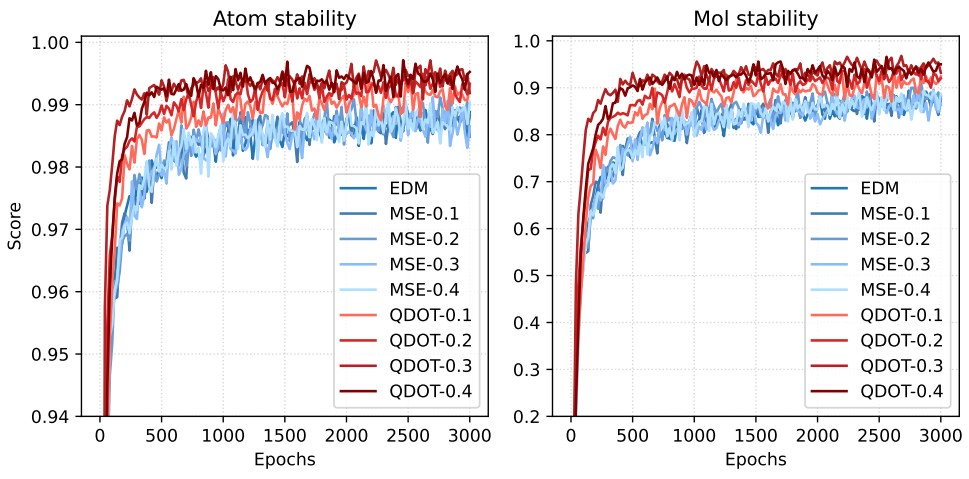

(a) Stability trends across epochs on the EDM model

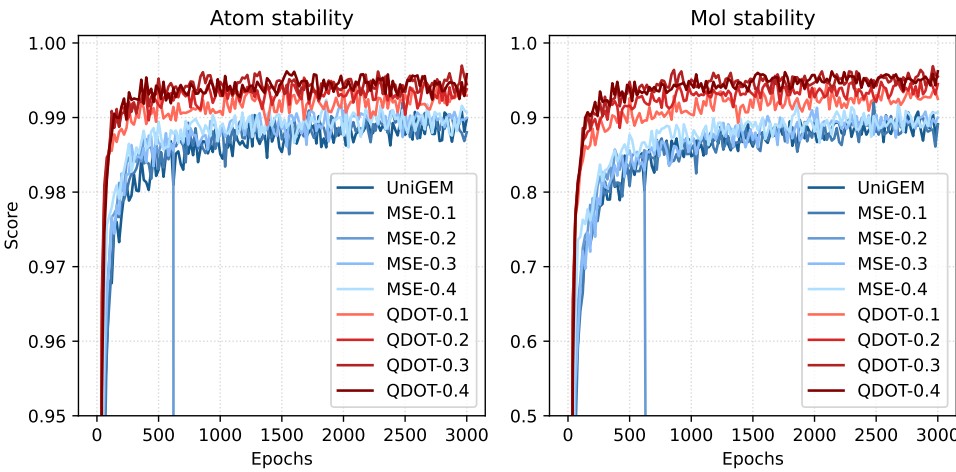

(b) Stability trends across epochs on the UniGEM model

Figure 14: Convergence trends of Atom Stability and Molecule Stability for various methods on the EDM and UniGEM models. The horizontal axis represents the number of training epochs, and the vertical axis represents the stability percentage. The comparison includes the baseline models and variants with different weights for the MSE and IQDOT loss terms (MSE-0.1 to MSE-0.4 and IQDOT-0.1 to IQDOT-0.4).

