# OpenReview forum: "QDOT: An Efficient Quantile-weighted Distance Metric for Geometric Comparison via Optimal Transport"
_ICLR.cc/2026/Conference — Submitted to ICLR 2026_

### Official Review · Reviewer_dUpr · 2025-10-19

**Soundness:** 2
**Presentation:** 3
**Contribution:** 3
**Rating:** 6
**Confidence:** 4

**Summary:**

The paper “QDOT: An Efficient Quantile-Weighted Distance Metric for Geometric Comparison via Optimal Transport” proposes a new geometric metric called QDOT (Quantile-weighted Distance Optimal Transport) to efficiently compare data distributions across different metric spaces. Traditional methods like the Wasserstein and Gromov-Wasserstein distances either lose geometric information or are computationally expensive. QDOT overcomes these issues by using quantile-weighted distance means (QDMs) as isometry-invariant anchors and constructing a quantile-based representation that preserves geometric structure. The method achieves theoretical guarantees, it is a valid metric on isometry classes, has convergence rates comparable to the Wasserstein distance, and supports rotation and translation invariance. The authors also propose an integral variant (IQDOT) with quasi-linear time complexity $\mathcal{O}(n\log n). Experiments on tasks like cross-space point cloud matching, transfer learning, and molecule generation show that QDOT offers empirical performance and computational efficiency, outperforming existing geometric comparison methods while maintaining theoretical soundness.

**Strengths:**

1. QDOT is rigorously defined as a proper metric on isometry classes, with proofs ensuring identity, symmetry, and triangle inequality properties.

2. By using quantile-weighted anchors, it naturally handles transformations like rotation, translation, and reflection, crucial for geometric data.

3. The integral version (IQDOT) achieves quasi-linear complexity $\mathcal{O}(n\log n)$, significantly faster than Gromov–Wasserstein and similar metrics.

4. Demonstrated good performance across diverse tasks, cross-space alignment, transfer learning, and molecular generation, showing good generalization ability.

5. Experiments and comparisons with baselines (GW, EGW, SGW, etc.) confirm both efficiency and accuracy advantages.

6. Integrating QDOT as a loss function improves deep learning models (e.g., molecule generation), showing its usefulness beyond theoretical contexts.

**Weaknesses:**

1. The quantile-weighted representation and multiple layers of transformation make the method conceptually and computationally more intricate to implement.

2.  Performance may depend on the choice of quantile levels and the Gaussian kernel bandwidth, which require tuning.

3. The sample complexity is bad i.e., $n^{-1/d}$ which scales exponentially with dimension.

4. The quantile-weighted features, though powerful, may obscure intuitive geometric interpretation compared to classical OT or GW frameworks.

**Questions:**

1. Does Integral-QDOT has good sample complexity as sliced optimal transport [1]?

2. How does QDOT compare theoretically to Sliced Gromov–Wasserstein (SGW) beyond computational complexity?

[1] An Introduction to Sliced Optimal Transport, Nguyen.

---

> ### Author Response · Authors · 2025-11-24
> **Response to Reviewer dUpr (Part 1 / 3)**
>
> Thank you for appreciating the rigorous metric definition, the handling of geometric transformations, and the method's generalization ability. We proceed to answer your questions and concerns below.
>
> - **On W1 (Concept and Computation of QW representations)**
>
>     **Intuition:**
>
>     We clarify the intuition behind QDOT as follows:
>     1.  Most metrics measuring distributional distance can be formulated as Equation (3) (Line 131). However, their representations often suffer from either information loss or a lack of isometric invariance, preventing them from being well-defined metrics.
>     2.  To address information loss, inspired by the **principle of trilateration**, we utilize distances to distinct anchors to characterize all distance information within the distribution.
>     3.  To ensure the isometric invariance of the anchors, we first derive an isometrically invariant distribution. By weighting this distribution at different quantile levels, we obtain a set of Quantile-weighted Distance Means (QDMs). These QDMs serve as our anchors, and the distances to these QDMs constitute our representation.
>     4.  Applying the Wasserstein distance to these representations yields our final QDOT distance, which aligns with the form of Equation (3).
>
>     **Complexity Analysis:**
>
>     For $n$ samples in $\mathbb{R}^d$ space with $k$ quantile levels, the computational complexity is analyzed as follows:
>     1.  Computing the norm of each sample: $O(nd)$.
>     2.  Computing $k$ quantiles of the norm distribution: $O(n\log n)$.
>     3.  Computing weights corresponding to $k$ quantiles: $O(nk)$.
>     4.  Computing $k$ QDMs: $O(ndk)$.
>     5.  Computing distances to $k$ QDMs: $O(ndk)$.
>
>     In summary, the computation of the representation is **(quasi-)linear** with respect to the sample size, dimensionality, and the number of quantiles, making it extremely easy to implement.
>
> - **On W2 (Tuning parameters)**
>
>     We acknowledge that the algorithm involves parameter tuning; however, our parameters possess clear physical interpretations, have low sensitivity, and are highly controllable in practical problems.
>
>     **Mechanism of Hyperparameters:**
>
>     1.  **The Bandwidth Parameter $\sigma$:** $\sigma$ relates to the weights used to solve the QDM. We perform Gaussian smoothing to obtain points in the vicinity of $F^{-1} _ {||X||}(q)$ and compute their mean to enhance robustness. If $\sigma$ is too small, weights become more **uniform**, causing the QDM to average over a large range of $F^{-1} _ {||X||}(q)$, leading to the collapse of QDMD into a polar coordinate distribution. If $\sigma$ is excessively large, weights **shrink to the single point** corresponding to $F^{-1} _ {||X||}(q)$, reducing robustness.
>     2.  **The Quantile Levels $q$:** $q$ determines the center of the weighted average. For some data, $q=1$ is suboptimal (e.g., a single outlier furthest from the center). Choosing $q$ close to 1 makes the result susceptible to extreme values. To avoid this, we generally constrain $q$ to be selected equidistantly within $[\delta, 1 - \delta]$. Furthermore, the selection of $q$ is directly related to the number of quantiles ($k$). Generally, **a larger number of quantiles yields more robust results**.
>
>     **Empirical Selection Strategy:**
>
>     1.  **For Bandwidth Parameter $\sigma$:** The selection generally relates to the number of support points ($n$). In Cross Space experiments ($n\sim 10^4$), we selected $\sigma = 1000$. In Transfer Learning experiments ($n\sim 10^3$), we selected $\sigma = 200$. In Molecule Generation experiments ($n\sim 30$), we selected $\sigma = 10$. Empirically, maintaining **$\sigma$ in the same order of magnitude as $n$** yields good results.
>     2.  **For Quantile Levels $q$:** For Cross Space and Molecule Generation experiments, since the position corresponding to $q=1$ varies frequently, we used a default $\delta = 0.1$ to avoid outliers. For Transfer Learning experiments, where distributions are static and classes are structurally similar, we set $\delta = 0$ to fully utilize all quantile information. For the number of quantiles, while "more is better," diminishing returns exist; we generally select between 30 and 100.

---

> ### Author Response · Authors · 2025-11-24
> **Response to Reviewer dUpr (Part 2 / 3)**
>
> - **On W2 (Tuning parameters)**
>
>     **Ablation Study:**
>
>     We demonstrate the impact of $\sigma$ and the number of quantiles $k$ via ablation studies on Cross-Space and Transfer Learning tasks.
>
>     1) **Cross Space Experiments:** We considered $\sigma \in \\{200, 1000, 5000\\}$ and $k \in \\{20, 100, 500\\}$. Results are shown below:
>
>     |Parameters|TMSE 3D|TMSE 2D_1st|TMSE 2D_2nd|TMSE 2D_3rd|TMSE Avg.|IR 3D (%)|IR 2D_1st (%)|IR 2D_2nd (%)|IR 2D_3rd (%)|IR Avg. (%)|
>     |-|-|-|-|-|-|-|-|-|-|-|
>     |σ=200, k=20|0.22|0.25|0.23|0.26|0.24|70.49|62.79|67.70|62.82|65.95|
>     |σ=200, k=100|0.22|0.24|0.23|0.25|0.24|70.82|63.34|68.05|63.09|66.33|
>     |σ=200, k=500|0.22|0.24|0.23|0.25|0.24|70.79|63.36|68.05|63.09|66.32|
>     |σ=1000, k=20|0.22|0.24|0.23|0.25|0.24|70.68|63.68|68.11|62.82|66.32|
>     |σ=1000, k=100|0.22|0.24|0.23|0.25|0.24|71.15|63.75|68.49|63.20|66.65|
>     |σ=1000, k=500|0.22|0.24|0.23|0.25|0.24|71.12|63.73|68.46|63.18|66.62|
>     |σ=5000, k=20|0.22|0.24|0.23|0.26|0.24|70.83|64.46|68.07|62.71|66.52|
>     |σ=5000, k=100|0.22|0.24|0.23|0.25|0.23|71.35|64.04|68.52|63.22|66.78|
>     |σ=5000, k=500|0.22|0.24|0.23|0.25|0.23|71.33|63.97|68.50|63.23|66.76|
>     |**Summary(mean±std)**|**0.22±0.00**|**0.24±0.01**|**0.23±0.00**|**0.25±0.01**|**0.24±0.01**|**70.95±0.28**|**63.68±0.45**|**68.22±0.27**|**63.04±0.19**|**66.47±0.25**|
>
>     As shown, the QDOT alignment results are not particularly sensitive to the selection of $\sigma$ and $k$. Generally, $k=100$ is sufficient, and increasing $k$ further yields minimal improvement.
>
>     2) **Transfer Learning Experiments:** We considered $n = 1024$, $\sigma \in \\{50, 200, 1000\\}$, $k \in \\{10, 50, 200\\}$.
>
>     |Params (Mo→Sh)|airplane|car|chair|guitar|lamp|laptop|table|Avg|
>     |-|-|-|-|-|-|-|-|-|
>     |σ=50,k=10|94.80|78.06|81.93|96.82|50.42|96.90|83.53|82.53|
>     |σ=50,k=50|95.46|82.74|82.70|96.44|50.81|96.67|84.50|83.46|
>     |σ=50,k=200|95.32|82.96|82.78|96.44|50.36|96.90|84.50|83.42|
>     |σ=200,k=10|94.94|80.07|83.10|97.46|50.10|96.45|84.03|83.12|
>     |σ=200,k=50|94.86|84.18|83.82|96.56|51.13|96.45|84.95|83.89|
>     |σ=200,k=200|95.65|84.41|84.11|96.32|50.23|97.12|85.30|84.15|
>     |σ=1000,k=10|94.98|80.73|82.01|97.71|48.48|95.57|82.68|82.26|
>     |σ=1000,k=50|95.35|84.08|84.78|95.93|50.81|97.12|84.90|84.14|
>     |σ=1000,k=200|95.72|85.63|84.73|96.19|50.55|96.90|85.18|84.37|
>     |**Summary(mean±std)**|**95.23±0.33**|**82.54±2.30**|**83.33±1.02**|**96.65±0.55**|**50.32±0.72**|**96.68±0.46**|**84.40±0.81**|**83.48±0.70**|
>
>
>     |Params (Sh→Mo)|airplane|car|chair|guitar|lamp|laptop|table|Avg|
>     |-|-|-|-|-|-|-|-|-|
>     |σ=50,k=10|98.62|83.84|89.79|97.65|79.17|95.27|94.92|92.58|
>     |σ=50,k=50|98.90|86.53|90.19|98.43|78.47|95.27|95.33|93.13|
>     |σ=50,k=200|98.76|88.89|90.09|98.82|79.86|95.86|95.73|93.49|
>     |σ=200,k=10|98.62|83.16|90.60|97.25|79.86|95.27|96.14|92.97|
>     |σ=200,k=50|98.89|87.20|91.10|98.43|81.94|97.04|96.74|93.97|
>     |σ=200,k=200|98.76|86.87|92.11|98.82|79.86|97.04|96.54|94.14|
>     |σ=1000,k=10|98.48|81.14|88.27|97.65|76.39|96.45|95.12|91.76|
>     |σ=1000,k=50|98.21|83.84|89.69|99.22|81.25|97.63|96.95|93.13|
>     |σ=1000,k=200|97.80|83.84|91.30|98.82|79.86|97.04|94.92|93.10|
>     |**Summary(mean±std)**|**98.56±0.34**|**85.03±2.31**|**90.35±1.04**|**98.34±0.63**|**79.63±1.50**|**96.32±0.87**|**95.82±0.75**|**93.14±0.67**|
>
>     Except for some numerical instability when $k=10$, the performance gaps are small, indicating that our method is not overly sensitive to hyperparameters.
>
>     In conclusion, **our parameters are practically meaningful, and defining an approximate range based on data structure yields strong performance.**

---

> ### Author Response · Authors · 2025-11-24
> **Response to Reviewer dUpr (Part 3 / 3)**
>
> - **On W3 (Convergence rate)**
>
>     Although our convergence rate depends on dimensionality ($O(n^{-1/d})$), this is a highly competitive result compared to other metrics. In most scenarios, the Wasserstein distance converges at $O(n^{-1/d})$ (Fournier et al., 2015), and the Gromov-Wasserstein distance converges at $O(n^{-2/\max(d, 4)})$ (Zhang et al., 2024), both suffering from the curse of dimensionality.
>
> - **On W4 (Geometric interpretation)**
>
>     QDOT also possesses a geometric interpretation: it compares distances to distinct, isometrically invariant anchors. Additionally, in the discrete case, our cost function is $c(x,y) = \sum_{i=1}^k|d_X(x, c^X_{q_i}) - d_Y(y, c^Y_{q_i})|$, where $c^X_{q_i}$ are isometric anchors. In contrast, the discrete Gromov-Wasserstein distance can be interpreted as a Wasserstein distance governed by the cost function $c(x,y) = \sum_{i=1}^n\sum_{j=1}^n\Pi_{ij}|d_X(x, x_i) - d_Y(y, y_j)|$, where the pairs $(x_i, y_j)$ are matched via the coupling $\Pi_{ij}$. This observation highlights that QDOT shares certain structural similarities with GW.
>
> - **On Convergence rate of IQDOT**
>
>     While a full investigation is currently ongoing, we observe that methods such as Sliced Wasserstein (SW) (Nadjahi et al., 2020), IPRW (Lin et al., 2021), and IPRHCP (Li et al., 2024) achieve dimension-independent convergence rates by projecting distributions onto lower-dimensional spaces. Given that IQDOT shares a similar projection-based mechanism, we are optimistic that its sample convergence rate is also independent of the ambient dimension. However, due to the theoretical intricacies involved, we plan to address the rigorous proof in future work.
>
> - **Comparison to Sliced GW (SGW)**
>
>     Beyond computational advantages, QDOT is an isometric metric under certain conditions, whereas SGW is only a pseudo-metric. Furthermore, QDOT possesses strict rotational invariance, which is difficult to guarantee for SGW. Additionally, without using sorting as an approximation, the 1D solution for SGW remains a non-convex problem; in contrast, the QDOT solution is strictly convex, resulting in greater robustness.
>
> **References:**
>
> (Fournier et al., 2015) Fournier, N., & Guillin, A. (2015). On the rate of convergence in Wasserstein distance of the empirical measure. Probability theory and related fields, 162(3), 707-738.
>
> (Zhang et al., 2024) Zhang, Z., Goldfeld, Z., Mroueh, Y., & Sriperumbudur, B. K. (2024). Gromov–Wasserstein distances: Entropic regularization, duality and sample complexity. The Annals of Statistics, 52(4), 1616-1645.
>
> (Genevay et al., 2019) Genevay, A., Chizat, L., Bach, F., Cuturi, M., & Peyré, G. (2019). Sample complexity of sinkhorn divergences. In The 22nd international conference on artificial intelligence and statistics (pp. 1574-1583). PMLR.
>
> (Nadjahi et al., 2020) Nadjahi, K., Durmus, A., Chizat, L., Kolouri, S., Shahrampour, S., & Simsekli, U. (2020). Statistical and topological properties of sliced probability divergences. Advances in Neural Information Processing Systems, 33, 20802-20812.
>
> (Lin et al., 2021) Lin, T., Zheng, Z., Chen, E., Cuturi, M., & Jordan, M. I. (2021). On projection robust optimal transport: Sample complexity and model misspecification. In International Conference on Artificial Intelligence and Statistics (pp. 262-270). PMLR.
>
> (Li et al., 2024) Li, T., Meng, C., Xu, H., & Yu, J. (2024). Hilbert curve projection distance for distribution comparison. IEEE Transactions on Pattern Analysis and Machine Intelligence, 46(7), 4993-5007.

---

> > ### Comment · Reviewer_dUpr · 2025-11-24
> >
> > Thank you very much for your response.
> >
> > On the sample complexity, I believe that $\mathcal{O}(n^{-1/d})$ is not a highly competitive result. For example, entropic approximation of Wasserstein/ Gromov Wasserstein has the complexity of  $\mathcal{O}(n^{-1/2}$ under assumption [1]. Sliced Wasserstein has the sample complexity of $\mathcal{O}(n^{-1/2})$ [2]. Is there any ways to improve the sample complexity for the proposed QDOT?
> >
> >
> >
> > [1] Gromov-Wasserstein Distances: Entropic Regularization, Duality, and Sample Complexity
> > [2] Statistical and Topological Properties of Sliced Probability Divergences

---

> ### Author Response · Authors · 2025-11-25
> **Response to Reviewer dUpr**
>
> We appreciate your timely reply and for pointing out the issue regarding the convergence rate of the Sinkhorn algorithms.
>
> When utilizing the Sinkhorn solver, following the argument in the proof of Theorem 2 (lines 1031–1048), we observe that the Entropic QDOT distance admits the following decomposition:
>
> $$
> \begin{aligned}
> \mathcal{QD} _ {\epsilon}(\mathcal{X}, \mathcal{Y})
> &= \inf _ {\pi} \int_{X \times Y} \\|\phi_q^X(x) - \phi_q^Y(y)\\| \\, d\pi(x,y)
>     + \epsilon \mathrm{KL}(\pi \mid \mu_X \otimes \mu_Y) \\\\
> &\le \inf _ {\pi} \int _ {X \times Y}C_1 d(x,y) + C_2 \sum _ {i=1}^k d(m_i^X, m_i^Y) \\, d\pi(x,y)
>     + \epsilon \mathrm{KL}(\pi \mid \mu_X \otimes \mu_Y) \\\\
> &\le C_2 \sum _ {i=1}^k d(m_i^X, m_i^Y)
>     + C_1 \inf _ {\pi} \int _ {X \times Y} d(x,y) \\, d\pi(x,y)
>     + \epsilon \mathrm{KL}(\pi \mid \mu_X \otimes \mu_Y) \\\\
> &= C_2 \sum _ {i=1}^k d(m_i^X, m_i^Y)
>     + C_1\mathcal{W} _ {\epsilon/C_1}(\mu_X, \mu_Y).
> \end{aligned}
> $$
>
> Here, the first term corresponds to the QDM discrepancy, for which we have $\mathbb{E}[\sum _ {i=1}^k d(m_i^X, m_i^{\mathbb{X}_n})] = O(n^{-1/2})$. The second term represents the Sinkhorn divergence. According to previous studies (Genevay et al., 2019; Zhang et al., 2024), for a sufficiently large $\epsilon$, we have $\mathbb{E}[\mathcal{W} _ {\epsilon/C_1}(\mu _ X, \mu _ {\mathbb{X}_n})] = O(n^{-1/2})$. Consequently, combining these results, we conclude that **the convergence rate of $\mathcal{QD} _ {\epsilon}$ is also $O(n^{-1/2})$.**
>
> Thanks again for highlighting this point, and we will incorporate this result into the updated version of the paper.
>
> **References:**
>
> (Genevay et al., 2019) Genevay, A., Chizat, L., Bach, F., Cuturi, M., & Peyré, G. (2019). Sample complexity of sinkhorn divergences. In The 22nd international conference on artificial intelligence and statistics (pp. 1574-1583). PMLR.
>
> (Zhang et al., 2024) Zhang, Z., Goldfeld, Z., Mroueh, Y., & Sriperumbudur, B. K. (2024). Gromov–Wasserstein distances: Entropic regularization, duality and sample complexity. The Annals of Statistics, 52(4), 1616-1645.

---

> ### Author Response · Authors · 2025-11-27
> **Gentle Reminder**
>
> Dear Reviewer dUpr,
>
> We hope this message finds you well. As we are now in the **final stretch of the discussion period**, we would like to verify if our updates, the supplementary experiments, and our last response have cleared up your concerns. If you have any additional thoughts or feedback, we would be happy to hear them. Your guidance is invaluable to us, and we strive to improve our work as much as possible before the deadline.
>
> Thank you for your thorough review and for your time.

---

> > ### Comment · Reviewer_dUpr · 2025-11-27
> >
> > Thank you very much for the reply,
> >
> > I wonder if you can summarize sample complexities and computational complexities of variants of QDOT  in a table. Which variant is recommended in practice?

---

> > > ### Author Response · Authors · 2025-11-27
> > > **Response to Reviewer dUpr**
> > >
> > > We appreciate your positive feedback. We summarize the computational complexity and sample convergence rates of the algorithms involved in this paper as follows:
> > >
> > > |Methods|Solver|Time Complexity|Sample Rate|
> > > |-|-|-|-|
> > > |QDOT|EMD|${O}(n^2\log n)$|${O}(n^{-1/d})$|
> > > |Entropic QDOT|Sinkhorn|${O}(n^2\log n)$|${O}(n^{-1/2})$|
> > > |IQDOT|Wasserstein 1D|${O}(n\log n)$|${O}(n^{-1/d})$ (at least)|
> > >
> > > In practical applications, we offer the following recommendations:
> > > *  For constructing a specific metric space (e.g., in our transfer learning experiments): We recommend using QDOT for small sample sizes and IQDOT for large sample sizes.
> > > *  For scenarios requiring a Monge map (e.g., in our cross-space experiments): We recommend using QDOT; For scenarios requiring detailed coupling information: We recommend using Entropic QDOT (EQDOT).
> > > *  For tasks involving complex network learning (e.g., in our molecular generation experiments): Due to the ease of gradient computation, we recommend using IQDOT.

---

### Official Review · Reviewer_kpCb · 2025-10-31

**Soundness:** 1
**Presentation:** 2
**Contribution:** 2
**Rating:** 2
**Confidence:** 4

**Summary:**

The authors propose the Quantile-Weighted Distance Metric for Geometric Comparison via Optimal Transport (QDOT) and Integral QDOT to address the structural alignment problem. This problem is closely related to the Gromov–Wasserstein problem, as both aim to match the geometric structures of two distributions that may differ in dimensionality. The proposed QDOT method first computes $k$ quantile means for a source and a target discrete distribution, then constructs cost matrices between the samples and the quantile means to finally use them to compute their Wasserstein distance and corresponding coupling matrix. To mitigate the computational limitations of QDOT during the Wasserstein distance computation, the authors introduce Integral QDOT, which replaces the full Wasserstein distance with the average of $d$ one-dimensional Wasserstein distances. Both methods are evaluated on two shape matching tasks (camell-galop and ModelNet40-ShapeNet) as well as a molecule generation task. The setups are evaluated using different metrics such as TMSE and accuracy.

**Strengths:**

- The structure and flow of the paper is good, this allows one to easily understand the proposed methods.
- The idea behind aligning quantiles instead of samples is novel and it seems it was not explored in the literature.

**Weaknesses:**

My main concerns about the paper are summarized below:
- The use of one-dimensional Wasserstein distance in integral QDOT is not well supported. As a result, Integral QDOT may fail to fully reflect the structural alignment between distributions as it does account inter dimensional relationship, this can potentially oversimplify complex dependencies. This would make the method not suitable for several, if not most, of the applications.
- There is a lack of discrete methods tested in the experimental section, methods such as (Alvarez, 2018), (Sebbouh, 2024) and (Klein, 2023) are also related and should be included to have a better understanding about the method’s performance and how it is positioned with respect to state-of-the-art approaches.
- In Subsection 4.1, it is unclear for me why TMSE and IR were chosen as metrics as they require a known transport map $\mathcal{T}$ (as pointed out in Appendix D.2). The details about the construction of this function are not explored in the paper and I think it is crucial to understand to which extent the reported metrics represent the alignment capabilities of the methods. More precisely, these metrics can give misleading results if $\mathcal{T}$ only ensures that points are close in space but not truly matched. In that case, even a poor coupling $\Pi$ could produce low TMSE or high IR scores without reflecting real alignment accuracy.
- In addition to the previous point, no ground truth alignment or true baseline is provided; therefore it is difficult to know if the values in Table 1 actually represent a good alignment or not. I am also not sure about how the 2D projections in this experiment were obtained and why they should be used to assess the alignment abilities of the methods.
- In addition to the previous point, no ground truth alignment or true baseline is provided; therefore it is difficult to know if the values in Table 1 actually represent a good alignment or not. I am also not sure about how the 2D projections in this experiment were obtained and why they should be used to assess the alignment abilities of the methods.
- I believe the plot with losses (Figure 4) is not informative and I think it would be better to replace it with a figure containing the visual results of the obtained alignments for this experiment.
- In subsection 4.3, the IQDOT method’s performance is not consistent. This is mainly seen in the experiments denoted as Mo$\rightarrow$Sh. As the rest of the experiments (4.1 and 4.4) do not include results for IQDOT, it is difficult to ensure that this approach is suitable for them. I raise this concern as the computation of one-dimensional Wasserstein distances to obtain the coupling may restrict its applicability (as previously mentioned).
- I have some concerns and observations regarding the experimental setup in subsection 4.4. From Table 3, it is difficult to interpret the obtained results as the metrics and the problem to solve are not properly introduced. Therefore, it is hard to say whether the obtained results represent a good alignment or not. It is important to clearly explain the metrics as they may be meaningful for different applications, but irrelevant in the case of structural alignment. Additionally, as the paper is oriented to a community mostly familiar with machine learning and artificial intelligence, I believe it is necessary to introduce the problem with more details as well as the computed metrics.
- I think the experimental section should also report the distortion and see if it correlates with the value of QDOT. I also suggest and I think it would also be interesting to report it after step 9 and also after step 10 in Algorithm 1, this would help to understand how impactful is the use of quantile alignment while finding the optimal coupling.
- I think the paper would benefit from including images of the obtained aligned point clouds in the case of experiments in 4.1, 4.2 and Appendix D.1. In this last case, only the noised distribution is shown and I think it is important to also add the source, target and the predicted points.
- The paper lacks experiments on uncorrelated setups, or at least, this point is never specified in the paper. Testing the methods on source and target distributions with unknown optimal pairs is an insightful way to show the performance and limitations of a GW-like solver that aims to capture and keep the geometric structure of the distributions. The concept of (un)correlatedness has been extensively explored in (Aramayo, 2025).

**Minor comments**
- In Table 1 it is unclear for me why the Entropic Gromov-Wasserstein method is slower than QDOT if both use the same Sinkhorn solver.
- No reference for Algorithm 2 in the main text.
- The notation for $\phi$ is currently confusing. In certain instances, it appears as $\phi_{\sharp\mu_X}^X$, but based on the definition in Equation 6, it is unclear how the pushforward operation is being applied.
- I believe the current notation for dimensions and number of samples is confusing and since it is used everywhere in the paper to understand the proposed ideas, I would suggest modifying it to improve the readability of the paper. One example about the importance of this appears in Definition 4 where $q$ is used to denote the quantiles while in 3.3, $q$ represents the dimensionality of the target distribution.
- From Algorithm 1, it is unclear how the optimal coupling $\Pi$ is computed, as the algorithm appears to only output the QDOT distance/metric. However, in Appendix D.2 (Lines 1234–1235), it is stated that the algorithm computes the coupling. Therefore, it is necessary to explicitly specify the coupling as an output in the algorithm to avoid ambiguity.

**To summarize,** I believe the paper has some interesting theoretical contributions regarding the novelty of the proposed QDOT method; however, the practical aspects of the paper require major revisions. More precisely, the reported metrics seem to be not the most adequate to show the true performance of the solver and the authors should also report more insightful metrics such as the GW distortion, SInkhorn divergence or MMD. On the other hand, the proposed IQDOT method seems to be weaker as it uses an average of one-dimensional Wasserstein distances, which does not consider inter dimensional relationships and it may not be useful for more complex tasks. For these reasons, I recommend the paper to be rejected.

(Alvarez, 2018) David Alvarez-Melis and Tommi Jaakkola. Gromov-wasserstein alignment of word embedding spaces. In Proceedings of the 2018 Conference on Empirical Methods in Natural Language Processing, pp. 1881–1890, 2018.

(Sebbouh, 2024) Othmane Sebbouh, Marco Cuturi, and Gabriel Peyre. Structured transforms across spaces with ´ cost-regularized optimal transport. In International Conference on Artificial Intelligence and Statistics, pp. 586–594. PMLR, 2024.

(Klein, 2023) Dominik Klein, Theo Uscidda, Fabian Theis, and Marco Cuturi. Generative entropic neural optimal ´ transport to map within and across spaces. arXiv preprint arXiv:2310.09254, 2023.

(Aramayo, 2025) Aramayo, X., Nekrashevich, M., Mokrov, P., Burnaev, E., & Korotin, A. (2025). Uncovering Challenges of Solving the Continuous Gromov-Wasserstein Problem.

**Questions:**

- In subsection 4.3, what are the dimensionalities of the source and the target distributions?
- How are the 2D projections obtained for the experiments in Subsection 4.1?
- Why are experiments with IQDOT not present in Table 1 and Table 3?
- In Algorithm 1, the norm $\|x\|_2$ represents the Euclidean norm? If yes, I would suggest keeping the same notation as in Equation 4, as it is confusing in the current state.
- Was the method tested on $q\ge d$? In other words, was it tested for low to high dimensional setups?
- Does the QDOT metric align with other metrics such as distortion from the GW problem?
- What loss is reported in Figure 4?

---

> ### Author Response · Authors · 2025-11-24
> **Clarifications**
>
> We thank the reviewer for the detailed comments.
>
> **Before addressing specific concerns, we would like to clarify three fundamental points regarding the scope and contribution of our work:**
>
> 1.  **A Metric is not just used for Alignment:** Our primary contribution lies in proposing an isometric invariant metric (Theorem 1, Theorem 3) and establishing its convergence rate (Theorem 2), rather than merely introducing an alignment method. The references suggested (Alvarez et al., 2018; Sebbouh et al., 2024; Klein et al., 2024; Aramayo et al., 2025) focus predominantly on alignment strategies without proposing specific metric properties or providing relevant metric-based experiments. We acknowledge the reviewer's expertise in shape matching and alignment; however, we respectfully request that attention also be directed toward our theoretical and experimental contributions regarding the **metric properties** of (I)QDOT.
>
> 2.  **The Role of IQDOT:** We respectfully disagree with the assessment that our IQDOT method is unsatisfactory simply because it may not function as a superior alignment tool. In fact, most widely adopted sliced methods (e.g., SW, Max-SW, SGW, SFGW) are utilized **not for their explicit alignment capabilities, but for their metric properties**, which facilitate the construction of effective metric spaces for various downstream tasks. In this work, we effectively apply IQDOT to both transfer learning and molecule generation tasks, where it achieves strong results. Given its rigorous metric properties (Theorem 3), computational efficiency (Sec. 4.2), and demonstrated empirical performance, we believe IQDOT merits a more favorable evaluation.
>
> 3.  **Suitability of Suggested References:** We have reviewed the four references provided for comparison and note the following:
>     *   (Alvarez et al., 2018) is an application paper of Gromov-Wasserstein, which serves as our main baseline.
>     *   The official implementation for (Sebbouh et al., 2024) is currently unavailable/deleted.
>     *   (Klein et al., 2024) proposes a supervised method within a Flow-matching framework, requiring additional training datasets, making it an unfair comparison for our unsupervised setting.
>     *   Regarding (Aramayo et al., 2025), we observe that this work shares the same title and similar content with `ICLR 2026 Conference Submission 7908`. We kindly ask the reviewer to clarify the propriety of citing this work.
>
>     Consequently, these methods are either already compared, difficult to implement, strictly supervised, or potentially concurrent work. We hope the reviewer considers these factors when suggesting comparisons.

---

> ### Author Response · Authors · 2025-11-24
> **Response to Reviewer kpCb (Part 1 / 2)**
>
> We now address the main concerns:
>
> - **On W1 (Applications of IQDOT)**
>
>     Crucially, IQDOT is proven to be a well-defined metric, which is a strong theoretical result. This property facilitates the construction of metric spaces for machine learning tasks. As shown in the camel trajectory experiment, the IQDOT distance accurately characterizes the four-period gait cycle of the camel. We also embed distributions from different spaces into a sample metric space using pairwise IQDOT distances, enabling cross-domain classification via KNN. Besides that, serving as a loss function, IQDOT enhances structural information and generalization in molecule generation, achieving SOTA results on multiple metrics.
>
> - **On W2 (More Comparisons)**
>
>     For the reasons mentioned in our opening remarks, we cannot provide comparisons to the specific suggested methods. However, we have supplemented our results with comparisons to established methods in the field, including PGW (Kerdoncuff et al., 2021), QGW (Chowdhury et al., 2021), and LRGW (Scetbon et al., 2022). The results are as follows:
>
>     |Methods|3D(TMSE)|2D₁(TMSE)|2D₂(TMSE)|2D₃(TMSE)|Avg.(TMSE)|3D(IR)|2D₁(IR)|2D₂(IR)|2D₃(IR)|Avg.(IR)|Time(×10²s)|
>     |-|-|-|-|-|-|-|-|-|-|-|-|
>     |PGW|0.35|0.34|0.32|0.33|0.33|53.85|55.38|55.71|51.35|54.07|0.42|
>     |QGW|0.56|0.62|0.53|0.64|0.59|23.42|18.30|27.98|17.74|21.85|**0.19**|
>     |LRGW|0.38|0.35|0.42|0.45|0.40|46.00|49.53|39.40|36.35|42.82|1.53|
>     |**QDOT**|**0.22**|**0.24**|**0.23**|**0.25**|**0.24**|**71.15**|**63.75**|**68.48**|**63.20**|**66.65**|0.74|
>
>     It is evident that the QDOT method achieves the best performance under the same order of computational cost, highlighting its efficiency.
>
> - **On W3-W5 (The Ground Truth)**
>
>     In fact, our experiments utilized the `camel-gallop` point cloud sequence and did not involve cross-category comparisons. Consequently, the ground truth is inherently defined by the temporal indices of the point cloud frames. We invite the reviewer to verify the dataset details, and we believe this clarification effectively resolves all of your confusion.
>
> - **On W6 & Q7 (The Loss Plot)**
>
>     To provide context: metrics such as OT and GW utilize a scalar to measure similarity between distributions. Therefore, benchmarking whether a *metric* works well is crucial. In this experiment, we plotted the metric sequence between a running camel and static 3D/2D camels. Since the camel runs for 4 cycles, a robust metric should clearly characterize these four periods. Figure 4 demonstrates that **both QDOT and IQDOT accurately capture this periodicity**, highlighting their capability for cross-space measurement. We hope the reviewer recognizes the significance of this result.
>
> - **On W7.1 (Transfer Learning Results)**
>
>     We reiterate that Transfer Learning is not an alignment result but a classification result obtained by constructing a metric space and applying 1-KNN. We are unsure where the reviewer perceives an inconsistency. If the reviewer refers to the performance drop in Mo$\rightarrow$Sh compared to Sh$\rightarrow$Mo, we consider this normal phenomenon, as the ShapeNet dataset has significantly more training samples (15,402) than ModelNet (3,072).
>
> - **On W7.2 (IQDOT Results)**
>     We apologize for the confusion. Due to computational burdens, the Molecule Generation task utilizes IQDOT as the penalty term. We have corrected this in the updated PDF. Consequently, the effectiveness of IQDOT is actually reported across Sections 4.1 to 4.4, and it was never intended to be evaluated primarily as an alignment method.
>
> - **On W8 (Molecular Generation Results)**
>
>     Our method calculates the metric between the noise and ground truth during the diffusion process and adds this as a penalty term to the generative model's objective. This enhances the original objective ($l_2$ norm) with geometric structure constraints and generalization capabilities.
>     Regarding evaluation metrics, we followed the settings of EDM (Hoogeboom et al., 2022), GeoLDM (Xu et al., 2023), and UniGEM (Feng et al., 2025). As described in UniGEM: "Following the approach of these baselines, we sample 10,000 molecules and evaluate atom stability, molecule stability, validity, and uniqueness...". These are standard metrics in the field.
>
> - **On W9 & Q6 (Distortion on QDOT)**
>
>     While distortion analysis is standard for alignment methods, we emphasize that QDOT is primarily proposed as a metric. Therefore, evaluating its coupling against external costs is not strictly necessary to validate its proposed metric properties. Nevertheless, we have supplemented Figure 4 with additional distortion results to address this concern.
>     We employed equidistant sampling to select approximately $n \approx 1000$ points.
>     As shown in Figure 10 (Lines 1355-1363), the QDOT loss and the GW loss maintain a high degree of consistency under the same coupling. This validates the effectiveness of QDOT as a metric.

---

> ### Author Response · Authors · 2025-11-24
> **Response to Reviewer kpCb (Part 2 / 2)**
>
> - **On W10 (Alignment Results)**
>
>     *   **Sec 4.1:** The alignment differences between (E)GW and QDOT are minor.
>     *   **Sec 4.2:** This is a time cost comparison; alignment visualization is not the focus.
>     *   **Sec 4.3:** Transfer learning involves specific distance values, not alignment.
>     *   **Appendix D.1:** This experiment demonstrates the empirical continuity and ablation of QDOT, unrelated to alignment.
>     For these reasons, we emphasize that providing extensive alignment results is not the focus of this paper. We urge the reviewer to distinguish between this work and pure alignment literature, and to appreciate the importance of metric properties.
>
> - **On W11 (Comparison to Potential ICLR 2026 Work)**
>
>     Due to the title and nature of the citation provided in the review, we are unable to perform this comparison.
>
> - **On W12 (EOT is faster than EGW)**
>
>     It is well established that EGW requires running the Entropic OT algorithm multiple times until the coupling converges, whereas QDOT only requires a single pass. Thus, EOT is naturally faster.
>
> - **On W13, W15 & W16 (Minor Typos)**
>
>     We thank the reviewer for pointing these out. They have been corrected.
>
> - **On W14 (Pushforward Operation)**
>
>     As is well known, the Wasserstein distance is defined on measures. The result of a pushforward operation is still a measure, and the definition of the pushforward operation is provided in Section 2 (Line 102).
>
> - **On Q1 (Dimension of Point Clouds)**
>
>     Both the source and target distributions are point clouds; therefore, the dimension is 3.
>
> - **On Q2 (Obtain 2D Projections)**
>
>     We first rotate the reference point cloud and then project it onto three orthogonal principal planes (XY, XZ, YZ) to obtain the 2D projections.
>
> - **On Q3 (IQDOT in Table 1 & 3)**
>
>     In Table 1, IQDOT is not used for alignment. For Table 3, please refer to our response to **W7.2**.
>
> - **On Q4 (Typos in Algorithm 1)**
>
>     Thank you for the correction; we have fixed these typos.
>
> - **On Q5 (High Dimensional Settings)**
>
>     In low dimensional scenarios, $q$ is generally set to be greater than $d$ to ensure robustness. For high-dimensional experiments, following (Li et al., 2024), we added a high-dimensional experiment in the Appendix D.4 (Lines 1476-1500). The results demonstrate that our method maintains good stability on high-dimensional steeings.
>
> - **On Q6:** Please refer to our response to **W9**.
> - **On Q7:** Please refer to our response to **W6**.
>
> In summary, we hope the above response effectively addresses the critical issues you raised and helps you re-evaluate the contribution of our work. Please let us know if any clarifications are still needed.
>
> **References:**
>
> (Alvarez et al., 2018) David Alvarez-Melis and Tommi Jaakkola. Gromov-wasserstein alignment of word embedding spaces. In Proceedings of the 2018 Conference on Empirical Methods in Natural Language Processing.
>
> (Sebbouh et al., 2024) Othmane Sebbouh, Marco Cuturi, and Gabriel Peyre. Structured transforms across spaces with cost-regularized optimal transport. In International Conference on Artificial Intelligence and Statistics, pp. 586–594. PMLR, 2024.
>
> (Klein et al., 2024) Klein, D., Uscidda, T., Theis, F., & Cuturi, M. (2024). GENOT: Entropic (Gromov) Wasserstein flow matching with applications to single-cell genomics. Advances in Neural Information Processing Systems.
>
> (Aramayo et al., 2025) Aramayo, X., Nekrashevich, M., Mokrov, P., Burnaev, E., & Korotin, A. (2025). Uncovering Challenges of Solving the Continuous Gromov-Wasserstein Problem.
>
> (Kerdoncuff et al., 2021) Kerdoncuff, T., Emonet, R., & Sebban, M. (2021). Sampled gromov wasserstein. Machine Learning, 110(8), 2151-2186.
>
> (Chowdhury et al., 2021) Chowdhury, S., Miller, D., & Needham, T. (2021). Quantized gromov-wasserstein. In Joint European Conference on Machine Learning and Knowledge Discovery in Databases. Cham: Springer International Publishing.
>
> (Scetbon et al., 2022) Scetbon, M., Peyré, G., & Cuturi, M. (2022). Linear-time gromov wasserstein distances using low rank couplings and costs. In International Conference on Machine Learning. PMLR.
>
> (Hoogeboom et al., 2022) Hoogeboom, E., Satorras, V. G., Vignac, C., & Welling, M. (2022). Equivariant diffusion for molecule generation in 3d. In International conference on machine learning. PMLR.
>
> (Xu et al., 2023) Xu, M., Powers, A. S., Dror, R. O., Ermon, S., & Leskovec, J. (2023). Geometric latent diffusion models for 3d molecule generation. In International Conference on Machine Learning (pp. 38592-38610). PMLR.
>
> (Feng et al., 2025) Feng, S., Ni, Y., Ma, Z. M., Ma, W. Y., & Lan, Y. (2025). UniGEM: A Unified Approach to Generation and Property Prediction for Molecules. In The Thirteenth International Conference on Learning Representations.
>
> (Li et al., 2024) Li, T., Meng, C., Xu, H., & Yu, J. (2024). Hilbert curve projection distance for distribution comparison. IEEE Transactions on Pattern Analysis and Machine Intelligence.

---

### Official Review · Reviewer_r88Z · 2025-10-31

**Soundness:** 3
**Presentation:** 3
**Contribution:** 3
**Rating:** 6
**Confidence:** 4

**Summary:**

This paper introduces QDOT (Quantile-weighted Distance Optimal Transport), a new metric for geometric comparison of data distributions, particularly across heterogeneous metric spaces. The method constructs isometry-invariant representations by leveraging quantile-weighted distance means (QDMs), then computes Wasserstein distances between transformed representations. This approach preserves intrinsic geometric information while achieving favorable theoretical and computational properties. The authors prove that QDOT and its integral variant (IQDOT) are valid metrics on the space of isometry classes of metric measure spaces, with empirical convergence rates no worse than Wasserstein distance, and O(n log n) computational complexity. Experiments on cross-space alignment, transfer learning, and molecular generation demonstrate strong performance and practical relevance.

**Strengths:**

1). Strong conceptual novelty and theoretical grounding. The introduction of quantile-weighted distances as canonical anchors represents an elegant and original solution to the long-standing trade-off between isometry invariance and information preservation in geometric metrics. The theoretical results (Theorems 1–3) are rigorous, clearly presented, and address fundamental properties such as identity, symmetry, triangle inequality, and convergence.

2). Efficient and scalable computation. The proposed IQDOT variant achieves quasi-linear complexity (O(n log n)), which is a substantial improvement over the cubic or quadratic complexity of Gromov–Wasserstein and related metrics. The empirical runtime results confirm this advantage convincingly.

3). Comprehensive and meaningful experiments. In cross-space alignment (camel-gallop dataset), QDOT matches or surpasses GW and EGW while being over 30× faster.

**Weaknesses:**

1). While the quantile-based weighting is conceptually appealing, additional intuition or ablation on the choice of quantile levels and bandwidth σ would strengthen the understanding of sensitivity.

2). The theoretical section could briefly relate QDOT to classical results in empirical OT convergence (e.g., emphasizing where quantile weighting introduces regularization).

3). Molecular generation experiments could include visual examples to illustrate the qualitative improvements achieved with QDOT.

**Questions:**

See the weaknesses.

---

> ### Author Response · Authors · 2025-11-24
> **Response to Reviewer r88Z (Part 1 / 2)**
>
> We are grateful for your recognition of the strong theoretical grounding, conceptual novelty, and the scalability of our proposed metric. Please see our detailed response to your comments below.
>
> - **On W1: Intuition and Ablation on parameters**
>
>     First, we elucidate the mechanism of the bandwidth and quantile levels and their impact on the outputs.
>
>     1.  **The Bandwidth Parameter $\sigma$:** This parameter governs the weights used to compute the Quantile-weighted Distance Means (QDMs). We employ Gaussian smoothing to obtain the mean of points in the vicinity of $F^{-1} _ {||X||}(q)$, thereby enhancing algorithm robustness. If $\sigma$ is too small, the weights become nearly **uniform**. This causes the QDM to average over a large range of $F^{-1} _ {||X||}(q)$, leading the QDMD to collapse into a polar coordinate distribution. If $\sigma$ is excessively large, the weights **contract to the single point** corresponding to $F^{-1} _ {||X||}(q)$, rendering the algorithm less robust.
>     2.  **The Quantile Levels $q$:** This determines the points around which the weighted average is performed. For certain datasets, choosing $q=1$ is suboptimal; for instance, if a single outlier is furthest from the center, a $q$ value close to 1 will be heavily influenced by this extreme value. To mitigate this, we generally constrain $q$ to be selected equidistantly within the interval $[\delta, 1 - \delta]$. Additionally, the selection of $q$ is directly related to the number of quantiles $k$; generally, **a larger number of quantiles yields more robust results**.
>
>     **Empirical Parameter Selection:**
>
>     1.  **Bandwidth Parameter $\sigma$:** The selection of $\sigma$ typically correlates with the number of support points $n$. In the Cross Space experiments ($n\sim 10^4$), we selected $\sigma = 1000$. In the Transfer Learning experiments ($n\sim 10^3$), we selected $\sigma = 200$. In the Molecule Generation experiments ($n\sim 30$), we selected $\sigma = 10$. Empirically, we find that maintaining $\sigma$ in the same order of magnitude as $n$ yields effective results.
>     2.  **Quantile Levels $q$:** For Cross Space and Molecule Generation experiments, since the position corresponding to $q=1$ varies frequently (different distributions often correspond to different points), we used a default $\delta = 0.1$ to avoid the influence of outliers. For Transfer Learning experiments, as the distributions are static and classes are similar, we set $\delta = 0$ to fully utilize information from all quantiles. Regarding the number of quantiles, while "more is better," the marginal benefit decreases. We generally select between 30 and 100.
>
>     **Ablation Study Results:**
>
>     We present ablation studies on Cross-Space and Transfer Learning tasks to demonstrate the impact of $\sigma$ and the number of quantiles $k$.
>
>     1) **Cross Space Experiments:** We considered $\sigma \in \{200, 1000, 5000\}$ and $k \in \{20, 100, 500\}$. The results are as follows:
>
>     |Parameters|TMSE 3D|TMSE 2D_1st|TMSE 2D_2nd|TMSE 2D_3rd|TMSE Avg.|IR 3D (%)|IR 2D_1st (%)|IR 2D_2nd (%)|IR 2D_3rd (%)|IR Avg. (%)|
>     |-|-|-|-|-|-|-|-|-|-|-|
>     |σ=200, k=20|0.22|0.25|0.23|0.26|0.24|70.49|62.79|67.70|62.82|65.95|
>     |σ=200, k=100|0.22|0.24|0.23|0.25|0.24|70.82|63.34|68.05|63.09|66.33|
>     |σ=200, k=500|0.22|0.24|0.23|0.25|0.24|70.79|63.36|68.05|63.09|66.32|
>     |σ=1000, k=20|0.22|0.24|0.23|0.25|0.24|70.68|63.68|68.11|62.82|66.32|
>     |σ=1000, k=100|0.22|0.24|0.23|0.25|0.24|71.15|63.75|68.49|63.20|66.65|
>     |σ=1000, k=500|0.22|0.24|0.23|0.25|0.24|71.12|63.73|68.46|63.18|66.62|
>     |σ=5000, k=20|0.22|0.24|0.23|0.26|0.24|70.83|64.46|68.07|62.71|66.52|
>     |σ=5000, k=100|0.22|0.24|0.23|0.25|0.23|71.35|64.04|68.52|63.22|66.78|
>     |σ=5000, k=500|0.22|0.24|0.23|0.25|0.23|71.33|63.97|68.50|63.23|66.76|
>     |**Summary(mean±std)**|**0.22±0.00**|**0.24±0.01**|**0.23±0.00**|**0.25±0.01**|**0.24±0.01**|**70.95±0.28**|**63.68±0.45**|**68.22±0.27**|**63.04±0.19**|**66.47±0.25**|
>
>     As shown in the table, the QDOT alignment results are not particularly sensitive to the selection of $\sigma$ and $k$. Furthermore, $k=100$ generally provides sufficient performance, with minimal additional gain from larger $k$.

---

> ### Author Response · Authors · 2025-11-24
> **Response to Reviewer r88Z (Part 2 / 2)**
>
> - **On W1: Intuition and Ablation on parameters**
>     2. **Transfer Learning Experiments:** We considered $n = 1024$, $\sigma \in \{50, 200, 1000\}$, and $k \in \{10, 50, 200\}$. The results are as follows:
>
>     |Params (Mo→Sh)|airplane|car|chair|guitar|lamp|laptop|table|Avg|
>     |-|-|-|-|-|-|-|-|-|
>     |σ=50,k=10|94.80|78.06|81.93|96.82|50.42|96.90|83.53|82.53|
>     |σ=50,k=50|95.46|82.74|82.70|96.44|50.81|96.67|84.50|83.46|
>     |σ=50,k=200|95.32|82.96|82.78|96.44|50.36|96.90|84.50|83.42|
>     |σ=200,k=10|94.94|80.07|83.10|97.46|50.10|96.45|84.03|83.12|
>     |σ=200,k=50|94.86|84.18|83.82|96.56|51.13|96.45|84.95|83.89|
>     |σ=200,k=200|95.65|84.41|84.11|96.32|50.23|97.12|85.30|84.15|
>     |σ=1000,k=10|94.98|80.73|82.01|97.71|48.48|95.57|82.68|82.26|
>     |σ=1000,k=50|95.35|84.08|84.78|95.93|50.81|97.12|84.90|84.14|
>     |σ=1000,k=200|95.72|85.63|84.73|96.19|50.55|96.90|85.18|84.37|
>     |**Summary(mean±std)**|**95.23±0.33**|**82.54±2.30**|**83.33±1.02**|**96.65±0.55**|**50.32±0.72**|**96.68±0.46**|**84.40±0.81**|**83.48±0.70**|
>
>     |Params (Sh→Mo)|airplane|car|chair|guitar|lamp|laptop|table|Avg|
>     |-|-|-|-|-|-|-|-|-|
>     |σ=50,k=10|98.62|83.84|89.79|97.65|79.17|95.27|94.92|92.58|
>     |σ=50,k=50|98.90|86.53|90.19|98.43|78.47|95.27|95.33|93.13|
>     |σ=50,k=200|98.76|88.89|90.09|98.82|79.86|95.86|95.73|93.49|
>     |σ=200,k=10|98.62|83.16|90.60|97.25|79.86|95.27|96.14|92.97|
>     |σ=200,k=50|98.89|87.20|91.10|98.43|81.94|97.04|96.74|93.97|
>     |σ=200,k=200|98.76|86.87|92.11|98.82|79.86|97.04|96.54|94.14|
>     |σ=1000,k=10|98.48|81.14|88.27|97.65|76.39|96.45|95.12|91.76|
>     |σ=1000,k=50|98.21|83.84|89.69|99.22|81.25|97.63|96.95|93.13|
>     |σ=1000,k=200|97.80|83.84|91.30|98.82|79.86|97.04|94.92|93.10|
>     |**Summary(mean±std)**|**98.56±0.34**|**85.03±2.31**|**90.35±1.04**|**98.34±0.63**|**79.63±1.50**|**96.32±0.87**|**95.82±0.75**|**93.14±0.67**|
>
>     As indicated in the tables, apart from some numerical instability when $k=10$, the differences between results are not substantial. This demonstrates that our method is relatively insensitive to parameter variations.
>
>     In summary, we believe **our parameters possess practical significance and low sensitivity. Determining an approximate range based on the data structure is sufficient to achieve strong performance**.
>
> - **On W2: Convergence Rate**
>
>     In our proof (Lines 1030-1048), we decompose the QDOT discrepancy into the Wasserstein discrepancy and the QDM discrepancy. We observe that the convergence rate of the QDM discrepancy is $O(n^{-1/2})$, which is equal to or faster than the Wasserstein discrepancy in any scenario. Consequently, the overall convergence rate of QDOT remains consistent with that of the Wasserstein distance. We acknowledge that our use of the phrase "at least as fast as" in Theorem 2 (Lines 249) may have been confusing; we apologize for this and have rectified the phrasing in the main text.
>
> - **On W3: Visual examples**
>
>     We have added visual examples of samples generated by the IQDOT-regularized model in the Appendix D.5 (Page 30). It can be observed that the IQDOT-regularized model tends to preserve stronger structural information and exhibits greater stability.
>
> In summary, we trust that the above response clarifies your concerns successfully and encourages you to support our paper in the upcoming discussion and decision phases. Thank you for your time.

---

> ### Author Response · Authors · 2025-11-27
> **Gentle Reminder**
>
> Dear Reviewer r88Z,
>
> We hope this message finds you well. Since we are entering the **final week** of the discussion phase, we are writing to inquire if the modifications in our revision and the added experiments have met your expectations. We welcome any further advice or feedback you might have to help us improve the manuscript. Your insights have been very helpful to us, and we are keen to make any necessary improvements based on your suggestions.
>
> Thank you very much for your hard work and review.

---

### Official Review · Reviewer_dG4E · 2025-11-01

**Soundness:** 2
**Presentation:** 3
**Contribution:** 2
**Rating:** 4
**Confidence:** 3

**Summary:**

The paper proposes QDOT - a Quantile‑weighted Distance Optimal Transport metric for comparing probability distributions defined in (potentially different) Euclidean spaces. The core idea is to build isometry‑invariant anchors called Quantile‑weighted Distance Means (QDMs) by weighting points with a Gaussian centered at quantiles of the norm distribution; distances from points to these anchors produce a representation (QDMD) on which a standard Wasserstein distance is computed. An “integral” variant (IQDOT) aggregates 1‑D Wasserstein distances across quantiles to achieve quasi‑linear time. The paper proves that (under dimensionality conditions on the anchor set) both QDOT and IQDOT are metrics on isometry classes and that empirical convergence is at least as fast as the classical Wasserstein rate $n^{-\frac{1}{d}}$. Experiments cover cross‑space point‑cloud alignment, point‑cloud transfer learning and molecule generation where QDOT is used as a loss in diffusion models. Experimental results show comparable or improved alignment quality over Gromov Wassesrein and Entropic Gromov Wassesrstin with orders of magnitude speedups, quasi‑linear scaling for IQDOT, strong transfer accuracy and improved stability/validity for molecular generation.

**Strengths:**

The primary methodology of the paper is based on defining anchors by norm‑quantiles and building representations from distances to those anchors, which is conceptually simple and gives immediate invariance to orthogonal transforms. The metric properties together with rotational invariance of QDOT and IQDOT, under centering assumptions, make sense to me, together with the fact that the distances retain the Wasserstein convergence rates .The computational complexity is perhaps the most practically relevant and useful aspect of the proposed metrics.

**Weaknesses:**

1. The metric property of QDOT relies on the dimensionaility condition presented in Theorem 1 (Lines 223-224), and the a similar scenario is true for IQDOT based on Theorem 3. For any centered, radially (or centrally) symmetric distribution (for e.g., an isotropic Gaussian distribution ) one has that $\mathbb{E}(f(\||X\||_{2})X)$, and hence all QDMs, as defined in Equation 5 (Lines 195-195) will trivially reduce to the same value 0. In that case, Equation 6 shows that $\phi^{X}(x,q)=\||x\||_2$ for any $q \in (0,1)$, so QDOT compares only the radial norm distributions and can identify non‑isometric spaces with identical norm distributions, violating identity of indiscernibles without the assumption that all distributions are first centered around the origin. So the centering assumption around 0 is very crucial.

2. Corollary 1 claims location invariance (Lines 252-253), yet the representation explicitly includes $\varphi_{0}(x) = \||x\||_{2}$, which by itself is not translation‑invariant. The paper suggests centering as "recommended” pre‑processing (Appendix C.1 Lines 1105-1106), but then the corollary does not make this centering explicit. The paper should either: (i) prove translation invariance with $\varphi_0$ and without pre‑centering, or (ii) make centering a required step in the definition/algorithms and restate invariance accordingly.

**Questions:**

Please see the Weaknesses section.

---

> ### Author Response · Authors · 2025-11-24
> **Response to Reviewer dG4E**
>
> We thank you for highlighting the conceptual simplicity, rotational invariance, and the practical computational complexity of our method. We have provided a detailed clarification regarding your points below.
>
> - **On W1.1: Intuition of Centering**
>
>     Consider two mm-spaces $\mathcal{X}=(X,d_X, \mu_X)$ and $\mathcal{Y}=(Y,d_Y, \mu_Y)$. To obtain a family of isometry-invariant QDMs, we first aim to construct an isometry-invariant distribution. A natural intuition is to identify an isometric pair $(x_0, y_0)$ such that if $\mathcal{X}$ and $\mathcal{Y}$ are isometric via an isometry $f:X \rightarrow Y$, the correspondence $f(x_0) = y_0$ holds (implying that $d_X(\cdot, x_0)$ and $d_Y(\cdot, y_0)$ share the same distribution).
>
>     To find such a matched pair $(x_0, y_0)$, we identify the **Barycenter** of the metric space as an optimal candidate. The Barycenter of $\mathcal{X}$ is defined as:
>     $$
>     b_X =  \operatorname{argmin} _ {x'\in X} \mathbb{E}_{\mu_X}(d_X^2(X, x'))
>     $$
>     Similarly, let $b_Y$ be the Barycenter of $\mathcal{Y}$. The Barycenter possesses two desirable properties:
>     1.  **Isometry Invariance:** If $\mathcal{X}$ and $\mathcal{Y}$ are isometric with mapping $f:X \rightarrow Y$, then $b_Y = f(b_X)$. This holds because both minimize the expectation of the squared distance, and the property $d_X(x_0, x_1) = d_Y(f(x_0), f(x_1))$ is guaranteed by the isometry $f$.
>     2.  **Euclidean Correspondence:** In Euclidean space, the Barycenter corresponds exactly to the distribution mean: $b_X = \mathbb{E}_{\mu_X}(X)$.
>
>     Based on these properties, we require a set of matched points $(x_0, y_0)$, and the Barycenter serves as this naturally matched pair. In Euclidean space, since this corresponds to the mean, we implement this as a centering operation. Theoretically, our method **works with any pre-matched points $(x_0, y_0)$**. We have clarified this intuition at the beginning of Section 3 in the revised manuscript.
>
> - **On W1.2: How to deal with symmetric distributions**
>
>     We propose a specific strategy to handle symmetric distributions using QDOT. The core idea is to perform comparison on the **minimal unique "piece"** of a regular shape (inspired by Takeda et al., 2025).
>
>     As illustrated in Figure 7 (Lines 1164-1180), we observe that for simple 2D shapes such as squares, rectangles, and circles, the Quantile-weighted Distance Means (QDMs) is located at the origin ($\dim = 0$). Since these are axially symmetric, we split them along the origin (upper/lower split).
>     Upon computing the QDM for the split shapes, we find they lie on the y-axis ($\dim = 1$), implying they remain symmetric about the y-axis. After a second split, the QDM of the rectangle is no longer collinear, meaning its $\dim = 2$, thus satisfying our condition. Similarly, the square satisfies the condition after additional split. In contrast, a circle appears to allow infinite splitting, with its $\dim$(QDM) always remaining 1.
>
>     We now describe how QDOT couples with these shapes:
>     1.  **If both shapes are 4-Symmetric:** After two splits, the QDMs of both shapes satisfy the condition. In this case, alignment can be performed successfully, and the metric properties are well-preserved.
>     2.  **If one shape is 4-Symmetric, but the other is 8-Symmetric or higher (e.g., $\infty$-Symmetric):** The alignment result may fail because the pieces on symmetric sides result in the same QDMD embeddings, leading to unrecognizability. However, the **metric property remains valid**. This is because the condition only requires one shape to satisfy the rank requirement (note that our condition is $dim(m^X)=dim(X)$ **or** $dim(m^Y)= dim(Y)$). Since the QDM dimension of the first shape becomes 2 after two cuts, QDOT remains a well-defined metric.
>     3.  **If both shapes are $\infty$-Symmetric:** This implies both are radially symmetric distributions. In this case, the QDOT comparison may directly reduce to comparing their radial distributions. Since **the radial distribution encapsulates all information for radially symmetric distributions**, QDOT remains a well-defined metric.
>
>     Through this approach, QDOT is capable of handling a range of symmetric and radially symmetric distributions.
>
> - **On W2: Translation Invariance**
>
>     We have incorporated our intuition regarding centering into the beginning of Section 3 (Lines 161-175); please refer to our response to **W1.1** for details. Furthermore, we have revised the wording in the Appendix C.1 (Lines 1188-1194) to clarify that for two unknown distributions situated in Euclidean spaces, centering is necessary.
>
> In summary, we hope the above response clears up your concerns and helps you reconsider your evaluation of our work. Please feel free to contact us if you have any further questions.
>
> (Takeda et al., 2025) Takeda, S., & Akagi, Y. (2025). Gromov-Wasserstein Problem with Cyclic Symmetry. In Proceedings of the Computer Vision and Pattern Recognition Conference.

---

> ### Author Response · Authors · 2025-11-27
> **Gentle Reminder**
>
> Dear Reviewer dG4E,
>
> We hope this message finds you well. With the discussion phase **approaching its final days**, we would appreciate it if you could confirm whether our response and the revised paper have satisfactorily addressed your comments. If there are any remaining points you wish for us to consider, please feel free to share them. Your expertise is highly appreciated, and we are eager to address any outstanding issues to strengthen our work.
>
> Many thanks for your time and contribution to improving our paper.

---

### Official Review · Reviewer_HU2b · 2025-11-02

**Soundness:** 3
**Presentation:** 2
**Contribution:** 3
**Rating:** 6
**Confidence:** 4

**Summary:**

This paper introduces QDOT (Quantile-weighted Distance Optimal Transport), a novel metric for comparing probability distributions across heterogeneous metric spaces. The key idea is to construct isometry-invariant anchor points (QDMs) using quantile-weighted distance means, then represent each distribution by distances to these anchors, enabling comparison via standard optimal transport. The authors prove QDOT satisfies metric properties on isometry classes and achieves convergence rates comparable to Wasserstein distance. An efficient integral variant (IQDOT) reduces complexity to O(n log n). Experiments demonstrate strong performance on cross-space alignment, transfer learning, and molecular generation tasks.

**Strengths:**

1. The paper provides rigorous proofs that QDOT is a well-defined metric on isometry classes (Theorem 1) and achieves favorable convergence rates (Theorem 2). The integral variant achieves O(n log n) complexity, making it significantly more practical than GW-based methods while maintaining theoretical guarantees, a rare combination in this space.

2. The quantile-weighted distance mean construction is elegant and well-motivated by trilateration principles. Unlike prior work, QDOT explicitly constructs the shared representation space rather than relying on implicit embeddings, addressing a fundamental limitation of existing geometric comparison methods.

3. The experiments span multiple domains (point cloud alignment, transfer learning, molecule generation) and consistently demonstrate both efficiency gains and competitive or superior performance. The molecular generation results are particularly impressive, achieving new state-of-the-art stability metrics while accelerating training convergence.

**Weaknesses:**

1. The requirement that QDMs span the full dimension is crucial for the metric property but receives insufficient treatment. When might this fail in practice? How sensitive is the method to near-degenerate cases? The paper claims this is "generally satisfied" but provides no empirical analysis or failure cases.

2.  The choice of quantile levels (the vector q) and bandwidth parameter σ appear critical but are not thoroughly investigated. How should practitioners select these? The experiments use different settings across tasks without clear justification, and there's no sensitivity analysis to understand robustness.

3. In the transfer learning experiments, only SGW is compared due to computational constraints, but recent efficient baselines like sliced methods are missing. The molecule generation ablation with MSE-reweighting is helpful but doesn't fully isolate QDOT's geometric properties from other potential effects.

4. While both are proven to be metrics, the relationship between them is unclear. Does IQDOT approximate QDOT, or are they fundamentally different? The paper would benefit from bounds relating the two or empirical analysis of when they agree/disagree.

5. The transition from intuition (trilateration) to formal definition could be smoother. Section 3.1's centering assumption is introduced casually but affects the entire framework. Some notation is inconsistent (e.g., switching between X and X for the space vs. sample matrix).

**Questions:**

1. Can you provide guidance on hyperparameter selection? Specifically, how should the number of quantiles k, their positions q, and bandwidth σ be chosen for new applications? Are there theoretical principles or heuristics that practitioners can follow?

2. How does performance degrade when the dimensionality condition is violated? It would be valuable to see experiments with distributions where the QDMs don't span the full space, to understand the practical robustness of the method and whether approximate metric properties still hold.

---

> ### Author Response · Authors · 2025-11-24
> **Response to Reviewer HU2b (Part 1 / 4)**
>
> We appreciate your positive feedback on our theoretical results, the construction of the integral variant, and the performance of our method in molecular generation. Our responses to your specific questions are detailed below.
>
> - **On W1 & Q2 (Dimensional Condition)**
>
>     The condition is inspired by the principle of trilateration. It is easy to find that when a distribution exhibits regular structures (e.g., symmetry, rotational invariance), the condition may not be satisfied.
>
>     **Our Strategy:**
>
>     To address symmetric structures, we propose a strategy based on using the **minimal unique "piece"** of the regular shape for coupling (similar to Takeda et al., 2025).
>     As illustrated in Figure 7 in Appendix B.4 (Lines 1162-1180), we observe that for simple 2D shapes such as squares, rectangles, and circles, their Quantile-weighted Distance Means (QDMs) are located at the origin ($\dim = 0$). Since these are axially symmetric, we consider splitting them along the origin (upper/lower split).
>     Upon computing the QDM for the split shapes, we find they lie on the y-axis ($\dim = 1$), implying they remain symmetric about the y-axis. After a second split, the QDM of the rectangle is no longer collinear, meaning its $\dim = 2$, thus satisfying our condition. Similarly, the square satisfies the condition after the third split. In contrast, a circle appears to allow infinite splitting, with its $\dim$(QDM) always remaining 1.
>
>     **Coupling Analysis:**
>
>     We now explain how QDOT couples with these shapes:
>     1.  **Both shapes are 4-Symmetric:** After two splits, the QDMs of both shapes satisfy the condition. In this case, alignment can be performed, and metric properties are well-preserved.
>     2.  **One is 4-Symmetric, the other is 8-Symmetric or higher (e.g., $\infty$-Symmetric):** The alignment result fails because the pieces on symmetric sides result in the same QDMD embeddings, making them unrecognizable. However, the **metric property still holds**. This is because the condition only requires one distribution to satisfy the requirement (note that our condition is $\dim(m^X)=\dim(X)$ **or** $\dim(m^Y)= \dim(Y)$). Since the dimension of the QDM for the first shape becomes 2 after two cuts, QDOT remains a well-defined metric.
>     3.  **Both are $\infty$-Symmetric:** This implies both shapes are radially symmetric distributions. In this case, the QDOT comparison may directly reduce to comparing their radial distributions, which is also a well-defined metric.
>     4. Other cases (such as rotational symmetry) can be handled using the same logic. As regular shapes did not appear in our experiments, we do not expand on them further. While there must be some extreme distributions that do not satisfy the condition, we consider them generally rare and exclude them from discussion.
>
>     After analyzing these cases, we believe our conditions are generally satisfied in practical scenarios, particularly in our experiments involving low-dimensional real data.
>
> - **On W2 & Q1 (Hyperparameter Selection)**
>
>     First, we explain the mechanism of the hyperparameters and their effect on the outputs:
>
>     1.  **The Bandwidth Parameter $\sigma$:** $\sigma$ is related to the weights used in solving the QDM. We employ Gaussian smoothing to obtain points near $F^{-1} _ {||X||}(q)$ and compute their mean to enhance algorithm robustness. If $\sigma$ is too small, the weights become more **uniform**, causing the obtained QDM to average over a large range of $F^{-1} _ {||X||}(q)$, which leads to the QDMD collapsing into a polar coordinate distribution. If $\sigma$ is extremely large, the weights **shrink to a single point** corresponding to $F^{-1} _ {||X||}(q)$, reducing the robustness of the algorithm.
>     2.  **The Quantile Levels $q$:** $q$ determines the points around which we perform the weighted average. For some data, $q=1$ is not an optimal choice. For instance, if there is only one outlier point furthest from the center, selecting a $q$ close to 1 makes the result susceptible to this extreme value. To avoid this, we generally constrain $q$ to be selected equidistantly within $[\delta, 1 - \delta]$. Furthermore, the selection of $q$ is directly related to the number of quantiles; generally, **a higher number of quantiles yields more robust results**.

---

> ### Author Response · Authors · 2025-11-24
> **Response to Reviewer HU2b (Part 2 / 4)**
>
> - **On W2 & Q1 (Hyperparameter Selection)**
>
>     **Empirical Selection in Experiments:**
>
>     1.  **For Bandwidth Parameter $\sigma$:** The selection of $\sigma$ is generally related to the number of support points $n$. In the Cross Space experiments ($n\sim 10^4$), we selected $\sigma = 1000$. In the Transfer Learning experiments ($n\sim 10^3$), we selected $\sigma = 200$. In the Molecule Generation experiments ($n\sim 30$), we selected $\sigma = 10$. Empirically, we find that maintaining $\sigma$ in the same order of magnitude as described above yields good results.
>     2.  **For Quantile Levels $q$:** For Cross Space and Molecule Generation experiments, since the position corresponding to $q=1$ varies frequently (different distributions often correspond to different points), we used a default $\delta = 0.1$ to avoid the influence of outliers. For Transfer Learning experiments, as the distributions are static and classes are similar, we set $\delta = 0$ to fully utilize information from all quantiles. Regarding the number of quantiles, while "more is better," the marginal benefit decreases; we generally select between 30 and 100.
>
>     **Ablation Study:**
>
>     We present ablation studies on Cross-Space and Transfer Learning tasks to demonstrate the impact of $\sigma$ and the number of quantiles $k$.
>
>     1. **Cross Space Experiments:** We considered $\sigma \in \\{200, 1000, 5000\\}$ and $k \in \\{20, 100, 500\\}$. The results are as follows:
>
>     |Parameters|TMSE 3D|TMSE 2D_1st|TMSE 2D_2nd|TMSE 2D_3rd|TMSE Avg.|IR 3D (%)|IR 2D_1st (%)|IR 2D_2nd (%)|IR 2D_3rd (%)|IR Avg. (%)|
>     |-|-|-|-|-|-|-|-|-|-|-|
>     |σ=200, k=20|0.22|0.25|0.23|0.26|0.24|70.49|62.79|67.70|62.82|65.95|
>     |σ=200, k=100|0.22|0.24|0.23|0.25|0.24|70.82|63.34|68.05|63.09|66.33|
>     |σ=200, k=500|0.22|0.24|0.23|0.25|0.24|70.79|63.36|68.05|63.09|66.32|
>     |σ=1000, k=20|0.22|0.24|0.23|0.25|0.24|70.68|63.68|68.11|62.82|66.32|
>     |σ=1000, k=100|0.22|0.24|0.23|0.25|0.24|71.15|63.75|68.49|63.20|66.65|
>     |σ=1000, k=500|0.22|0.24|0.23|0.25|0.24|71.12|63.73|68.46|63.18|66.62|
>     |σ=5000, k=20|0.22|0.24|0.23|0.26|0.24|70.83|64.46|68.07|62.71|66.52|
>     |σ=5000, k=100|0.22|0.24|0.23|0.25|0.23|71.35|64.04|68.52|63.22|66.78|
>     |σ=5000, k=500|0.22|0.24|0.23|0.25|0.23|71.33|63.97|68.50|63.23|66.76|
>     |**Summary(mean±std)**|**0.22±0.00**|**0.24±0.01**|**0.23±0.00**|**0.25±0.01**|**0.24±0.01**|**70.95±0.28**|**63.68±0.45**|**68.22±0.27**|**63.04±0.19**|**66.47±0.25**|
>
>     As shown above, the QDOT alignment results are not particularly sensitive to the selection of $\sigma$ and $k$. Generally, $k=100$ yields sufficiently good performance, with larger $k$ providing minimal additional gains.
>
>     2. **Transfer Learning Experiments:** We considered $n = 1024$, $\sigma \in \\{50, 200, 1000\\}$, and $k \in \\{10, 50, 200\\}$. The results are as follows:
>
>     |Params (Mo→Sh)|airplane|car|chair|guitar|lamp|laptop|table|Avg|
>     |-|-|-|-|-|-|-|-|-|
>     |σ=50,k=10|94.80|78.06|81.93|96.82|50.42|96.90|83.53|82.53|
>     |σ=50,k=50|95.46|82.74|82.70|96.44|50.81|96.67|84.50|83.46|
>     |σ=50,k=200|95.32|82.96|82.78|96.44|50.36|96.90|84.50|83.42|
>     |σ=200,k=10|94.94|80.07|83.10|97.46|50.10|96.45|84.03|83.12|
>     |σ=200,k=50|94.86|84.18|83.82|96.56|51.13|96.45|84.95|83.89|
>     |σ=200,k=200|95.65|84.41|84.11|96.32|50.23|97.12|85.30|84.15|
>     |σ=1000,k=10|94.98|80.73|82.01|97.71|48.48|95.57|82.68|82.26|
>     |σ=1000,k=50|95.35|84.08|84.78|95.93|50.81|97.12|84.90|84.14|
>     |σ=1000,k=200|95.72|85.63|84.73|96.19|50.55|96.90|85.18|84.37|
>     |**Summary(mean±std)**|**95.23±0.33**|**82.54±2.30**|**83.33±1.02**|**96.65±0.55**|**50.32±0.72**|**96.68±0.46**|**84.40±0.81**|**83.48±0.70**|
>
>
>     |Params (Sh→Mo)|airplane|car|chair|guitar|lamp|laptop|table|Avg|
>     |-|-|-|-|-|-|-|-|-|
>     |σ=50,k=10|98.62|83.84|89.79|97.65|79.17|95.27|94.92|92.58|
>     |σ=50,k=50|98.90|86.53|90.19|98.43|78.47|95.27|95.33|93.13|
>     |σ=50,k=200|98.76|88.89|90.09|98.82|79.86|95.86|95.73|93.49|
>     |σ=200,k=10|98.62|83.16|90.60|97.25|79.86|95.27|96.14|92.97|
>     |σ=200,k=50|98.89|87.20|91.10|98.43|81.94|97.04|96.74|93.97|
>     |σ=200,k=200|98.76|86.87|92.11|98.82|79.86|97.04|96.54|94.14|
>     |σ=1000,k=10|98.48|81.14|88.27|97.65|76.39|96.45|95.12|91.76|
>     |σ=1000,k=50|98.21|83.84|89.69|99.22|81.25|97.63|96.95|93.13|
>     |σ=1000,k=200|97.80|83.84|91.30|98.82|79.86|97.04|94.92|93.10|
>     |**Summary(mean±std)**|**98.56±0.34**|**85.03±2.31**|**90.35±1.04**|**98.34±0.63**|**79.63±1.50**|**96.32±0.87**|**95.82±0.75**|**93.14±0.67**|
>
>     The tables indicate that, except for slight numerical instability when $k=10$, the differences between results are not substantial. This demonstrates that our method is not overly sensitive to these parameters.
>
>     In summary, we conclude that **our parameters possess practical significance and low sensitivity; determining an approximate range based on the data structure is sufficient to achieve strong performance**.

---

> ### Author Response · Authors · 2025-11-24
> **Response to Reviewer HU2b (Part 3 / 4)**
>
> - **On W3.1 (More comparisons to sliced methods)**
>
>     We appreciate the suggestion. However, there is currently a lack of fast algorithms for measuring isometric invariant metrics. To the best of our knowledge, recent fast algorithms (e.g., Kerdoncuff et al., 2021; Scetbon et al., 2022; Li et al., 2023; Piening et al., 2025) typically entail a complexity of at least $O(n^2)$, which is difficult to apply in our experimental setting. Nevertheless, we have additionally considered several classic and recent sliced methods (Bonneel et al., 2016; Deshpande et al., 2019; Li et al., 2024). The results are as follows:
>
>     |Methods(Mo→Sh)|airplane|car|chair|guitar|lamp|laptop|table|Avg.|Time(h)|
>     |-|--|-|-|-|-|-|-|-|-|
>     |SW|96.69|95.77|94.81|0.00|68.71|97.78|89.60|86.03|17.36|
>     |MSW|96.25|95.55|95.53|0.00|67.29|98.23|89.91|86.09|17.05|
>     |HCP|93.38|95.55|95.85|0.25|74.73|98.23|91.29|86.90|22.58|
>     |**IQDOT**|95.35|87.86|85.01|96.44|52.55|98.00|87.15|85.42|5.98|
>
>     |Methods(Sh→Mo)|airplane|car|chair|guitar|lamp|laptop|table|Avg.|Time(h)|
>     |-|-|-|-|-|-|-|-|-|-|
>     |SW|100.00|95.96|96.76|0.00|93.06|95.86|97.76|89.36|17.36|
>     |MSW|99.86|95.62|97.67|0.00|93.75|96.45|96.54|89.45|17.05|
>     |HCP|98.48|96.63|97.47|0.00|92.36|97.63|98.78|89.52|22.58|
>     |**IQDOT**|98.89|91.91|93.22|98.03|77.77|97.63|97.56|95.05|5.98|
>
>     It is evident that IQDOT exhibits superior comprehensive performance.
>     Notably, for the guitar category, QDOT attains near-perfect accuracy, whereas other methods almost completely fail. As shown in Figure 11 (Lines 1458-1475), guitars in ModelNet are **horizontal** while those in ShapeNet are **vertical**, and this orientation mismatch breaks sliced-based methods, highlighting the importance of isometry invariance in transfer learning.
>
> - **On W3.2 (Ablation study on molecule generation experiments)**
>
>     We have conducted two additional ablation studies to disentangle the contributions of the **matching mechanism** versus the **structural constraints** inherent in our method.
>
>     We observe that the standard MSE loss assumes a fixed one-to-one correspondence between points, strictly constraining absolute positional information. In contrast, QDOT involves a re-matching process (via the last 1D-Wasserstein distance) and focuses on constraining intrinsic structural information. To isolate these effects, we designed the following variants:
>
>     1.  **Sliced Wasserstein (SW):** We utilize the Sliced Wasserstein distance on the point coordinates as the loss function. This allows for re-matching similar to QDOT, but it operates directly on positional information. This tests the benefit of the matching mechanism on coordinate space.
>     2.  **Representation MSE(RMSE):** We compute the QDOT representations (distances to QDMs) but strictly bypass the final 1D-Wasserstein matching step (i.e., we calculate the direct distance between the representations of the source and target based on their original indices). This acts as a structural regularizer that enforces geometric consistency but assumes the original one-to-one point correspondence (similar to MSE).
>
>     We compare these against the Baseline, standard MSE-0.3, and the IQDOT-0.3 method. The results are presented below:
>
>     |Method|Atom Sta. (%)|Mol Sta. (%)|Valid (%)|V * U (%)|
>     |-|-|-|-|-|
>     |UniGEM|98.90$_{\pm0.03}$|89.40$_{\pm0.02}$|94.58$_{\pm0.07}$|92.75$_{\pm0.11}$|
>     |MSE-0.3|99.00$_{\pm0.07}$|89.48$_{\pm0.70}$|95.16$_{\pm0.16}$|93.34$_{\pm0.18}$|
>     |SW-0.3|99.23$_{\pm0.02}$|92.76$_{\pm0.07}$|96.46$_{\pm0.01}$|91.86$_{\pm0.22}$|
>     |RMSE-0.3| 99.22$_{\pm0.02}$ | 92.98$_{\pm0.03}$ |96.68$_{\pm0.01}$| 91.71$_{\pm0.20}$ |
>     | **IQDOT-0.3** |99.44$_{\pm 0.01}$|95.23$_{\pm 0.18}$|97.94$_{\pm 0.06}$|83.88$_{\pm \text{0.30}}$|
>
>     As shown in the table above, both the SW and RMSE methods improve the stability of the generative model and the validity of the generated molecules. However, the magnitude of these improvements is lower than that of the IQDOT method. This corroborates our conclusion that IQDOT enhances model generalization via OT and augments the structural information of generated molecules through structural loss.

---

> ### Author Response · Authors · 2025-11-24
> **Response to Reviewer HU2b (Part 4 / 4)**
>
> - **On W4 (Relationship between QDOT and IQDOT)**
>
>     It is evident that both QDOT and IQDOT utilize the same QDMD representations in their numerical implementations. Since IQDOT is formulated as a pairwise 1-D Wasserstein distance, the inequality $\mathcal{IQD} \leq \mathcal{QD}$ consistently holds in practice. Consequently, the convergence $\mathcal{QD}(\mu_n, \mu) \rightarrow 0$ implies $\mathcal{IQD}(\mu_n, \mu)\rightarrow 0$. Furthermore, as both constitute distinct well-defined metrics, under the requisite conditions, they share the equivalence property regarding in distinguish ability: $\mathcal{QD}(\mu, \nu) = 0 \Leftrightarrow \mathcal{IQD}(\mu, \nu) = 0$. We have added these connections in Appendix B.4 (Lines 1153-1160).
>
> - **On W5 (Intuition)**
>
>     We appreciate the suggestion and have polished the expression from "intuition to formal definition." Briefly, we first elucidate via trilateration that anchors can **preserve distributional distance information**. We then introduce the distance to the barycenter as a **naturally isometric invariant** distribution. Finally, we derive a family of weights and corresponding QDMs by weighting at the quantiles of this isometric invariant distribution. The "centering assumption" is introduced to **construct the distance to the barycenter in Euclidean space.**
>     Regarding the notation issues, we have revised the relevant expressions at the beginning of Section 3.3 (Lines 143-175).
>
> - **On Q1:** Please refer to our response to **W2**.
> - **On Q2:** Please refer to our response to **W1**.
>
> In summary, we hope the above response satisfactorily addresses your concerns and reinforces your confidence in supporting our work in the following discussion and decision phases. Thanks in advance for your consideration.
>
> **References:**
>
> (Takeda et al., 2025) Takeda, S., & Akagi, Y. (2025). Gromov-Wasserstein Problem with Cyclic Symmetry. In Proceedings of the Computer Vision and Pattern Recognition Conference (pp. 21011-21020).
>
> (Kerdoncuff et al., 2021) Kerdoncuff, T., Emonet, R., & Sebban, M. (2021). Sampled gromov wasserstein. Machine Learning, 110(8), 2151-2186.
>
> (Scetbon et al., 2022) Scetbon, M., Peyré, G., & Cuturi, M. (2022). Linear-time gromov wasserstein distances using low rank couplings and costs. In International Conference on Machine Learning (pp. 19347-19365). PMLR.
>
> (Li et al., 2023) Li, M., Yu, J., Xu, H., & Meng, C. (2023). Efficient approximation of Gromov-Wasserstein distance using importance sparsification. Journal of Computational and Graphical Statistics, 32(4), 1512-1523.
>
> (Piening et al., 2025) Piening, M., & Beinert, R. (2025). A novel sliced fused Gromov-Wasserstein distance. arXiv preprint arXiv:2508.02364.
>
> (Bonneel et al., 2016) Bonneel, N., Rabin, J., Peyré, G., & Pfister, H. (2015). Sliced and radon wasserstein barycenters of measures. Journal of Mathematical Imaging and Vision, 51(1), 22-45.
>
> (Deshpande et al., 2019) Deshpande, I., Hu, Y. T., Sun, R., Pyrros, A., Siddiqui, N., Koyejo, S., ... & Schwing, A. G. (2019). Max-sliced wasserstein distance and its use for gans. In Proceedings of the IEEE/CVF conference on computer vision and pattern recognition (pp. 10648-10656).
>
> (Li et al., 2024) Li, T., Meng, C., Xu, H., & Yu, J. (2024). Hilbert curve projection distance for distribution comparison. IEEE Transactions on Pattern Analysis and Machine Intelligence, 46(7), 4993-5007.

---

> ### Author Response · Authors · 2025-11-27
> **Gentle Reminder**
>
> Dear Reviewer HU2b,
>
> We hope this message finds you well. As the discussion period is **drawing to a close**, we wanted to check if our updated manuscript and the new experimental results have fully resolved your initial concerns. Should you have any further questions or require additional clarification, please do not hesitate to let us know. We deeply value your constructive feedback and are committed to further refining our paper.
>
> Thank you for your dedicated time and effort in reviewing our work.

---

### Author Response · Authors · 2025-12-02
**Summary of the Rebuttal and Discussion**

Dear ACs, SACs, and PCs,

We sincerely thank you for the additional time and effort devoted to our paper. Given the length of the discussion, we summarize the key points below.

- **Main Replies**

    We summarize our responses to the common concerns as follows:

    1. **On the Dimensional Condition:** We clarified that the dimensional condition for QDOT is satisfied in most scenarios. Furthermore, an example shows that, by applying a "minimal unique piece" strategy, QDOT maintains its metric properties even on regular shapes, such as symmetric distributions, demonstrating broad applicability.
    2. **On Parameter Selection:** We explained the mechanism of the parameters and provided the specific values used in each experiment to guide hyperparameter selection. An ablation study further shows that our method is insensitive to parameter variations.

    We believe these concerns have been fully addressed and should no longer be considered weaknesses of the paper.

- **Replies to Specific Reviewers**

    We summarize the reviewers' evaluations, concerns, and our feedback below:

    1. **Reviewer HU2b** appreciated our theory, QDM design, and experimental results. Beyond the common issues, the reviewer suggested adding transfer learning baselines and ablation studies for molecule generation, asked about the relationship between IQDOT and QDOT, and provided writing suggestions. In response, we added sliced methods for transfer learning, included ablation studies for molecule generation, and answered the remaining questions.
    2. **Reviewer dG4E** recognized our computational performance and mainly raised concerns about the dimensional condition and the dependence of translational invariance on centering. In response, in addition to addressing the common issues, we introduced the barycenter in metric spaces and its isometric invariance, showing that centering in Euclidean space is a special case.
    3. **Reviewer r88Z** praised the novelty, theoretical grounding, efficiency, and experimental performance of our method. The reviewer pointed out the common issue of parameter selection, questioned the composition of the convergence rate, and suggested visualizing the molecule generation. In response, we addressed the common issues, explained that the convergence rate primarily depends on the Wasserstein term, and added the requested visualizations.
    4. **Reviewer kpCb** appeared to have significant misunderstandings. We clarified that our core contribution lies in metric properties and explained the role of IQDOT, and we advised against citing unreasonable submissions (specifically, comparing against `ICLR 2026 Conference Submission 7908`). Although Reviewer kpCb raised a large number of questions, most of which were extremely cherry-picked or trivial, we responded to every point to prevent potential misunderstandings for other readers.

    5. **Reviewer dUpr** recognized our work across multiple dimensions. The reviewer's concerns focused on conceptual and geometric understanding, computational difficulty, hyperparameter selection, and sample complexity. We addressed every issue raised. Following our response, the reviewer remained concerned about the sample convergence rate, noting that entropic methods can achieve an $O(n^{-1/2})$ rate. We thanked the reviewer for this insight and proved that QDOT using a Sinkhorn solver can also achieve this rate. Finally, we summarized the complexity, convergence rates, and practical recommendations for different versions of QDOT.

    In summary, we actively engaged with every comment from every reviewer, aiming to resolve their issues. Unfortunately, only Reviewer dUpr engaged in the discussion, and our exchange showed that we resolved their concerns. For Reviewers HU2b, dG4E, r88Z, kpCb, we believe we also addressed their concerns, although we did not receive timely confirmation from them.

- **Our Requests**

    For the reasons outlined above, we sincerely suggest the following during your evaluation:

    1. Please prioritize the comments and feedback from **Reviewer dUpr**, as they were the only reviewer who participated in the discussion (before 26 Nov 2025, 18:48, AOE).
    2. Please carefully review the comments from **Reviewers HU2b, dG4E, and r88Z** alongside our replies to verify that we have addressed their concerns.
    3. Please carefully evaluate the comments made by **Reviewer kpCb** in light of our clarifications and detailed answers. We hope that these extreme comments, based on misunderstandings, will not be grounds for rejection.

We understand that evaluating a paper with low reviewer engagement is a heavy task. However, we earnestly hope that our work will be treated fairly. We believe our efforts during the discussion period deserves more feedback. We sincerely hope you will consider our summary and requests above.

Thank you once again for the time and energy you have dedicated to reviewing this paper.

Best regards,

QDOT Authors

---

### Meta-Review · Area_Chair_zW3P · 2025-12-29

**Summary:**

The authors consider optimal transport problem for measures supported in different metric spaces. The authors propose quantile-weighted distance projection to map input distributions in different spaces to quantile-vector representation, and then leverage standard optimal transport to compared these quantile-vector representations, namely, Quantile-weighted Distance Optimal Transport (QDOT).

The Reviewers agree that the proposed quantile-vector representation with optimal transport is novel and is fast for computation. However, the Reviewers also have some critical concerns on the role of centering (e.g., vector-quantile representation based on norms/radial representation); translation/location invariance; dimension effects. Additionally, it is questionable whether the vector-quantile transformation is injective or not (i.e., different points, but having the same norm).

Overall, we think the contribution is interesting for the community. However, some key properties should be addressed rigorously, which may be beyond a simple revision, a more round of review seems necessary.

**Reviewer Concerns:**

The Reviewers have some following concerns:

+ Reviewer HU2b: requirement that QDMs span the full dimension; choice for key hyperparameters, e.g., vector quantile, bandwidth; missing sliced approach; relation of QDOT and IQDOT; representation by quantile vector (injective transform for representation?); centering assumption; dimension effects in empirical finding results;

+ Reviewer dG4E: dimensionality condition for theoretical finding results on metric properties; centering role for metric; claim on location invariance? translation invariance without pre-centering?

+ Reviewer r88Z: hyperparameters; relation of quantile-vector representation for theoretical finding results;

+ Reviewer kpCb: usage of one-dimensional Wasserstein distance in integral QDOT; weak experimental settings; weak analysis on empirical finding results; uncorrelated empirical setups? relation with GW

+ Reviewer dUpr: concept and computation of quantile-weighted representation and multiple layers of transformation; hyperparameters; sample complexity; geometric interpretation for quantile-weighted features;

**Reviewer Scores:**

The authors provide additional empirical results on hyperparameters; certain additional baselines; focus on metric properties rather than alignment.

We think the authors provide several additional empirical results for the hyperparameters. However, the role of centering is crucial. It is also better to analyze the dimension effects rigorously both in theory and experiments. The quantile-vector representation is not proved/guaranteed to be injective yet, which makes limitation for the theoretical finding results on metric properties; the translation/location invariance is not well-supported.

---

### Decision · Program_Chairs · 2026-01-26

Reject